# A deep catalogue of protein-coding variation in 983,578 individuals

Kathie Y. Sun[1,45], Xiaodong Bai[1,45], Siying Chen[1], Suying Bao[1], Chuanyi Zhang[1], Manav Kapoor[1], Joshua Backman[1], Tyler Joseph[1], Evan Maxwell[1], George Mitra[1], Alexander Gorovits[1], Adam Mansfield[1], Boris Boutkov[1], Sujit Gokhale[1], Lukas Habegger[1], Anthony Marcketta[1], Adam E. Locke[1], Liron Ganel[1], Alicia Hawes[1], Michael D. Kessler[1], Deepika Sharma[1], Jeffrey Staples[1], Jonas Bovijn[1], Sahar Gelfman[1], Alessandro Di Gioia[1], Veera M. Rajagopal[1], Alexander Lopez[1], Jennifer Rico Varela[1], Jesús Alegre-Díaz[2], Jaime Berumen[2], Roberto Tapia-Conyer[2], Pablo Kuri-Morales[2,3], Jason Torres[4], Jonathan Emberson[4], Rory Collins[4], Regeneron Genetics Center*, RGC-ME Cohort Partners*, Michael Cantor[1], Timothy Thornton[1], Hyun Min Kang[1], John D. Overton[1], Alan R. Shuldiner[1], M. Laura Cremona[1], Mona Nafde[1], Aris Baras[1], Gonçalo Abecasis[1], Jonathan Marchini[1], Jeffrey G. Reid[1], William Salerno[1✉] & Suganthi Balasubramanian[1✉]

Rare coding variants that substantially affect function provide insights into the biology of a gene[1–3]. However, ascertaining the frequency of such variants requires large sample sizes[4–8]. Here we present a catalogue of human protein-coding variation, derived from exome sequencing of 983,578 individuals across diverse populations. In total, 23% of the Regeneron Genetics Center Million Exome (RGC-ME) data come from individuals of African, East Asian, Indigenous American, Middle Eastern and South Asian ancestry. The catalogue includes more than 10.4 million missense and 1.1 million predicted loss-of-function (pLOF) variants. We identify individuals with rare biallelic pLOF variants in 4,848 genes, 1,751 of which have not been previously reported. From precise quantitative estimates of selection against heterozygous loss of function (LOF), we identify 3,988 LOF-intolerant genes, including 86 that were previously assessed as tolerant and 1,153 that lack established disease annotation. We also define regions of missense depletion at high resolution. Notably, 1,482 genes have regions that are depleted of missense variants despite being tolerant of pLOF variants. Finally, we estimate that 3% of individuals have a clinically actionable genetic variant, and that 11,773 variants reported in ClinVar with unknown significance are likely to be deleterious cryptic splice sites. To facilitate variant interpretation and genetics-informed precision medicine, we make this resource of coding variation from the RGC-ME dataset publicly accessible through a variant allele frequency browser.

Exome sequencing has enabled the discovery of rare coding variants, and has thus provided insights into gene function that have accelerated the pace of disease-associated gene discovery across Mendelian and common disorders[1–3,6,9–12]. Furthermore, exome sequencing has identified protective alleles that highlight drug targets that could be amenable to pharmacological intervention[2,13–17]. For example, anti-PCSK9 drug therapy is based on the observation that a loss of PCSK9 function is associated with reduced levels of cholesterol[18].

Cataloguing rare coding variation can help with the implementation of precision medicine[19,20]. Large datasets of genetic variation that are representative of the human population are essential for the comprehensive discovery and interpretation of rare variants. Roadmaps for numerous large-scale sequencing studies have been proposed, and several efforts are now underway[21–24]. The Genome Aggregation Database[4] (gnomAD) and Trans-Omics for Precision Medicine[8] (TOPMed) initiatives have developed large public databases of genetic variation derived from approximately 200,000 individuals and 132,000 individuals, respectively. Here, we describe a harmonized collection of exonic data derived from 983,578 individuals who represent a diverse array of ancestries. We calculate continental and fine-scale ancestry-based allele frequencies across this dataset and make the data publicly available through the RGC research browser: https://rgc-research.regeneron.com/me.

## Survey of variation in the RGC-ME dataset

The Regeneron Genetics Center Million Exome (RGC-ME) dataset contains the genetic variation observed in 983,578 individuals. These data span dozens of collaborations, including large biobanks and health systems. All data were generated by Regeneron Genetics Center using

[1]Regeneron Genetics Center, Tarrytown, NY, USA. [2]Faculty of Medicine, National Autonomous University of Mexico (UNAM), Mexico City, Mexico. [3]Instituto Tecnológico y de Estudios Superiores de Monterrey, Monterrey, Mexico. [4]Clinical Trial Service Unit and Epidemiological Studies Unit, Nuffield Department of Population Health, University of Oxford, Oxford, UK. [45]These authors contributed equally: Kathie Y. Sun, Xiaodong Bai. *Lists of authors and their affiliations appear at the end of the paper. ✉e-mail: william.salerno@regeneron.com; suganthi.bala@regeneron.com

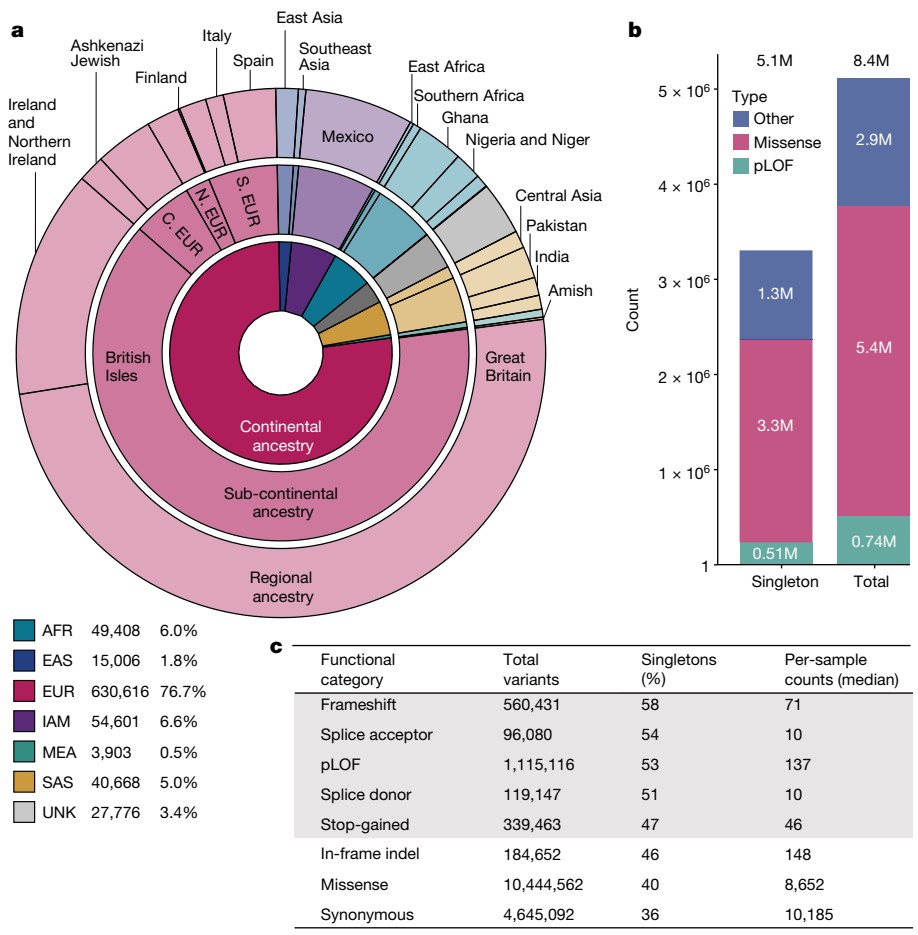

**Fig. 1 | Variant survey and population counts in the RGC-ME dataset.**
**a**, Summed proportional ancestry (sum of weighted ancestry probabilities) at continental, sub-continental and regional levels for 821,979 unrelated samples (Supplementary Table 1b). All subsequent variant counts and surveys have been performed in the unrelated analysis set. UNK, unknown. **b**, Count of variants unique to RGC-ME (that is, variants not in gnomAD v.3.1.2 genomes,

gnomAD v.2.1.1 exomes and TOPMed Freeze 8), broken down by singletons and variant functional category. M, million. **c**, Variant counts in different functional categories, proportion of singletons and per-individual median values. All counts were based on variants in the canonical transcript. pLOF includes frameshift, essential splice donor and acceptor (excluding splice sites in UTRs) and stop-gained variants.

a single harmonized sequencing and informatics protocol. Previously published datasets, such as the UK Biobank and the Mexico City Prospective Study, were reprocessed[9,25]. The RGC-ME dataset comprises both outbred and founder populations spanning African (AFR), European (EUR), East Asian (EAS), Indigenous American (IAM), Middle Eastern (MEA) and South Asian (SAS) continental ancestries, and includes cohorts with relatively high rates of consanguinity. More than 190,000 of the unrelated participants (23%) are of non-EUR ancestry in the RGC-ME dataset, as compared with 35,000 in gnomAD genomes (v.3.1.2), 53,000 in gnomAD exomes (v.2.1.1), and 91,000 in TOPMed Freeze 8, indicating that RGC-ME represents a large increase in the number of individuals of non-EUR ancestry in datasets of genetic variation[4,8] (Fig. 1a and Supplementary Table 1a).

We performed a comprehensive survey of genetic variation, encompassing single-nucleotide variants (SNV) and insertion–deletion (indel) variants. To estimate population allele frequencies, we focused on 821,979 unrelated samples (referred to hereafter as the 822K unrelated set; Supplementary Table 1a). We identified 16,425,629 unique mutated genomic positions (that is, sites) in autosomal and X-chromosomal coding regions, with one unique reference–alternate allele change (that is, variant) every two bases on average. In canonical transcripts within sequencing target regions, mutations at 35.6%, 32.2% and 9.5% of all possible genomic positions that can lead to synonymous, missense and stop-gained variants, respectively, were observed. In highly methylated CpG sites, we observed 95.0% of all possible synonymous,

92.2% of missense and 78.6% of stop-gained variants. Across all mutational contexts, 21.4% and 8.4% of all possible synonymous variants and stop-gained variants, respectively, were observed (Extended Data Fig. 1). Thus, RGC-ME represents a major advance towards the comprehensive discovery of rare variants.

Among coding variation in canonical transcripts, 1,115,116 pLOF variants were identified, which include those causing a premature stop, affecting essential splice donor and acceptor sites or causing frameshifts (Fig. 1c). Of these pLOF variants, 53.3% were observed as singletons; that is, only observed in one individual. In addition, 4,645,092 synonymous (35.7% as singletons) and 10,444,562 missense (40.0% as singletons) variants in canonical transcripts were detected. A total of 48% of coding variants in canonical transcripts were unique to RGC-ME and absent in other large-scale datasets[4,8] (Fig. 1b). Each sample had a median of 137 pLOF, 8,652 missense and 10,184 synonymous variants (Fig. 1c). AFR individuals had, on average, 18.6% more variants across all functional categories compared with individuals of other ancestries (Extended Data Fig. 2), as expected on the basis of the 'Out of Africa' model of human population history[26].

## Constrained genes

Population-scale sequencing allows the quantification of pLOF variation in genes, which is key to understanding the relationship between genes and diseases. Several gene constraint metrics have been developed

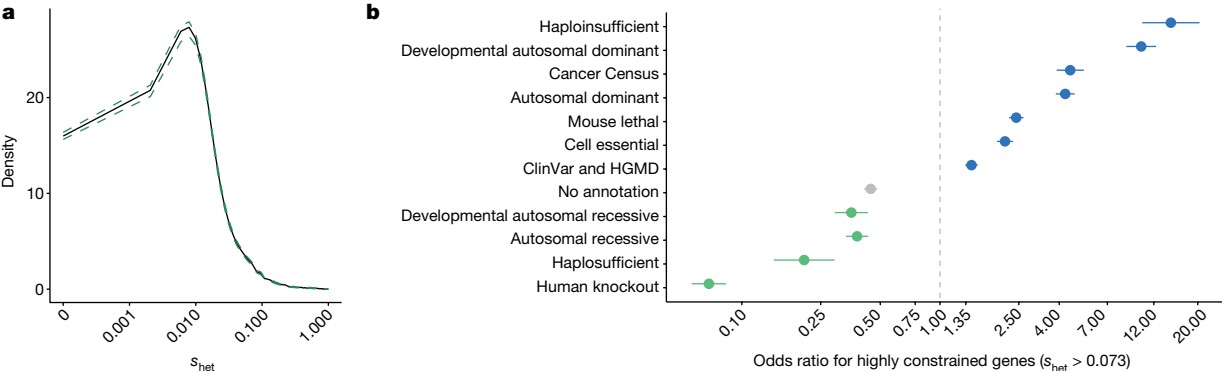

**Fig. 2 | Estimates of gene-level constraint, representing $s_{het}$, from the RGC-ME dataset. a**, Mean $s_{het}$ probability density for 16,710 canonical transcripts with 95% CIs calculated with 10,000 bootstrapped samples from the means of individual genes. **b**, Odds ratios (points) and 95% CIs (short horizontal lines; computed using standard error) for genes with $s_{het}$ cut-off > 0.073 (deemed highly constrained genes) to be included in each gene category listed on the

*y* axis compared with genes below the cut-off. Genes defined as 'human knockouts' are those with carriers of rare, biallelic pLOF variants observed in the RGC-ME dataset. A total of 16,710 canonical transcripts were included in each category, which contained at minimum 234 'true' genes (that category being haploinsufficient genes). HGMD, Human Gene Mutation Database.

to estimate the pLOF tolerance of genes[27]. Here, we estimated pLOF depletion using the cumulative frequency of pLOF variants in a gene to derive a selection coefficient, $s_{het}$, that quantifies fitness loss due to heterozygous pLOF variation[28]. We estimated the indispensability of 16,710 protein-coding genes on the basis of the observed number of rare pLOF variants per gene with a cumulative alternate allele frequency (AAF) of less than 0.1% compared with the expected number based on gene-specific mutation rates (Supplementary Table 2).

The mean $s_{het}$ value in the RGC-ME dataset for canonical transcripts was 0.073 (95% highest posterior density (HPD)): [0.043, 0.12] (median $s_{het}$ = 0.021) (Fig. 2a), which suggests that, on average, a pLOF would result in 7.3% lower evolutionary fitness relative to the reference allele. This estimate is comparable with the mean $s_{het}$ value of 0.073 [0.029, 0.18] (median = 0.028) computed using the same method on the ExAC dataset[29] ($n \approx 60,000$). Our sample size ($n \approx 822,000$) helped to accurately quantify rare pLOF variants and compute more precise constraint scores than the ExAC values. This finding is best illustrated in known haploinsufficient genes, which are expected to be more constrained and thus have larger $s_{het}$ values relative to all genes (Extended Data Fig. 3a). Compared with values that were computed with ExAC data, $s_{het}$ values for haploinsufficient genes in the RGC-ME dataset were significantly higher ($\Delta \bar{s}_{het}$ = 0.045, $P$ = 0.002) and had smaller 95% HPD ranges despite those larger means ($\Delta \text{Var}(s_{het})$ = −0.026, $P$ = 4.5 × 10$^{-21}$). Estimates for all genes were more precise in 822K samples compared with a randomly downsampled set of 60,000 samples from the RGC-ME dataset (Extended Data Fig. 3b), in which mean and median 95% HPD ranges were 6.2- and 4.0-fold larger, respectively.

The $s_{het}$ value is higher in genes that are associated with Mendelian diseases[28,30] (Extended Data Fig. 3a), and can differentiate groups of genes under varying degrees of selection (Fig. 2b). We used $s_{het}$ to identify constrained genes by comparing the $s_{het}$ scores of known high-constraint genes (haploinsufficient, autosomal dominant and developmental-specific autosomal dominant) with those of low-constraint genes (haplosufficient and genes with rare biallelic pLOF variants from the RGC-ME dataset) (Extended Data Fig. 4a). Among 1,476 genes in the 'high-constraint' and 3,893 genes in the 'low-constraint' groups, 89.1% of genes with a $s_{het}$ score greater than the mean (0.073) and 66.6% of genes with a $s_{het}$ score greater than the median (0.021) belonged to the high group (Supplementary Table 2). These thresholds served as cut-offs for mean and lower bound (2.5% HPD), respectively, to identify highly constrained genes with fitness deficits on a par with dominant disease-causing genes that also reflect uncertainty in the mean.

We compared $s_{het}$ to other published LOF constraint measures, such as LOEUF[4], and an alternate method for estimating $s_{het}$ based on approximate Bayesian computing[31], which we refer to as $s_{het\text{-}ABC}$ (Supplementary Figs. 2–4). Spearman rank correlations between $s_{het}$ from RGC-ME and these estimates were high (−0.768 with LOEUF; 0.778 with $s_{het\text{-}ABC}$). However, $s_{het}$ derived from RGC-ME had higher sensitivity and specificity in differentiating between constrained and unconstrained genes, compared with LOEUF and $s_{het\text{-}ABC}$ (Extended Data Fig. 4a).

Improving $s_{het}$ estimates is most valuable for genes with few expected pLOF mutations[4], particularly shorter genes. The RGC-ME dataset allowed $s_{het}$ to be estimated more precisely for the smallest quantiles of gene coding sequence (CDS) length (Extended Data Fig. 3c), and derived more informative constraint metrics using an allele-frequency-based approach and a larger sample. We derived constraint scores for 923 genes that had 5 or fewer expected pLOF variants, deemed underpowered for similar analyses with LOEUF[32]. These 923 underpowered genes were significantly shorter, with a mean CDS length of 573 base pairs compared to 1,797 base pairs for genes with more than 5 expected pLOF variants. Eighty-six genes were highly constrained, with a mean $s_{het}$ value greater than 0.073 and a lower bound greater than 0.021 (Extended Data Fig. 4b), and are promising candidates for efforts to discover new disease-associated genes. Thirty per cent (26 of 86) have been linked to human diseases or shown to be essential in mice or cell lines (Supplementary Table 2). These include well-studied genes with known importance in cellular function, such as the transcription factor TWIST1 (ref. 33), DNA- and RNA-binding protein BANF1 (refs. 34,35) and transactivator CITED2 (ref. 36).

Overall, 3,988 highly constrained genes had $s_{het}$ values greater than 0.073 and a lower bound greater than 0.021. Although 1,153 of these lack known associations with human diseases or lethal mouse knockout phenotypes, they are likely to have high functional importance. These constrained genes might lack disease associations because the loss of even a single copy is incompatible with life or causes reduced reproductive success without clinical disease[37].

## Constrained coding regions

Identifying sub-genic regions that are intolerant of mutations can reveal functionally important regions that would otherwise be missed when constraint scores are aggregated at the gene level. Models of local coding constraint are powerful tools for identifying protein domains with crucial functions and for variant prioritization[38–40]. In addition to gene constraint derived from pLOF variation, we also identified regions depleted of missense variation using the missense tolerance

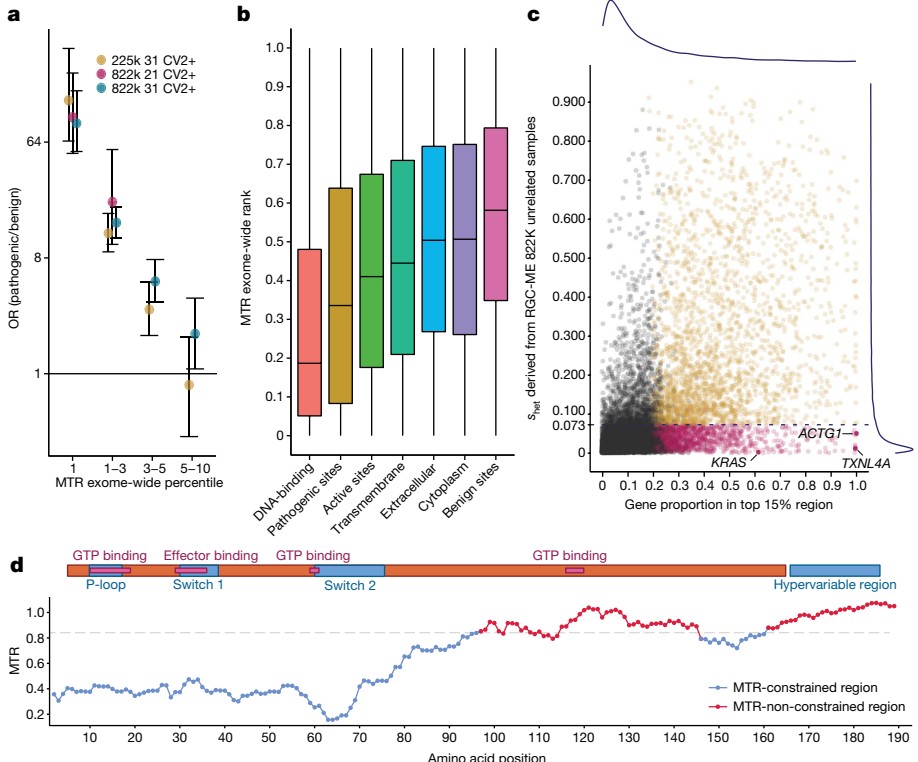

**Fig. 3 | Missense regional constraint captured by MTR. a**, Odds ratio (OR) (points) and 95% CIs (error bars, two-sided Fisher's exact test) of ClinVar pathogenic versus benign variants in MTR ranking regions across the whole exome. Comparisons include MTR calculated using the 822K unrelated samples from the RGC-ME dataset on 31-codon (blue) and 21-codon (pink) windows, and MTR calculated using a random subset of 225,000 samples from the larger 822K samples using a 31-amino-acid sliding window (yellow). MTR values include variants for which the FDR is less than 0.1. A total of 21,047 benign ClinVar variants and 12,872 pathogenic ClinVar variants (two stars or more (CV2+)) were included. **b**, MTR ranking distribution of different protein functional regions and variant groups (in order: DNA-binding sites ($n = 2,787$), ClinVar pathogenic sites ($n = 10,673$), active sites ($n = 2,787$), transmembrane region ($n = 2,787$), localized to extracellular ($n = 25,665$), localized to cytoplasm ($n = 32,994$) and ClinVar benign sites ($n = 20,739$)). Box plot shows median and 25–75% interquartile range. The whisker minima and maxima represent the smallest and largest data points within 1.5× the interquartile range from the lower quartile and upper quartile, respectively. Every functional category

was significantly more constrained than the category to its right with two-sided Wilcoxon rank-sum test (least significant p-value = $2 \times 10^{-4}$, Bonferroni correction). **c**, Distribution of the proportion of genes located in exome-wide top-15-percentile MTR regions against the heterozygous selection coefficient, $s_{het}$. Genes with a significant proportion in the most constrained 15-percentile MTR region are coloured in pink and yellow ($P < 0.05$, Bonferroni-corrected, one-sided binomial tests with $\pi_0 = 0.15$), stratified by LOF constraint ($s_{het} = 0.073$). Pink dots highlight genes that are tolerant of LOF, but have some regions depleted of missense variation. Blue lines indicate the density of genes with $s_{het}$ scores from 0 to 1 (right margin) and genes with a proportion of MTR in the top 15 percentile exome wide (above plot). **d**, MTR track of an oncogene, *KRAS*, a missense-specific constrained gene, along with the domain structure of the protein. The blue MTR-constrained region is defined by top-15-percentile exome-wide MTR rank. The N-terminal region containing amino acids 1–80 is depleted of missense variation, even though *KRAS* is tolerant of heterozygous LOF variation ($s_{het} = 0.002$).

ratio (MTR)[40,41], defined as the ratio of the observed to the expected proportion of missense variants adjusted by synonymous variation in a defined codon window. Using the 822K unrelated samples, we calculated the MTR for each amino acid along the CDS within sliding windows of 21 and 31 amino acids (MTR scores available on figshare; see 'Data availability') and characterized continuous segments of missense constrained regions (Supplementary Table 3).

Compared with benign missense variants, ClinVar pathogenic missense (two stars or more) variants were highly enriched in the top percentile of exome-wide MTR scores (odds ratio = 100.0 and 89.8, computed with 21- and 31-codon windows, respectively; Fig. 3a). Our sample size, which is nearly four times larger than that used in previous MTR estimates[41], resulted in improved discrimination between pathogenic and benign variants for top-10-percentile MTR scores in which we observed significant enrichment (Fig. 3a). This larger sample size enabled us to identify 24% more missense variants in the top-1-percentile constrained MTR scores (512,499 versus 413,147) compared with a subsampled set of 225,000, after adjusting for a false discovery rate (FDR) lower than 0.1. In addition, the increased power derived from 822K samples resulted in higher resolution for distinguishing pathogenic

from benign variants for MTR computed with 21-codon windows, albeit at the expense of having fewer scored missense variants overall (295,958 constrained missense variants).

Deleterious variants are expected to have lower allele frequencies than neutral variants, owing to negative selection. We can infer the functional importance of different classes of variation by comparing the proportion of singletons in each class. We computed the deleteriousness of variants using an updated mutability-adjusted proportion of singletons (MAPS) metric[5,32] and derived an MTR score threshold at which their MAPS score corresponds to that of missense variants that were predicted to be deleterious by five out of five prediction algorithms in dbNSFP (v.3.2; see Supplementary Information); that is, 5/5 missense variants. Variants with MTR values in the top-15-percentile exome-wide threshold (MTR < 0.841) were predicted to be as deleterious as 5/5 missense variants (Extended Data Fig. 5a). For 31-codon windows, 1.24% (129,990) of all missense variants (excluding known ClinVar pathogenic variants) observed in the RGC-ME dataset had significant MTR scores in the top 15 percentile. These missense variants in the top 15 percentile of exome-wide MTR are potentially deleterious and could be suitable for prioritization in projects aiming to discover disease-associated genes.

MTR is a useful metric of regional constraint that may capture functionally important segments within genes. We defined MTR-constrained regions as continuous regions within a protein that have variants with MTR values in the top-15-percentile threshold (Supplementary Information and Extended Data Fig. 6a). We identified 41,114 missense constrained regions in 12,349 genes (Supplementary Table 3). Our findings overlap with results from a previous study[38] that estimated the regional observed-to-expected missense ratio ($\gamma$) from around 60,000 ExAC samples (Extended Data Fig. 6b) to derive a composite missense deleterious score called MPC. We refer to the MPC-derived constrained regions as MPC segments, and compared these with MTR-constrained regions. MTR-constrained regions had a median length of 22 residues [14–35, quartile 1–quartile 3], compared with 358 [208, 579] in MPC segments. Overall, we identified 8.59 times more MTR-constrained regions than MPC segments ($\gamma \leq 0.612$, top 15 percentile) across 2,832 transcripts with data from both methods (Extended Data Fig. 6c).

We examined the distribution of de novo missense variants in MTR-constrained regions and observed a significant enrichment ($P = 2.61 \times 10^{-10}$) of variants identified in individuals with neurodevelopmental disorders (Extended Data Fig. 7a,b). Case variants were 1.85 times [1.50, 2.31 (95% CI)] more likely to occur in constrained regions, compared with controls. As expected, well-supported (two stars or more) ClinVar pathogenic missense variants were also highly enriched ($P \approx 0$) in MTR-constrained regions. Pathogenic variants were 8.82 times [8.17, 9.53 (95% CI)] more likely to occur in missense constrained regions than were benign variants.

Missense constrained sites were found in key functional regions, such as DNA-binding regions and active sites (Fig. 3b). Among membrane proteins, transmembrane regions ranked higher in MTR-constrained regions than did cytoplasmic and extracellular domains. We also compared the overlap of MTR-constrained and functional regions by computing Jaccard indices. Ubiquitin-conjugating (UBC) core domains and DNA-binding regions had the highest overlap with constrained regions (Jaccard index = 0.52 and 0.18, respectively), suggesting that, among UBC enzymes, more than half of the union set between MTR-constrained regions and core domains overlapped. Other enriched functional regions included protein kinases and nuclear receptor ligand-binding domains (Supplementary Table 4).

A total of 4,064 genes contained regions depleted in missense variation with a significant proportion of their coding sequence in the top 15 percentile of MTR (binomial test with $\pi_0 = 0.15$, $P < 0.05$ after multiple testing correction; Supplementary Table 5). To identify genes with signatures of missense-only constraint, we assessed the LOF-constraint metric, $s_{het}$, of these highly missense constrained genes (Fig. 3c). Among the 4,064 genes, 1,482 either were not LOF constrained or lacked $s_{het}$ estimates. These genes had significantly shorter CDS lengths than those of the 1,424 LOF-specific constrained genes ($P = 2.9 \times 10^{-40}$, Wilcoxon test; Extended Data Fig. 7c). Estimating region-level LOF constraint is difficult owing to strong selection against pLOF variants, which leads to a paucity of pLOF variation. MTR serves as a complementary lens for identifying, first, functionally important regions at a higher resolution than gene-level LOF constraint, and, second, regions within genes that are depleted of missense variation but tolerant of LOF variation. For example, *KRAS*, a well-known oncogene, is LOF tolerant ($s_{het} = 0.002$, LOEUF = 1.24); however, the first 80 amino acids (42%) of the protein sequence were ranked in the top 1 percentile of exome-wide MTR (Fig. 3d). This region includes the P-loop, switch 1 and switch 2 functional domains, which form crucial binding interfaces for effector proteins[42], and these results therefore highlight the importance of regional constraint metrics.

## Understanding 'human knockouts'

Identifying genes with biallelic pLOF variants provides an opportunity to understand gene function directly through the phenotypic

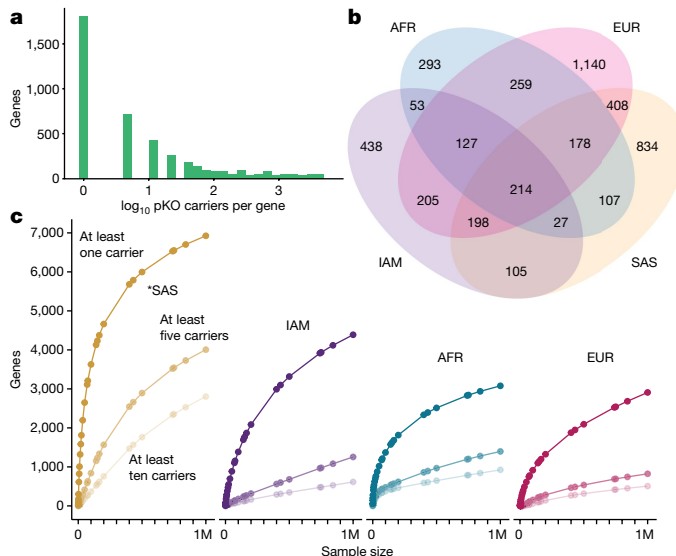

**Fig. 4 | Rare biallelic pLOF variants and 'human knockouts' in the RGC-ME dataset. a**, Distribution of the number of individuals per pKO on the $\log_{10}$ scale. Carriers of homozygous pLOFs and compound heterozygous variants were included in this analysis. **b**, Breakdown of the number of unique pKO genes observed in the RGC-ME dataset by ancestry. Both sets of rare biallelic variants—homozygous pLOFs and compound heterozygous—were included in this analysis. See Extended Data Table 1a for a breakdown by ancestry of each type. **c**, Projected accrual of pKO genes using homozygous pLOF variant data at hypothetical cohort sizes for each ancestry in 983,578 related individuals. Curves reflect the accrual of the expected number of genes with at least one, at least five and at least ten carriers, respectively, of a rare, homozygous pLOF. Asterisk denotes the inclusion of cohorts with a high rate of consanguinity.

characterization of individuals who have such variants—effectively, naturally occurring 'human knockouts'. The RGC-ME dataset includes founder populations and cohorts with high rates of consanguinity, contributing to a comprehensive collection of homozygous loss-of-function variation[25,43–45]. Overall, we identified 4,686 genes comprising 8,576 rare (AAF < 1%) homozygous pLOF variants in 64,852 individuals (Supplementary Table 6). Furthermore, we identified 1,205 genes with carriers of rare (AAF < 1%) heterozygous pLOF variants in *trans*; that is, compound heterozygotes, 162 of which lacked homozygous pLOFs. In total, 4,848 genes were discovered with carriers of biallelic pLOF variants in which both alleles of a gene were affected by pLOF variation and could be described as putative gene knockouts (pKOs). Of these, 1,751 (1,650 from homozygous pLOFs only) have not to our knowledge been previously reported. Biallelic pLOF variants in RGC-ME are rare; 64.3% of homozygous pLOF variants and 37.4% of pKOs were detected in one participant (Fig. 4a). As expected, cohorts with higher rates of consanguinity were enriched in homozygous pLOF variants, compared with outbred populations, despite smaller sample sizes (Fig. 4b,c and Extended Data Table 1).

pKOs were significantly less constrained, with a lower $s_{het}$ (on average −0.074 [−0.077, −0.071 (95% CI)], *t*-test) relative to all other genes. Only 2.67% of pKOs had an $s_{het}$ value greater than 0.073, as compared with 21.6% of all human genes, and 47.2% of pKOs were in the lowest quintile of $s_{het}$ scores exome-wide ($s_{het} < 7.07 \times 10^{-3}$). A caveat is that $s_{het}$, like most gene-specific measures of constraint, is designed to capture the effect of heterozygous LOF[46]. Although genes containing biallelic pLOF variants are under less heterozygous selective pressure, existing sample sizes are inadequate[47] to directly compute selection on homozygous variation. pKOs are overrepresented in drug and xenobiotic metabolism pathways (Supplementary Fig. 6).

Among very rare doubleton variants for which we observed exactly two copies of the alternate allele, we observed a clear excess of

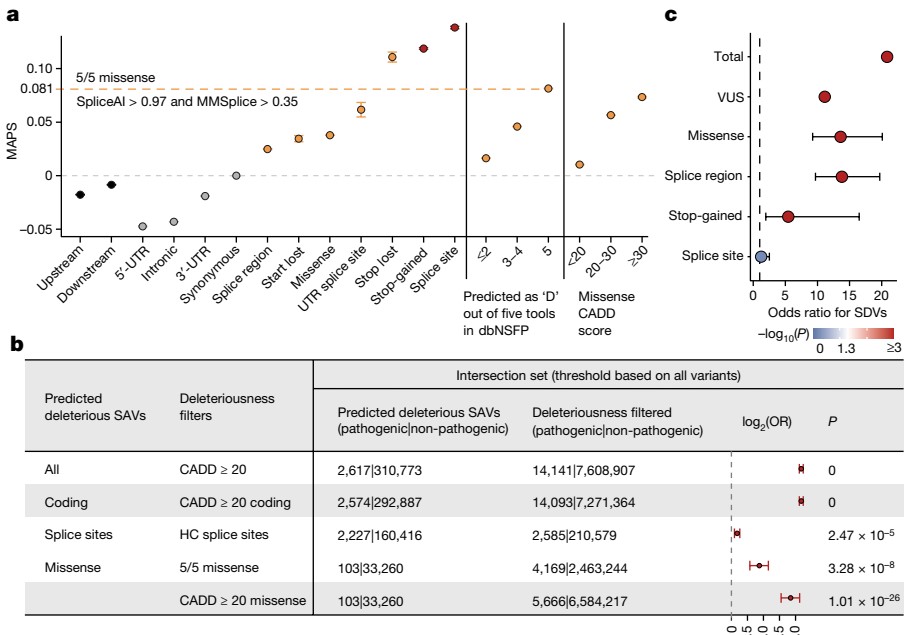

**Fig. 5 | Identification of deleterious variants that are predicted to affect splicing. a**, MAPS across different functional categories. Error bars show standard deviation around the mean proportion of singletons (points). The yellow dashed line represents the SpliceAI and MMSplice score threshold for variants that have a MAPS score equal to that of 5/5 missense variants (predicted deleterious by five algorithms). Variants with a SpliceAI score ≥ 0.35 or a MMSplice score ≥ 0.97 are predicted deleterious SAVs. Noncoding variants refers to intronic, downstream (variant located 5′ of a gene), upstream (variant located 3′ of a gene) and 5′ and 3′ UTR variants captured by exome sequencing. Coding variants are inclusive of canonical splice sites, splice region and UTR splice sites. All variants that passed quality control and were observed in unrelated individuals in the RGC-ME dataset were included in this analysis

($n$ = 34,512,842 variants). **b**, Enrichment of ClinVar pathogenic variants (two stars or more) in predicted SAVs compared with corresponding variant sets filtered by either LOFTEE, 5/5 missense deleteriousness models or CADD. Points represent odds ratios and bars depict 95% CIs (two-sided Fisher's exact test, no multiple testing correction). 'All variants' include 313,390 coding and noncoding variants, and 'splice sites' include essential and UTR splice sites; counts of variants included in each calculation are provided in the table. HC, high-confidence. **c**, Empirical validation of MAPS-predicted deleterious SAVs (intersection set): enrichment of predicted deleterious SAVs in experimentally validated SDVs compared with non-SDVs. Points represent odds ratios and bars depict 95% CIs (two-sided Fisher's exact test, no multiple testing correction). A total of $n$ = 36,636 variants, of which 346 SAVs are validated SDVs, are included.

homozygotes that is likely to be explained by population structure and background inbreeding. For example, among missense and synonymous variants, we observed 5,857 and 2,490 homozygotes among 1,580,917 and 679,335 doubleton variants, respectively, compared with a Hardy–Weinberg equilibrium (HWE) expectation of fewer than one homozygote in each case. These estimates corresponded to a background inbreeding coefficient of 0.37%. Among pLOF variants, we observed only 406 homozygotes among 129,405 doubleton variants (Supplementary Table 7). Although this number is much larger than HWE expectations, it is around 15% less than the expected 479 homozygotes calculated using an inbreeding coefficient of 0.37% ($P$ = 0.0095, Fisher's exact test). This suggests that a notable proportion of these homozygotes were never observed in our sample population.

Genes with biallelic inactivating mutations could reveal potential drug targets that can be disrupted with minimal side effects[43]. Drug targets with homozygous pLOF variants in humans are more likely to progress from phase I trials to approval[44]. Of 997 inhibitory preclinical targets listed in the Drug Repurposing Hub, 182 (18.3%) had at least one individual with a rare biallelic pLOF variant in the RGC-ME dataset[48]. In-depth phenotyping of human knockouts can help researchers to better understand the efficacy and side-effect profiles of these potential drug targets. Human knockouts provide a way to understand the consequences of lifelong deficiency of a gene[49].

## Annotation of splice-affecting variants

Several prediction tools[50–53] have been developed to understand the effects of genetic variants on alternative splicing. Although these tools mainly assess whether a variant affects splicing, some also provide a

pathogenicity metric or score threshold as a measure of deleteriousness. Predicted cryptic splice sites with SpliceAI scores greater than 0.8 have been validated at high rates using RNA sequencing and are as depleted at common allele frequencies as pLOF variants[50]. Here, we used human genetic data to optimize splice prediction score thresholds enriched for deleterious variants that affect splicing. We systematically quantified the deleteriousness of variants at various splice prediction score thresholds using the MAPS metric. As previously demonstrated[2,4], pLOF variants had the highest MAPS scores, followed by missense, synonymous and noncoding variants, respectively (Fig. 5a).

We used splice predictions from SpliceAI[50] and MMSplice[51] to group variants into predicted splice score bins, and identified the minimum threshold at which the MAPS score of the variants is equal to that of 5/5 missense variants (variants predicted to be deleterious by five out of five prediction methods). The proposed prediction score thresholds of 0.35 for SpliceAI and 0.97 for MMSplice pathogenicity (Fig. 5a and Extended Data Fig. 5b) identify predicted deleterious splice-affecting variants (SAVs).

A total of 296,696 predicted deleterious coding SAVs (inclusive of canonical splice sites, splice region and untranslated region (UTR) splice sites) in the RGC-ME dataset had scores that exceeded the MAPS-derived splicing thresholds for both SpliceAI and MMSplice (referred to as the intersection set; Extended Data Fig. 8a). Of these, 43.5% (129,118) were cryptic splice sites (that is, non-canonical splice sites). Unsurprisingly, canonical splice sites and variants within the splice region comprised the largest category of predicted deleterious SAVs. Both SpliceAI and MMSplice identified around 80% of LOFTEE (loss of function transcript effect estimator; ref. 4) high-confidence splice sites and around 10% of variants within splice regions as predicted

deleterious SAVs (Extended Data Fig. 8a,b). In addition, around 68% of LOFTEE low-confidence splice sites were predicted to be deleterious SAVs (94% of low-confidence splice sites were in the UTR). The impact of non-canonical splice variants on alternative splicing is often underestimated; we found that missense variants accounted for 11.3% of all predicted deleterious SAVs identified by both SpliceAI and MMSplice in the RGC-ME dataset (Extended Data Fig. 8a,b).

Predicted deleterious SAVs were enriched in well-supported ClinVar pathogenic variants (two stars or more) compared with other metrics of variant deleteriousness (Fig. 5b); for example, compared with combined annotation dependent depletion (CADD)[54,55] score $\geq 20$ (odds ratio = 4.5, $P = 0$). Missense SAVs were significantly enriched for pathogenic variants compared with 5/5 missense variants (odds ratio = 1.8, $P = 3.3 \times 10^{-8}$) and missense variants with CADD $\geq 20$ (odds ratio = 3.6, $P = 1.01 \times 10^{-26}$), respectively. Notably, splice sites in the intersection set were also significantly enriched for pathogenic variants compared to LOFTEE high-confidence splice sites, indicating that the MAPS-derived metric identifies deleterious splice sites. Similar results were obtained when we evaluated the enrichment of pathogenic variants compared with benign ones (Supplementary Table 8a).

We next assessed the MAPS-derived splice prediction thresholds for variants that have been experimentally assessed for splicing effects[56–58] (Supplementary Table 9). Predicted deleterious SAVs identified in the intersection set were significantly enriched in experimentally validated large-effect splice-disrupting variants (SDVs) compared with non-SDVs in all functional categories except the splice site category, although the odds ratio was greater than one for splice sites (Fig. 5c). Variants of unknown significance (VUSs) in ClinVar that were predicted as deleterious SAVs were also significantly enriched in experimentally validated SDVs (Fig. 5c). Of the 563 predicted deleterious SAVs assayed in the experimental data, 346 (61.5%) were SDVs and more than half were cryptic splice sites, including 13 ClinVar VUSs (Extended Data Fig. 8d).

We also derived stringent thresholds to identify SAVs by removing canonical splice sites and calibrating exclusively coding non-splice-site (nonSS) variants to a MAPS score comparable with 5/5 missense variants. These thresholds corresponded to a SpliceAI score of 0.43 and an MMSplice score of 0.97 (Extended Data Fig. 5c). Pathogenic enrichment was consistent when comparing deleterious coding nonSS and missense SAVs with corresponding variant categories filtered by CADD $\geq 20$ (Supplementary Table 8b,c). Consistent results were also obtained when comparing the enrichment of deleterious SAVs in SDVs to non-SDVs after applying thresholds for coding nonSS variants (Extended Data Fig. 8c,e).

## Clinical utility of rare variants

To understand the prevalence of disease-associated alleles in the general population, we identified well-supported ClinVar[59] pathogenic variants (two stars or more) across 2,042 genes in 822K unrelated RGC-ME samples. We found that 40.7% of pathogenic variants (20,343/49,990) were observed in the RGC-ME dataset, of which 99.6% (20,262) had an AAF of less than 0.1% and 17.8% (3,619) were observed once. In comparison, 20% (9,821) and 29% (14,700) of pathogenic variants were observed in ExAC exomes ($n \approx 60,000$) and gnomAD v.2.1.1 exomes ($n \approx 126,000$), respectively (Extended Data Fig. 9a). This highlights the importance of the RGC-ME dataset's larger sample size in identifying rare pathogenic variants. On average, individuals carry 1.58 pathogenic variants, with the majority of these individuals being heterozygous carriers of these variants. Specifically, 61.4% of the 822K unrelated individuals were heterozygous carriers of pathogenic recessive alleles in 1,143 of 2,659 known autosomal recessive genes (mean, 0.98 pathogenic alleles per person); 0.21% of the samples were homozygotes of pathogenic variants in 167 autosomal recessive genes; and 3.64% were heterozygous carriers of 353 of 1,629 total autosomal dominant genes. Pathogenic variant annotations should be interpreted cautiously owing to the incomplete penetrance of disease alleles[60].

The American College of Medical Genetics identified a set of genes (ACMG SF v.3.1) with clinically actionable variants that predispose individuals to diseases and for which medical interventions are available to reduce mortality and morbidity[61]. Among the 822K unrelated individuals, 22,846 (2.77%) had at least one ClinVar-reported (two stars or more) pathogenic missense or pLOF variant for 72 out of 76 autosomal genes on the ACMG list (Supplementary Table 10). As expected, two of the most prevalent pathogenic variations were the *HFE* Cys282Tyr allele (enriched in EUR, $n_{\text{EUR-homozygotes}} = 3,220$ and $AAF_{\text{EUR}} = 13.8\%$) and the *TTR* Val142Ile allele (enriched in AFR, $n_{\text{AFR}} = 1,670$ and $AAF_{\text{AFR}} = 3.4\%$).

We also tallied carriers of likely pathogenic pLOF variants (novel variants not yet reported as pathogenic in ClinVar) in 44 genes in which truncation is known to lead to disease. A total of 2,357 (0.3%) individuals in the RGC-ME dataset carried 1,407 likely pathogenic variants across 40 of these genes. In total, 3.06% of the individuals in the RGC-ME dataset were carriers of pathogenic or likely pathogenic variants. Excluding individuals with high-frequency pathogenic variants in the *HFE* (Cys282Tyr) and *TTR* (Val142Ile) genes, 2.38% of the individuals in the RGC-ME dataset carried an actionable variant (Supplementary Table 10). This number is comparable with those from other reports[6,7,62] of actionable variants, which range from 2% to 4.1% for gene sets that include ACMG v.2.0 and v.3.0. As expected, pathogenic variants are rare in large-scale studies of the general population. We found that 39% and 79% of pathogenic and likely pathogenic variants, respectively, were singletons. Focusing on non-ACMG genes, we found that 1.27% of individuals were heterozygous carriers of pathogenic variants in autosomal dominant genes, and 0.21% were homozygotes of pathogenic variants in autosomal recessive genes.

Because the RGC-ME dataset includes uniformly processed exome data from a relatively large proportion of individuals from continental ancestries other than EUR, we assessed the range of allele frequencies of variants present in ClinVar across four continental populations: AFR, EUR, IAM and SAS. Approximately 34% of unique pathogenic coding variants in equalized subsamples were observed only in individuals of non-EUR ancestry, which indicates that sampling diverse populations is necessary for the comprehensive identification of rare variation. Across all unrelated individuals, on average, those of EUR ancestry had 63% more pathogenic variants that were well characterized (rated two stars or more) per sample than did individuals of AFR ancestry. Conversely, individuals of EUR ancestry had, per sample, 25.6% fewer VUSs (Extended Data Fig. 9b,c) and 18.6% fewer variants across all functional types (Extended Data Fig. 2). In individuals of AFR ancestry, a consistent pattern of significantly fewer high-confidence (two stars or more) pathogenic variants ($-0.576$ [$-0.567$, $-0.585$ (95% CI)], $t$-test) to a surplus of VUSs (42.13 [421.97, 42.28]), compared with individuals of EUR ancestry, suggests that the most well-characterized pathogenic variants were depleted in this population (Extended Data Fig. 9b). Recruiting diverse individuals to enable the identification and characterization of novel pathogenic variants might help to address this ascertainment bias. Further analyses of pathogenic coding variants and differentiated alleles between ancestries are included in Supplementary Fig. 7 and Supplementary Table 11.

Understanding VUSs is currently a bottleneck in the interpretation of variation in clinically relevant genes and a challenge in clinical management[19]. Although VUSs have less empirical evidence for pathogenicity, they comprise the bulk of ClinVar, with more than one million variants. Notably, VUSs in regions of low MTR may be deleterious, comprising 5,079 (0.68%) VUSs in the top 1 percentile of MTR-constrained regions and 17,500 VUSs (2%) in the top 15 percentile (Supplementary Table 12). Using the MAPS-derived splicing score thresholds, we identified more than 11,000 candidate deleterious cryptic splice sites among VUSs (1,366 synonymous variants in 822 genes and 10,407 missense variants in 3,501 genes), offering potential insights into their functional consequences for clinical prioritization and interpretation efforts.

## Discussion

The RGC-ME dataset, derived from 983,578 exomes, provides a harmonized catalogue of around 20 million coding variants in individuals from a diverse array of ancestries and is publicly accessible at https://rgc-research.regeneron.com/me/home. Cataloguing variation at scale provides an opportunity to accurately estimate the frequency of rare variants—allowing us to precisely compute gene and regional constraint metrics, expand the compendium of rare human knockouts, annotate deleterious cryptic splice sites, characterize variant frequencies across different ancestries and assess the population prevalence of pathogenic variation. RGC-ME will be an invaluable resource for interpreting rare variants and is a step towards the realization of precision medicine.

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

**Regeneron Genetics Center**

**RGC Management and Leadership Team**
Gonçalo Abecasis[1], Aris Baras[1], Michael Cantor[1], Giovanni Coppola[1], Andrew Deubler[1], Aris Economides[1], Adolfo Ferrando[1], Luca A. Lotta[1], John D. Overton[1], Jeffrey G. Reid[1], Alan Shuldiner[1] & Katherine Siminovitch[1]

**Sequencing and Lab Operations**
John D. Overton[1], Christina Beechert[1], Erin D. Brian[1], Laura M. Cremona[1], Hang Du[1], Caitlin Forsythe[1], Zhenhua Gu[1], Kristy Guevara[1], Michael Lattari[1], Alexander Lopez[1], Kia Manoochehri[1], Prathyusha Challa[1], Manasi Pradhan[1], Raymond Reynoso[1], Ricardo Schiavo[1], Maria Sotiropoulos Padilla[1], Chenggu Wang[1] & Sarah E. Wolf[1]

**Clinical Informatics**
Michael Cantor[1], Amelia Averitt[1], Nilanjana Banerjee[1], Dadong Li[1], Sameer Malhotra[1], Justin Mower[1], Mudasar Sarwar[1], Deepika Sharma[1], Jeffrey C. Staples[1], Sean Yu[1] & Aaron Zhang[1]

**Genome Informatics and Data Engineering**
Jeffrey G. Reid[1], Mona Nafde[1], George Mitra[1], Sujit Gokhale[1], Andrew Bunyea[1], Krishna Pawan Punuru[1], Sanjay Sreeram[1], Gisu Eom[1], Sujit Gokhale[1], Benjamin Sultan[1], Rouel Lanche[1], Vrushali Mahajan[1], Eliot Austin[1], Sean O'Keeffe[1], Razvan Panea[1], Tommy Polanco[1], Ayesha Rasool[1], William Salerno[1], Xiaodong Bai[1,45], Lance Zhang[1], Boris Boutkov[1], Evan Edelstein[1], Alexander Gorovits[1], Ju Guan[1], Lukas Habegger[1], Alicia Hawes[1], Olga Krasheninina[1], Samantha Zarate[1], Adam J. Mansfield[1], Evan K. Maxwell[1], Suganthi Balasubramanian[1], Suying Bao[1], Kathie Sun[1] & Chuanyi Zhang[1]

**Analytical Genetics and Data Science**
Gonçalo Abecasis[1], Manuel Allen Revez Ferreira[1], Joshua Backman[1], Kathy Burch[1], Adrian Campos[1], Lei Chen[1], Sam Choi[1], Amy Damask[1], Sheila Gaynor[1], Benjamin Geraghty[1], Arkopravo Ghosh[1], Salvador Romero Martinez[1], Christopher Gillies[1], Lauren Gurski[1], Joseph Herman[1], Eric Jorgenson[1], Tyler Joseph[1], Michael Kessler[1], Jack Kosmicki[1], Nan Lin[1], Adam Locke[1], Priyanka Nakka[1], Jonathan Marchini[1], Karl Landheer[1], Olivier Delaneau[1], Maya Ghoussaini[1], Anthony Marcketta[1], Joelle Mbatchou[1], Arden Moscati[1], Aditeya Pandey[1], Anita Pandit[1], Charles Paulding[1], Jonathan Ross[1], Carlo Sidore[1], Eli Stahl[1], Maria Suciu[1], Timothy Thornton[1], Peter VandeHaar[1], Sailaja Vedantam[1], Scott Vrieze[1], Jingning Zhang[1], Rujin Wang[1], Kuan-Han Wu[1], Bin Ye[1], Blair Zhang[1], Andrey Ziyatdinov[1], Yuxin Zou[1], Olivier Delaneau[1], Maya Ghoussaini[1], Jingning Zhang[1], Kyoko Watanabe[1] & Mira Tang[1]

**Therapeutic Area Genetics**
Adolfo Ferrando[1], Giovanni Coppola[1], Luca A. Lotta[1], Alan Shuldiner[1], Katherine Siminovitch[1], Brian Hobbs[1], Jon Silver[1], William Palmer[1], Rita Guerreiro[1], Amit Joshi[1], Antoine Baldassari[1], Cristen Willer[1], Sarah Graham[1], Ernst Mayerhofer[1], Mary Haas[1], Niek Verweij[1], George Hindy[1], Jonas Bovijn[1], Tanima De[1], Parsa Akbari[1], Luanluan Sun[1], Olukayode Sosina[1], Arthur Gilly[1], Peter Dornbos[1], Juan Rodriguez-Flores[1], Moeen Riaz[1], Manav Kapoor[1], Gannie Tzoneva[1], Momodou W. Jallow[1], Anna Alkelai[1], Giovanni Coppola[1], Ariane Ayer[1], Veera Rajagopal[1], Sahar Gelfman[1], Vijay Kumar[1], Jacqueline Otto[1], Neelroop Parikshak[1], Aysegul Guvenek[1], Jose Bras[1], Silvia Alvarez[1], Jessie Brown[1], Jing He[1], Hossein Khiabanian[1], Joana Revez[1], Kimberly Skead[1] & Valentina Zavala[1]

**Research Program Management and Strategic Initiatives**
Lyndon J. Mitnaul[1], Marcus B. Jones[1], Esteban Chen[1], Michelle G. LeBlanc[1], Jason Mighty[1], Nirupama Nishtala[1], Nadia Rana[1], Jennifer Rico-Varela[1] & Jaimee Hernandez[1]

**Senior Partnerships and Business Operations**
Alison Fenney[1], Randi Schwartz[1], Jody Hankins[1] & Samuel Hart[1]

**Business Operations and Administrative Coordinators**
Ann Perez-Beals[1], Gina Solari[1], Jaimee Hernandez[1], Johannie Rivera-Picart[1], Michelle Pagan[1] & Sunilbe Siceron[1]

**RGC-ME Cohort Partners**

**Accelerated Cures**
David Gwynne[5]

**African Descent and Glaucoma Evaluation Study (ADAGES) III**
Jerome I. Rotter[6] & Robert Weinreb[6]

**Age-related macular degeneration in the Amish**
Jonathan L. Haines[7], Margaret A. Pericak-Vance[8] & Dwight Stambolian[9]

**Albert Einstein College of Medicine**
Nir Barzilai[10], Yousin Suh[10] & Zhengdong Zhang[10]

**Amish Connectome Project**
Elliot Hong[11]

**Amish Research Clinic**
Braxton Mitchell[11]

**The Australia and New Zealand MS Genetics Consortium**
Nicholas B. Blackburn[12], Simon Broadley[13], Marzena J. Fabis-Pedrini[14,15], Vilija G. Jokubaitis[16], Allan G. Kermode[15], Trevor J. Kilpatrick[17], Jeanette Lechner-Scott[18], Stephen Leslie[19], Bennet J. McComish[12], Allan Motyer[19], Grant P. Parnell[20], Rodney J. Scott[18], Bruce V. Taylor[12] & Justin P. Rubio[17]

**Center for Non-Communicable Diseases (CNCD)**
Danish Saleheen[21]

**Cincinnati Children's Hospital**
Ken Kaufman[22], Leah Kottyan[22], Lisa Martin[22] & Marc E. Rothenberg[22]

**Columbia University**
Abdullah Ali[23] & Azra Raza[23]

**Dallas Heart Study**
Jonathan Cohen[24]

**Diabetic Retinopathy Clinical Research (DRCR) Retina Network**
Adam Glassman[25]

**Duke University**
William E. Kraus[26], Christopher B. Newgard[26] & Svati H. Shah[26]

**Flinders University of South Australia**
Jamie Craig[27] & Alex Hewitt[27]

**Indiana Biobank**
Naga Chalasani[28], Tatiana Foroud[28] & Suthat Liangpunsakul[28]

**Indiana University School of Medicine**
Nancy J. Cox[29], Eileen Dolan[30], Omar El-Charif[30], Lois B. Travis[28], Heather Wheeler[31] & Eric Gamazon[29]

**Kaiser Permanente**
Lori Sakoda[32] & John Witte[32]

**Mayo Clinic**
Kostantinos Lazaridis[33]

**Mexico City Prospective Study (MCPS)**
Jesús Alegre-Díaz[2], Jaime Berumen[2], Rory Collins[4], Jonathan Emberson[4], Pablo Kuri-Morales[2,3], Roberto Tapia-Conyer[2] & Jason Torres[4]

**MyCode-DiscovEHR Geisinger Health System Biobank**
Adam Buchanan[34], David J. Carey[34], Christa L. Martin[34], Michelle N. Meyer[34], Kyle Retterer[34] & David Rolston[34]

**National Institute of Mental Health**
Nirmala Akula[35], Emily Besançon[35], Sevilla D. Detera-Wadleigh[35], Layla Kassem[35], Francis J. McMahon[35], Thomas G. Schulze[35],

**Northwestern University**
Adam Gordon[36], Maureen Smith[36] & John Varga[36]

**Penn Medicine Biobank**
Yuki Bradford[9], Scott Damrauer[9], Stephanie DerOhannessian[9], Theodore Drivas[9], Scott Dudek[9], Joseph Dunn[9], Ned Haubein[9], Renae Judy[9], Yi-An Ko[9], Colleen Morse Kripke[9], Meghan Livingstone[9], Nawar Naseer[9], Kyle P. Nerz[9], Afiya Poindexter[9], Marjorie Risman[9], Salma Santos[9], Giorgio Sirugo[9], Julia Stephanowski[9], Teo Tran[9], Fred Vadivieso[9], Anurag Verma[9], Shefali S. Verma[9], JoEllen Weaver[9], Colin Wollack[9], Daniel J. Rader[9] & Marylyn Ritchie[9]

**Primary Open-Angle African American Glaucoma Genetics (POAAG) study**
Joan O'Brien[9]

**Regeneron–Mt. Sinai BioMe Biobank**
Erwin Bottinger[37] & Judy Cho[37]

**UAB GWAS in African Americans with rheumatoid arthritis**
S. Louis Bridges[38]

**UAB Whole exome sequencing of systemic lupus erythematosus patients**
Robert Kimberly[38]

**University of California, Los Angeles**
Marlena Fejzo[39]

**University of Colorado School of Medicine**
Richard A. Spritz[40]

**University of Michigan Medical School**
James T. Elder[41,42], Rajan P. Nair[41], Philip Stuart[41] & Lam C. Tsoi[41]

**University of Ottawa**
Robert Dent[43] & Ruth McPherson[43]

**University of Pennsylvania**
Brendan Keating[9]

**University of Pittsburgh**
Erin E. Kershaw[25], Georgios Papachristou[25] & David C. Whitcomb[25]

**University of Texas Health Science Center at Houston**
Shervin Assassi[44] & Maureen D. Mayes[44]

**Vanderbilt University Medical Center**
Eric D. Austin[29]

[5]Accelerated Cures, Waltham, MA, USA. [6]Lundquist Institute, Torrance, CA, USA. [7]Case Western Reserve University, Cleveland, OH, USA. [8]University of Miami, Miami, FL, USA. [9]University of Pennsylvania, Philadelphia, PA, USA. [10]Albert Einstein College of Medicine, Bronx, NY, USA. [11]University of Maryland School of Medicine, Baltimore, MD, USA. [12]Menzies Institute for Medical Research, University of Tasmania, Hobart, Tasmania, Australia. [13]Griffith University, Gold Coast, Queensland, Australia. [14]Murdoch University, Perth, Western Australia, Australia. [15]Perron Institute for Neurological and Translational Science, Nedlands, Western Australia, Australia. [16]Monash University, Melbourne, Victoria, Australia. [17]Florey Institute of Neuroscience and Mental Health, Melbourne, Victoria, Australia. [18]University of Newcastle, Newcastle, New South Wales, Australia. [19]Melbourne Integrative Genomics, School of Mathematics and Statistics, University of Melbourne, Melbourne, Victoria, Australia. [20]University of Sydney, Sydney, New South Wales, Australia. [21]Center for Non-Communicable Diseases, Karachi, Pakistan. [22]Cincinnati Children's Hospital, Cincinnati, OH, USA. [23]Columbia University, New York City, NY, USA. [24]UT Southwestern Medical Center, Dallas, TX, USA. [25]University of Pittsburgh, Pittsburgh, PA, USA. [26]Duke University, Durham, NC, USA. [27]Flinders University of South Australia, Adelaide, South Australia, Australia. [28]Indiana University School of Medicine, Indianapolis, IN, USA. [29]Vanderbilt University Medical Center, Nashville, TN, USA. [30]University of Chicago, Chicago, IL, USA. [31]Loyola University, Chicago, IL, USA. [32]Kaiser Permanente, Oakland, CA, USA. [33]Mayo Clinic, Rochester, MN, USA. [34]Geisinger Health System, Danville, PA, USA. [35]National Institute of Mental Health, National Institutes of Health, Bethesda, MD, USA. [36]Northwestern University, Chicago, IL, USA. [37]Mount Sinai School of Medicine, New York City, NY, USA. [38]University of Alabama at Birmingham, Birmingham, AL, USA. [39]University of California, Los Angeles, Los Angeles, CA, USA. [40]University of Colorado School of Medicine, Aurora, CO, USA. [41]University of Michigan Medical School, Ann Arbor, MI, USA. [42]Ann Arbor Veterans Affairs Hospital, Ann Arbor, MI, USA. [43]University of Ottawa, Ottawa, Ontario, Canada. [44]University of Texas Health Science Center at Houston, Houston, TX, USA.

## Reporting summary

Further information on research design is available in the Nature Portfolio Reporting Summary linked to this article.

## Data availability

Genetic variation data for 821,979 unrelated individuals are made publicly available through the RGC-ME browser (https://rgc-research. regeneron.com/me/home). Features include genomic locations, alleles, fine-scale ancestry assignments, population-specific allele frequencies and functional annotations for the genetic variants. In addition, vcf files can be downloaded from the web portal. Exome-wide MTR scores are available for download from figshare: https://doi.org/10.6084/ m9.figshare.24587328 (ref. 63). The human reference genome GRCh38 can be obtained from ftp://ftp-trace.ncbi.nlm.nih.gov/1000genomes/ ftp/technical/reference/GRCh38_reference_genome/GRCh38_full_anal- ysis_set_plus_decoy_hla.fa. Ensembl Release 100 gene and transcript builds can be accessed from https://ftp.ensembl.org/pub/release-100/ gtf/homo_sapiens/ and corresponding gene and transcript reference nucleotide and protein sequence data from https://ftp.ensembl.org/ pub/release-100/fasta/homo_sapiens/. Individual-level sequence data have been deposited with the UK Biobank and are freely available to approved researchers. Instructions for access to UK Biobank data are available at https://www.ukbiobank.ac.uk/enable-your-research. Information about the data access policy for researchers interested in the Mexico City Prospective Study data can be found at https:// www.ctsu.ox.ac.uk/research/prospective-blood-based-study-of-150- 000-individuals-in-mexico. Regeneron can make GHS individual- level genomic data available to qualified academic noncommercial researchers through the Regeneron pre-clinical Research portal at https:// regeneron.envisionpharma.com/vt_regeneron/ under a data access agreement. Information about the data access policy, procedures and contact details for the cohorts included in this dataset can be obtained through the URLs given in the RGC-ME browser at https://rgc-research. regeneron.com/me/data-contributors. This information is also pro- vided in Supplementary Table 1c, with relevant references if available.

## Code availability

Publicly available software and packages used in this study are described in the Supplementary Information. In summary, sequenc- ing reads were generated using bcl2fastq v.2.20 and were mapped to references using BWA-MEM v.0.7.17. Variants were identified using DeepVariant v.0.10, aggregated with GLnexus v.1.4.3 and converted to bed, bim or fam format using PLINK v.1.9. Variants were annotated with VEP (Ensembl, v.100.4) and pLOF variants were further classified with the VEP LOFTEE plug-in. Array variants were phased using Eagle v.2.4 and imputed using Minimac4. PLINK v.2 was used for principal compo- nent analysis and to compute *F*st, a measure of genetic differentiation among population groups. The csq function in BCFtools v.1.18 was used to annotate in-frame indels resulting from a combination of frameshift indels on the same haplotype, and bedtools v.2.30.0 was used to deter- mine the genetic context and neighbouring nucleotides of variants. Picard LiftoverVcf v.3.0.0 was used to transform sequence coordi- nates to GRCh38. Relatedness was determined with PRIMUS: https:// primus.gs.washington.edu/primusweb/res/documentation.html. We adapted scripts from https://github.com/pjshort/dddMAPS to compute updated MAPS metrics. For identifying compound heterozygous vari- ants, exome variants were merged with a well-imputed common variant backbone and phased using SHAPEIT5 (https://github.com/odelaneau/ shapeit5). Large-scale data manipulation used Scala v.2.12 on a 10.4 LTS

runtime (Apache Spark v.3.2.1) with standard Spark functions. Beyond standard R packages, visualization tools and data-processing libraries (for example, dplyr, ggplot2 and data.table), we used rstan (v.2.33) to build Bayesian hierarchical models for calculating heterozygous selection coefficients, rmutil (v.4.1.2) to project LOF accrual and boot (v.4.1.1) for bootstrapping. Python code used standard packages (for example, scipy, numpy and pandas) for analysis, scikit-learn (v.1.0) to model variant quality (see Supplementary Tables 13 and 14) and sqla- lchemy (v.2.0.23) to store and query tables. Custom code to generate LOF projection curves is available at https://github.com/rgcgithub/ rgc_me_analysis.

63. Sun, K. Exome-wide MTR scores computed with RGC-ME data for all possible missense variants in canonical transcripts. *figshare* https://doi.org/10.6084/m9.figshare.24587328 (2024).

**Acknowledgements** This work was supported in part by the Intramural Research Program of the National Institute of Mental Health (ZIA-MH002843) and a grant from R01 NCI R01 CA157823. Ethical approval for the UK Biobank was previously obtained from the North West Centre for Research Ethics Committee (11/ NW/0382). The work described herein was approved by the UK Biobank under application number 26041. Informed consent was obtained for all study participants. Approval for Geisinger Health System MyCode analyses was provided by the Geisinger Health System Institutional Review Board under project number 2006-0258. Informed consent was obtained for all study participants. Appropriate consent for the Penn Medicine BioBank was obtained from each participant regarding the storage of biological specimens, genetic sequencing and genotyping and access to all available EHR data. This study was approved by the Institutional Review Board of the University of Pennsylvania and complied with the principles set out in the Declaration of Helsinki. All individuals participating in the Mayo–RGC project generation provided informed consent for the use of specimens and data in genetic and health research and ethical approval for project generation was provided by the Mayo Clinic Institutional Review Board (09-007763). All research performed in this study used de-identified data (without any Protected Health Information data) with no possibility of re-identifying any of the participants. Approval for the Indiana Biobank was provided by the Indiana University Institutional Review Board under project number 1105005445. For participants in the Mexico City Prospective Study, approval for the study was given by the Mexican Ministry of Health, the Mexican National Council of Science and Technology (0595 P-M) and the Central Oxford Research Ethics Committee (C99.260) and the Ethics and Research commissions from the Medicine Faculty at the National Autonomous University of Mexico (UNAM) (FMED/CI/SPLR/067/2015). All study participants provided written informed consent. Study participants were recruited from the BioMe Biobank Program of the Charles Bronfman Institute for Personalized Medicine at Mount Sinai Medical Center from 2007 onward. The BioMe Biobank Program (Institutional Review Board 07-0529) operates under a Mount Sinai Institutional Review Board-approved research protocol. All study participants provided written informed consent. The authors thank the RGC-ME Cohort Partners for contributions to this initiative. A list of cohorts (with links to the projects where available) and the data contributors can be accessed from the RGC-ME browser (https://rgc-research. regeneron.com/me/data-contributors). The authors thank everyone who made this work possible, the professionals from the member institutions who contributed to and supported this work and, most especially, all of the participants, without whom this research would not be possible. This study is funded by Regeneron Genetics Center and Regeneron Pharmaceuticals.

**Author contributions** Conceptualization: S. Balasubramanian, W.S., J.G.R., J.M., G.A., A.B. and H.M.K. Data generation: X.B., W.S., J.G.R., J.D.O, M.C., E.M., A.G., A. Mansfield, B.B., L.H., A. Marcketta, A.E.L. and A.L. Formal analysis: K.Y.S., S.C., S. Bao, C.Z., M.K., J. Backman, T.J., L.G. and A.H. Browser application: E.M., A.G., B.B., G.M., S.G. and M.N. Supervision: S. Balasubramanian and W.S. Writing (original draft): K.Y.S., X.B., S.C., S. Bao, M.K. and S. Balasubramanian. Writing (review and editing): K.Y.S., S. Balasubramanian, S. Bao, J.G.R., A.R.S., G.A., J.M., M.L.C., M.D.K., D.S., J.S., J. Backman, S.G., A.D.G., V.M.R., J.A., J. Bovijn, R.T.-C., P.K.-M., J.T., J.E., R.C., T.T. and C.Z. Project management: J.R.V.

**Competing interests** K.Y.S., X.B., S.C., S. Bao, C.Z., M.K., J. Backman, T.J., E.M., G.M., A.G., A. Mansfield, B.B., S.G., L.H., A. Marcketta, A.E.L., L.G., A.H., M.D.K., D.S., J.S., J. Bovijn, S.G., A.D.G., V.M.R., A.L., J.R.V., M.C., T.T., J.D.O., A.R.S., M.L.C., M.N., A.B., G.A., J.M., J.G.R., W.S. and S. Balasubramanian are current employees and/or stockholders of Regeneron Genetics Center or Regeneron Pharmaceuticals. H.M.K. was a former employee of Regeneron Genetics Center.

**Additional information**
**Correspondence and requests for materials** should be addressed to William Salerno or Suganthi Balasubramanian.

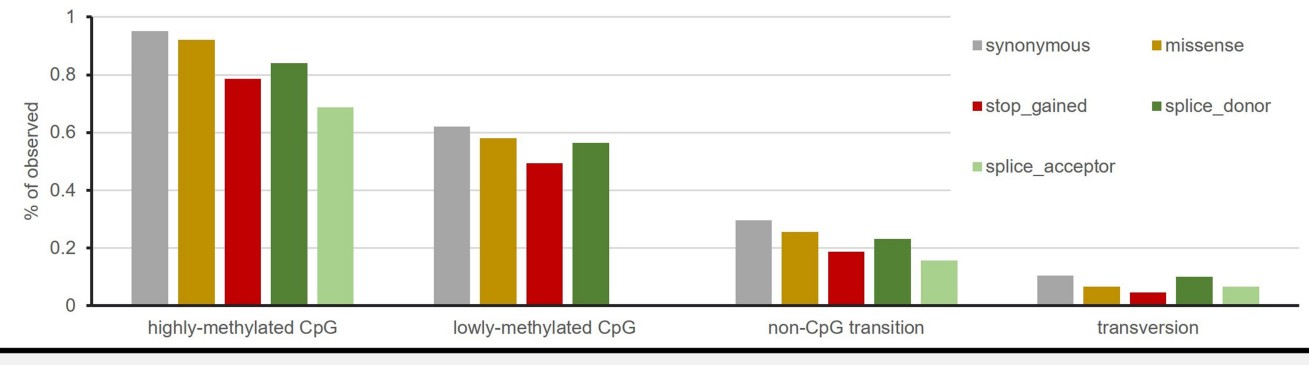

| Effect | Variant type | Methylation level* | # Observed | # Singletons | # Possible | % Observed | % Singletons |
|---|---|---|---|---|---|---|---|
| **synonymous** | CpG transition | 0 | 89,044 | 20,349 | 154,429 | 57.66% | 22.85% |
| | | 1 | 37,375 | 4,116 | 49,282 | 75.84% | 11.01% |
| | | 2 | 405,065 | 5,507 | 426,265 | 95.03% | 1.36% |
| | non-CpG transition | | 2,922,155 | 998,223 | 9,882,062 | 29.57% | 34.16% |
| | transversion | | 1,149,817 | 437,225 | 10,970,265 | 10.48% | 38.03% |
| **missense** | CpG transition | 0 | 130,568 | 32,974 | 246,054 | 53.06% | 25.25% |
| | | 1 | 61,225 | 7,801 | 84,582 | 72.39% | 12.74% |
| | | 2 | 822,365 | 24,695 | 892,153 | 92.18% | 3.00% |
| | non-CpG transition | | 4,801,333 | 1,745,547 | 18,821,504 | 25.51% | 36.36% |
| | transversion | | 4,550,200 | 1,879,483 | 50,792,179 | 8.96% | 41.31% |
| **splice_acceptor** | CpG transition | 0 | 0 | 0 | 4 | 0.00% | |
| | | 1 | 0 | 0 | 1 | 0.00% | |
| | | 2 | 11 | 0 | 16 | 68.75% | 0.00% |
| | non-CpG transition | | 14,153 | 6,655 | 90,720 | 15.60% | 47.02% |
| | transversion | | 12,480 | 6,110 | 186,473 | 6.69% | 48.96% |
| **splice_donor** | CpG transition | 0 | 91 | 38 | 203 | 44.83% | 41.76% |
| | | 1 | 96 | 14 | 129 | 74.42% | 14.58% |
| | | 2 | 1,213 | 66 | 1,442 | 84.12% | 5.44% |
| | non-CpG transition | | 20,717 | 8,738 | 89,528 | 23.14% | 42.18% |
| | transversion | | 19,428 | 9,358 | 195,347 | 9.95% | 48.17% |
| **stop_gained** | CpG transition | 0 | 1,509 | 517 | 4,082 | 36.97% | 34.26% |
| | | 1 | 1,863 | 261 | 2,750 | 67.75% | 14.01% |
| | | 2 | 41,834 | 3,954 | 53,251 | 78.56% | 9.45% |
| | non-CpG transition | | 143,649 | 65,779 | 765,157 | 18.77% | 45.79% |
| | transversion | | 149,284 | 71,567 | 3,204,397 | 4.66% | 47.94% |
| **frameshift** | insertion | | 391,470 | 204,038 | | | 52.12% |
| | deletion | | 169,515 | 87,282 | | | 51.49% |

**Extended Data Fig. 1 | Mutation saturation survey in RGC-ME data.** Counts are based on variant-transcript pairs. *Methylation level: mean methylation values across tissues of >0.65, 0.2–0.65, <0.2 correspond to methylation level of 2, 1, 0, respectively. CpG sites with methylation level of 2 are highly methylated, sites with methylation level 0 or 1 are grouped as lowly methylated sites. Variants are subset to target exome regions. Only variants that passed QC are included in the number of all possible variants.

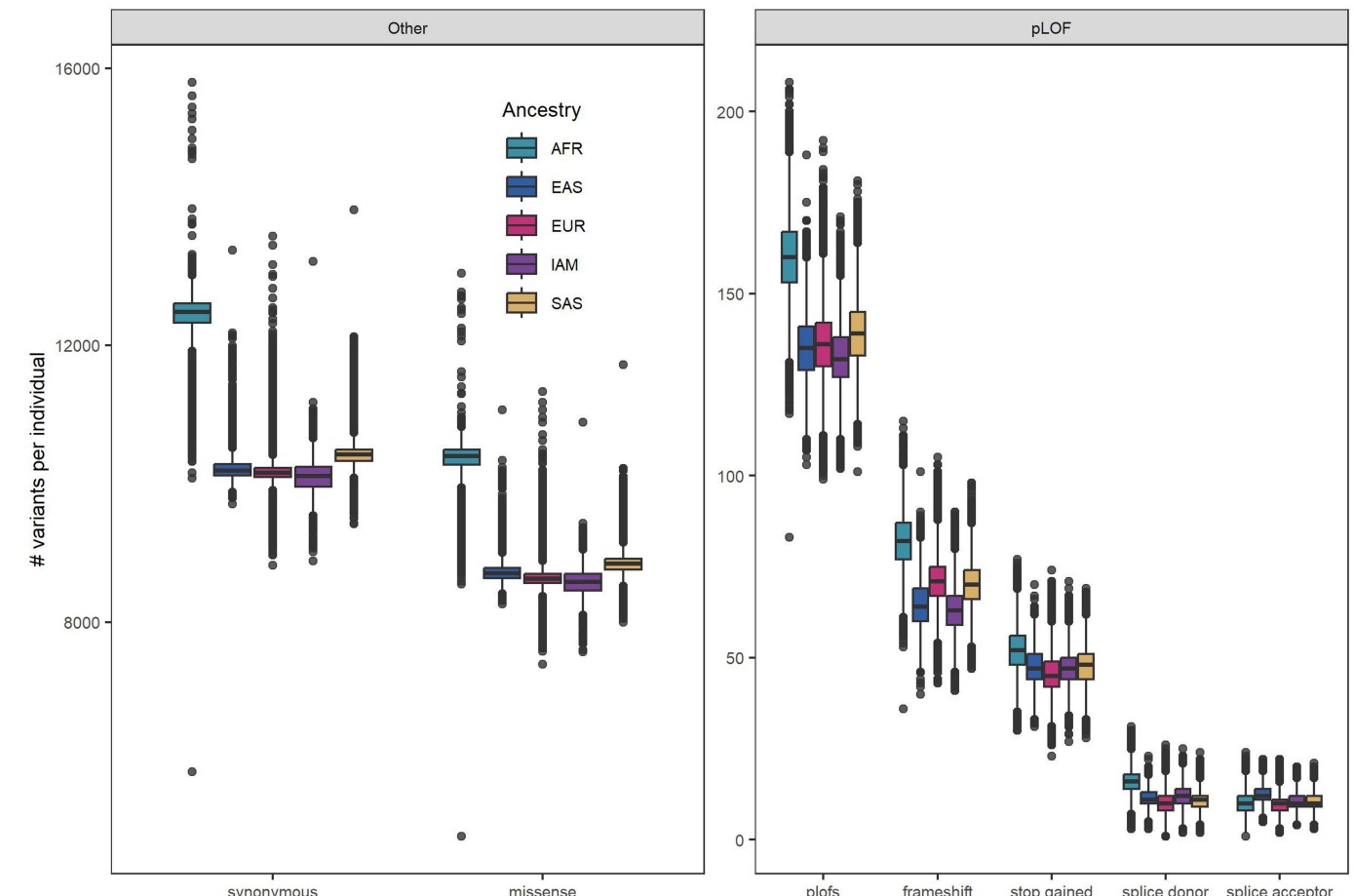

**Extended Data Fig. 2 | Box plots summarizing counts of variants observed in each unrelated sample in the RGC-ME dataset.** The lower bound, centre and upper bound of each box plot represents the 25, 50, and 75 percentiles of the distributions of counts. Points represent outliers, and whisker minima and maxima represent the smallest and largest points 1.5-times beyond the interquartile range. Individuals were assigned discrete ancestries based on overall fine-scale ancestry probabilities >50% and a total of 755,261 individuals were included. See Supplementary Table 1 for sample size breakdowns.

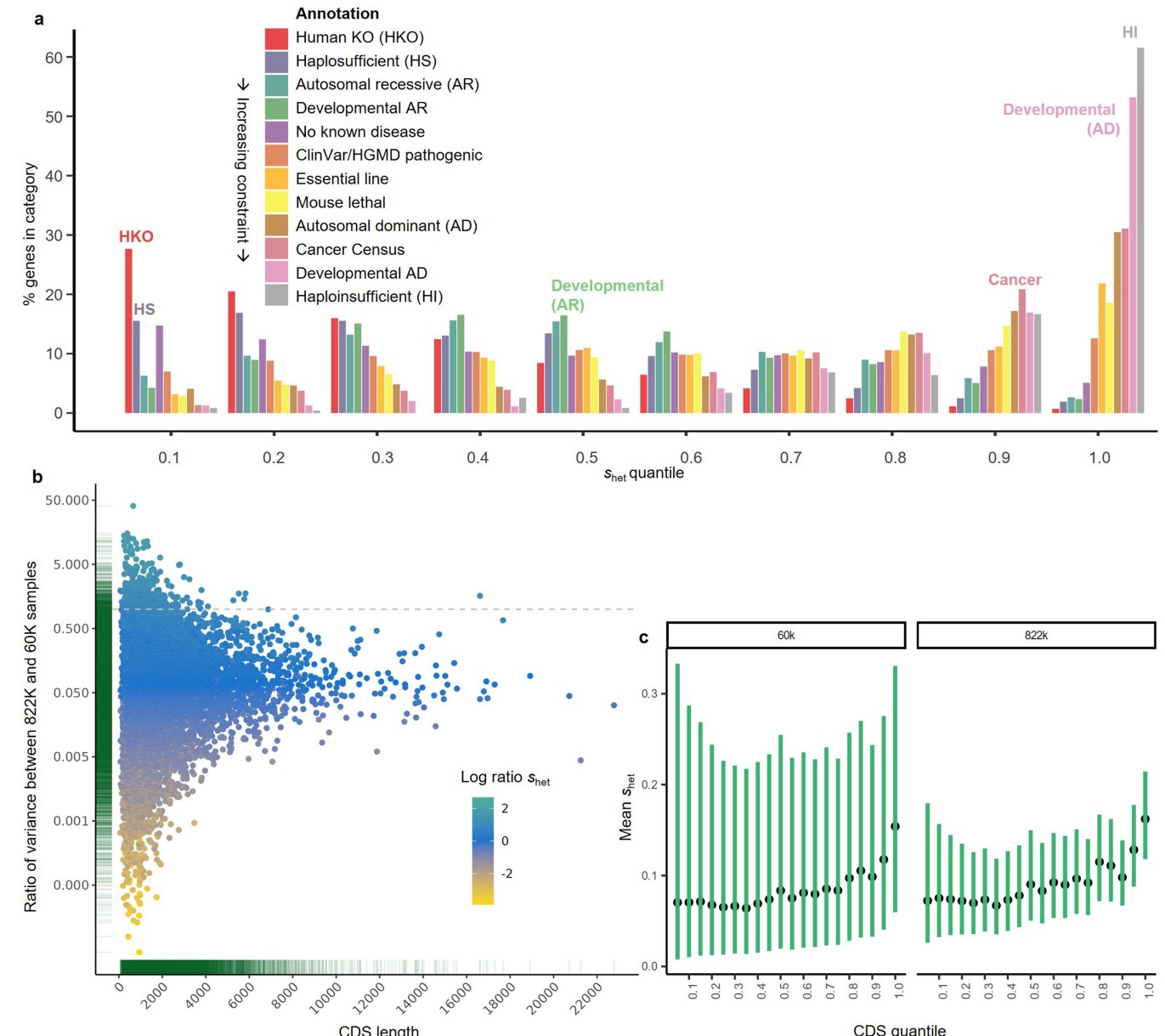

**Extended Data Fig. 3 | $s_{het}$ distribution across gene essentiality and disease categories, and evidence that a larger sample improves the precision of $s_{het}$ estimates. a**, Proportion of genes in each disease/essentiality annotation list (gathered from published literature and databases) that correspond to each $s_{het}$ decile. **b**, Ratio of variances from MCMC sampling for $s_{het}$ calculated on full dataset (824k) and randomly downsampled set of 60,000 individuals. The mean ratio around 0.05 suggests that gene-level variance from the full dataset is 20x smaller than variance for the same gene using the downsampled set. This is the case despite similar, or even higher, $s_{het}$ mean estimates as shown by the colour bar. **c**, Mean (points) and 95% HPD (green bars) from MCMC sampling for $s_{het}$ estimates of genes in each CDS length quantile for downsampled 60k (left) and full 822K (right) samples. HPD and variance in panels **b**,**c** were derived from MCMC sampling for 8,000 final iterations.

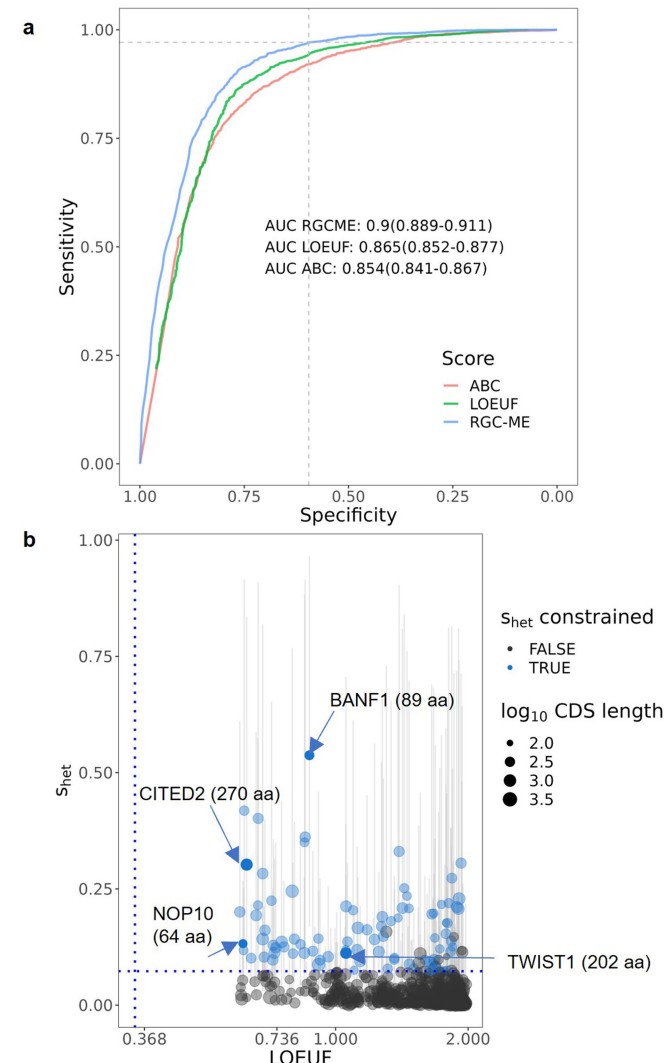

**Extended Data Fig. 4 | Comparisons between $s_{het}$ computed with RGC-ME and other gene constraint metrics. a**, Receiver operator curves showing the discrimination between the "high" and "low" constraint genes using different constraint metrics: $s_{het\text{-}RGCME}$, $s_{het\text{-}ABC}$, and LOEUF for transcripts with values from all three methods. The highly constrained category comprised 1,476 haploinsufficient and autosomal dominant (including developmental-specific) genes and the comparison group was represented by 3,893 haplosufficient and genes with rare, biallelic pLOFs in RGC-ME. Dotted lines indicate specificity and sensitivity for $s_{het} = 0.073$. Sensitivity = TP / (TP+FN), Specificity = TN / (TN+FP). Spearman rank correlations between $s_{het}$ from RGC-ME with these estimates are high (-0.768 with LOEUF, 0.778 with $s_{het\text{-}ABC}$). **b**, $s_{het}$ vs LOEUF results for 973 genes with ≤5 expected LOFs. 5 genes are highlighted for short coding sequence length (CDS), that are constrained according to $s_{het}$ (both mean>0.073 and lower bound>0.021) and unconstrained according to LOEUF. LOEUF is an alternate measure of pLOF-based gene constraint and ranges from 0 to around 2; a value < 0.35 is considered constrained. Error bars (grey lines) around $s_{het}$ denote 95% highest posterior density.

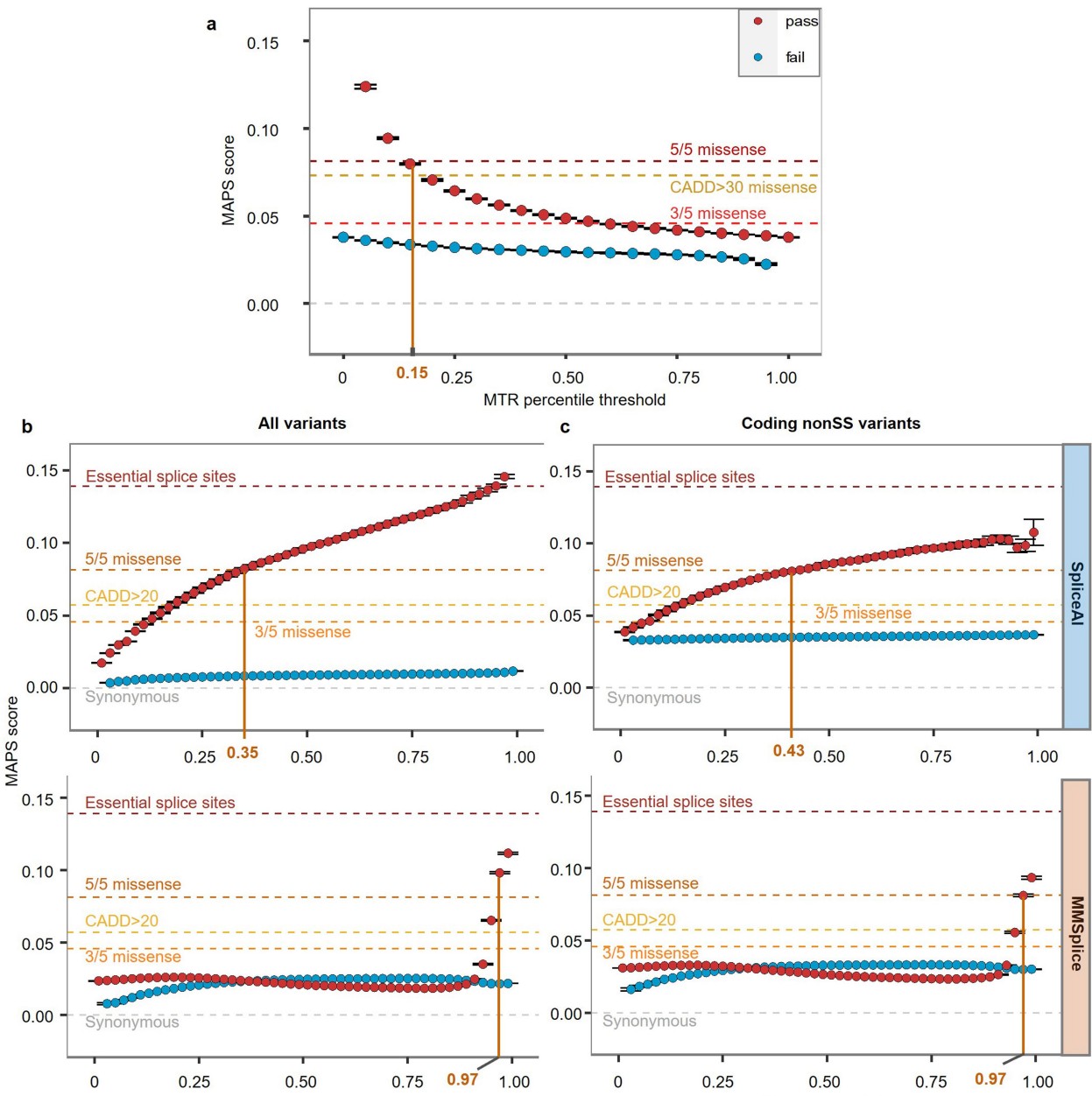

**Extended Data Fig. 5 | Determination of the MAPS score threshold to define constrained regions and SAVs. a–c**, To systematically evaluate the deleteriousness of missense variants at various MTR scores, we compared MAPS scores across MTR (**a**) and splice score (**b,c**) thresholds. **a**, MTR was divided from 5% to 100% with step size of 5% (lower MTR percentile is more constrained). For each MTR percentile threshold, we divided variants into two sets: variants that pass the percentile threshold (red dots), and variants greater than the percentile (blue dots). The y axis represents MAPS score for each set of variants; the x-axis shows the tested MTR percentile threshold. Error bars represent standard deviation around the mean proportion of singletons per bin. A total of 68,636,473 variants were included in this analysis. **b,c**, Schematic of MAPS-derived filters for SpliceAI and MMSplice for all variants (**b**) and coding nonSS variants (**c**), respectively. A list of prediction score thresholds were set up for both SpliceAI and MMSplice, ranging from 0.1 to 0.99, with step size of 0.02. For each threshold, we divided variants into two sets: the set of variants that passed the threshold, represented as red dots, and the set of variants that failed the filter, represented as blue dots. The y axis represents MAPS score for each set of variants; the x-axis shows the tested score threshold. All MAPS scores were calculated based on the set of variants that pass QC metrics and have splice prediction scores in RGC-ME unrelated samples. Error bars represent standard deviation around the mean proportion of singletons per bin. For all variants (**b**), the total number of variants were n = 12,886,467 and 20,604,651 for SpliceAI and MMSplice, respectively. For coding nonSS variants (**c**), n = 5,468,779 and 4,013,820 for SpliceAI and MMSplice, respectively.

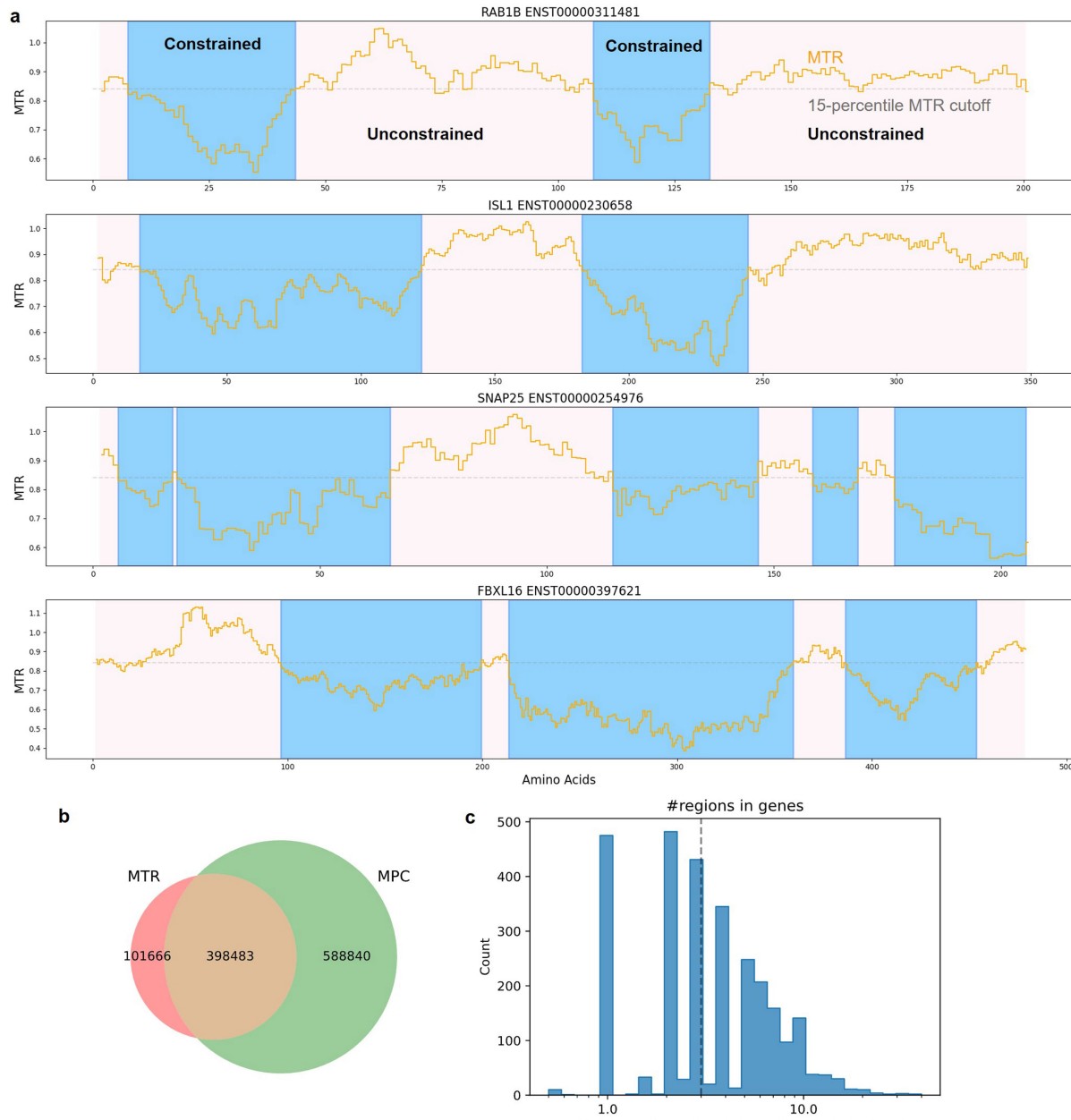

**Extended Data Fig. 6 | MTR-based segmentation and comparison of constrained regions in 2,832 genes that have constrained regions in both MTR and MPC (matched on transcript IDs). a**, Dashed line indicates the top-15-percentile exome-wide MTR threshold used for MTR-based segmentation (0.841). Blue and red regions represent MTR-constrained and unconstrained regions, respectively. **b**, Constrained MTR regions that overlap with MPC segments. 80% of MTR-constrained regions (represented by the combined area of red and yellow) overlap with MPC-constrained segments (yellow), whereas 40% of constrained MPC segments (represented by the combined area of green and yellow) are included in the intersection (yellow). The numbers indicated on the Venn diagram represent number of amino acids. Aside from 2,832 genes that had both MTR- and MPC-constrained regions, an additional 297 genes had only MPC-constrained segments, and 9,514 genes had only MTR-constrained regions (not included in the Venn diagram). **c**, Distribution of the fold change of the number of MTR-constrained regions and constrained MPC segments ($\gamma \leq 0.6$) per gene. Dashed line indicates the median fold change of 3. Data shown is for the 2,832 genes that have constrained region annotations both in MTR and MPC segments.

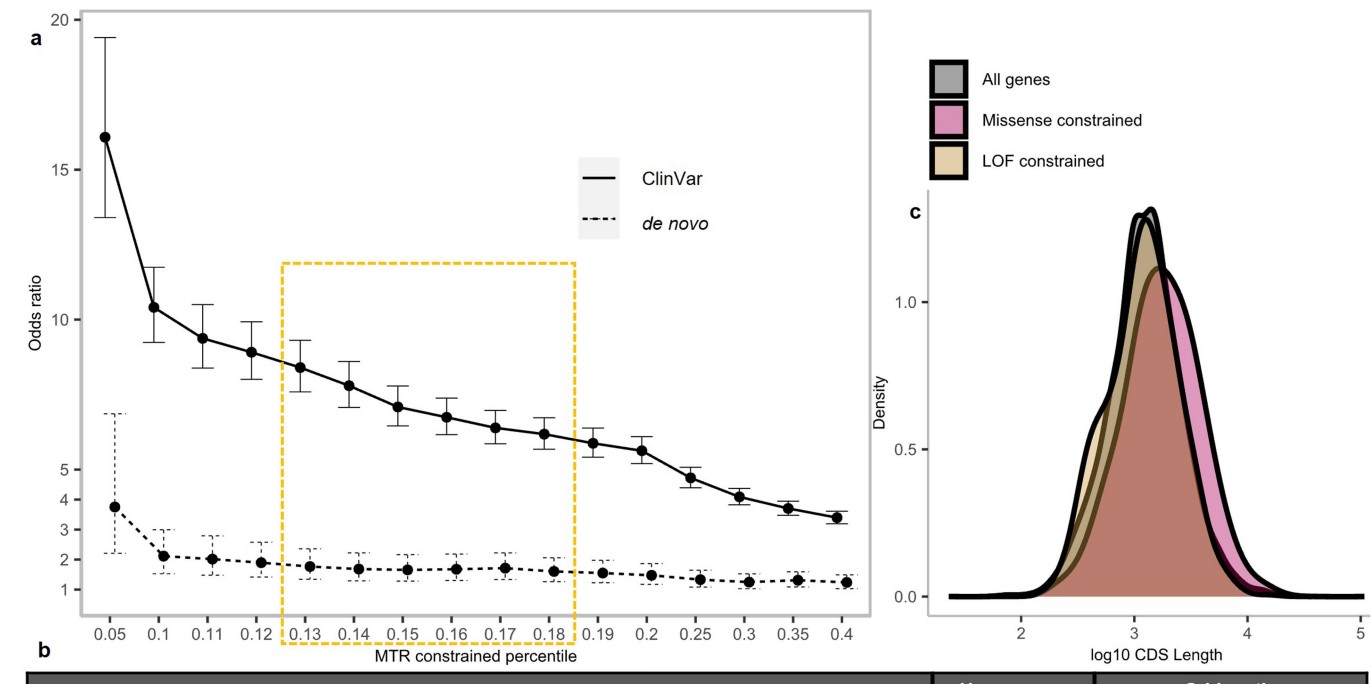

**Extended Data Fig. 7 | Constrained MTR regions are enriched in ClinVar pathogenic variants and shorter than LOF-specific constrained genes.**
**a**, Odds ratios (points) comparing enrichment of pathogenic versus benign ClinVar variants (solid line) and *de novo* variants in cases versus controls (dotted line) in MTR-constrained regions. Error bars represent 95% CIs (Fisher's exact test). The total number of variants in each comparison are shown in **b** for the MTR-percentile cut-offs highlighted in yellow. In total, 5,818 case and 553 control variants were used in the *de novo* analysis and 7,944 pathogenic and 11,993 benign variants were used in the ClinVar analysis. **b**, Table of case and control variants in constrained and unconstrained regions to compute statistical tests for ClinVar ("CV") and *de novo* ("DN") variants across 5 different MTR-percentile thresholds (13-17%, yellow boxed region in **a**). Statistics include

hypergeometric tests (p-value for enrichment of case and control variants in constrained regions) and odds ratios comparing enrichment of case vs control in constrained regions. The background rate of constrained regions among variants in the comparison set represented by "% constrained background". **c**, CDS length comparison between 1,482 missense-specific constrained genes (defined where >15% of gene is in the top 15 percentile of MTR, based on one-sided binomial tests with $\pi_0 = 0.15$, p < 0.05, Bonferroni corrected) and 1,424 LOF-specific constrained genes with $s_{het}$ score <0.073. Log10 CDS length for all 19,644 genes (canonical transcripts) shown in the grey curve. The missense-specific constrained genes had significantly shorter CDS length than LOF-specific constrained genes (p = $2.9 \times 10^{-40}$, two-sided Wilcoxon test).

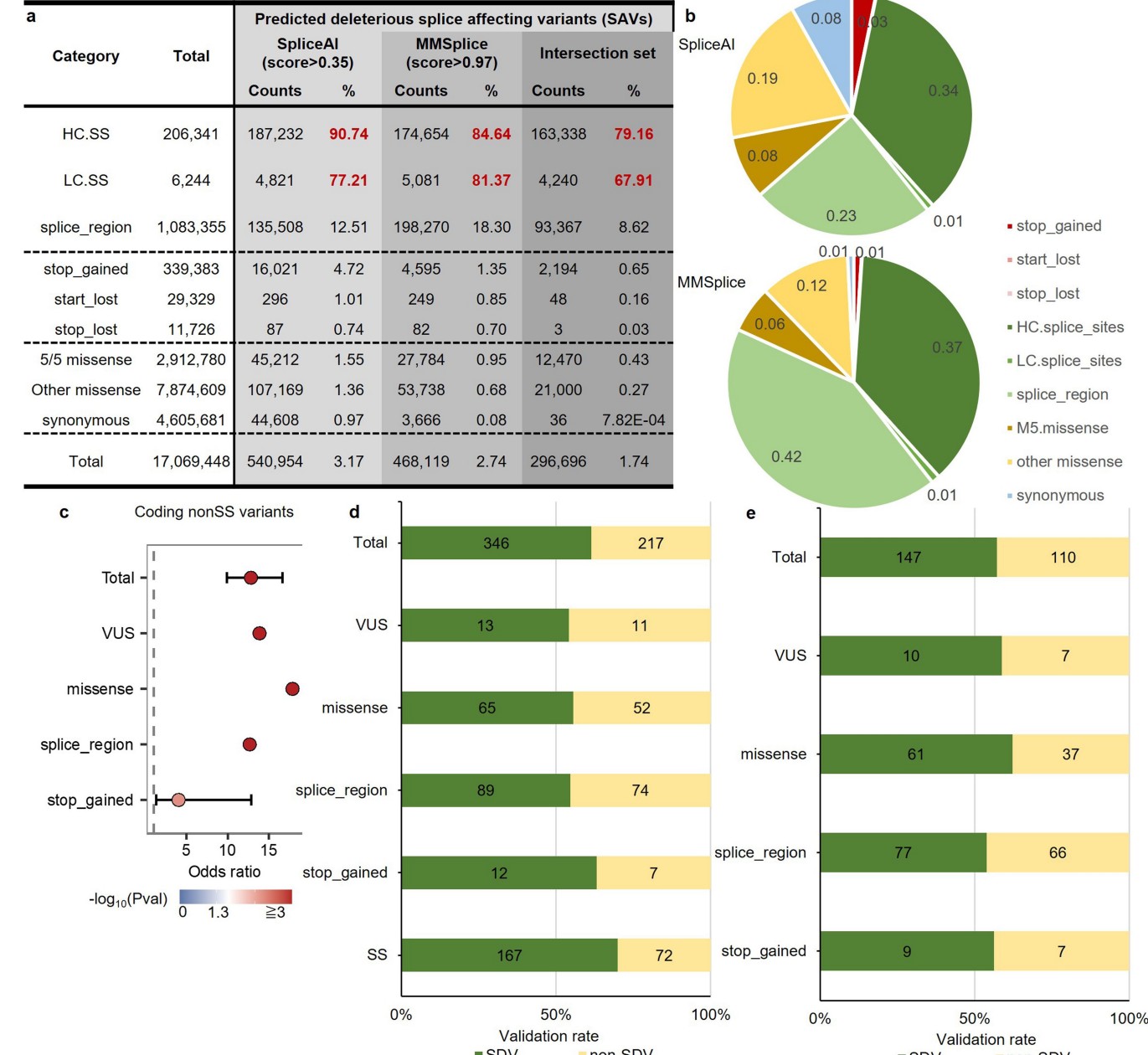

**Extended Data Fig. 8 | Characteristics of predicted deleterious SAVs.**
**a**, Summary of unique variant-transcript pairs of predicted deleterious SAVs in different functional categories using MAPS-defined prediction score thresholds for SpliceAI and MMSplice. Percents are computed out of total variants in each effect class. **b**, Distribution of different functional categories of predicted deleterious SAVs. (HC: High confidence, LC: Low confidence. These annotation tags are derived from LOFTEE). **c**, Empirical validation of MAPS-predicted deleterious coding nonSS SAVs: enrichment of predicted coding nonSS deleterious SAVs in experimentally validated SDVs compared to non-SDVs. Odds ratios (points) were derived using two-sided Fisher's exact test and error bars show 95% CIs. A total of n = 17,395 variants, of which 147 SAVs were validated SDVs, were included. **d,e**, Fraction of predicted deleterious SAVs (**d**) and coding nonSS SAVs (**e**) that were validated as SDVs by any of the three splice reporter assays.

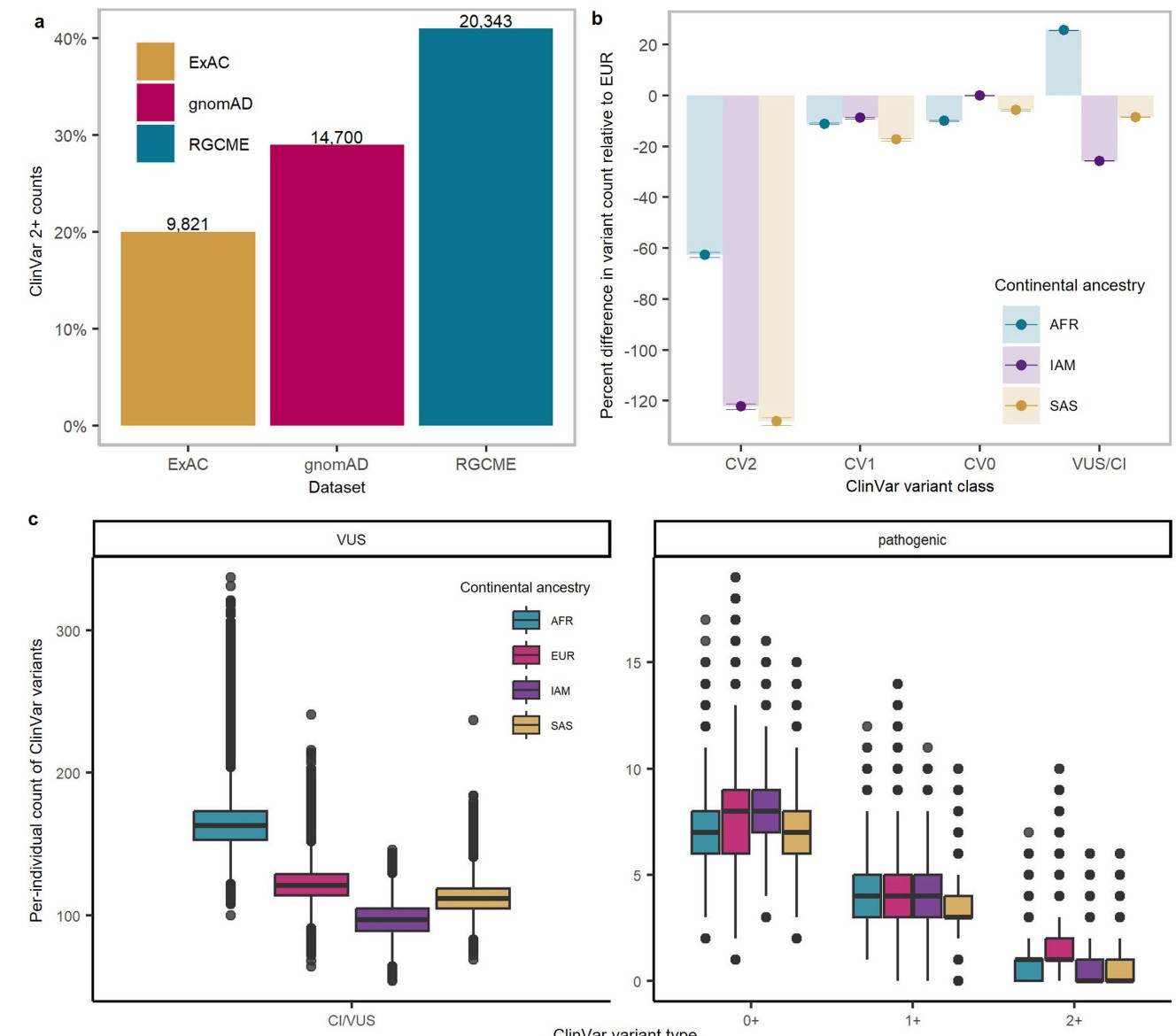

**Extended Data Fig. 9 | ClinVar variant counts. a**, Counts of autosomal pathogenic ClinVar high-confidence variants (two stars or more) observed in large-scale exome sequencing studies, including RGC-ME, gnomAD (exomes, v2.1.1), and ExAC, are indicated on the top of each bar. The left axis indicates the total coverage of ClinVar pathogenic variants represented in each dataset. **b**, Bars and points both depict the mean per cent difference of per-individual counts in ClinVar categories (pathogenic 0+, 1+, 2+, VUS+CI) across continental ancestries using all unrelated samples, with respect to EUR (e.g. [count$_{AFR}$ − count$_{EUR}$]/count$_{AFR}$ × 100). CV0, 1, and 2 refer to ClinVar pathogenic star rating 0+, 1+, and 2+ categories, respectively. VUS/CI combines variants annotated as "variants of unknown significance" and "conflicting interpretations". All per-individual counts in non-EUR were significantly different compared to counts in EUR (e.g. per-individual counts of ClinVar 2+ variants in AFR were

0.576 [0.567, 0.585] lower than those in EUR) except for ClinVar 0+ counts in IAM compared with EUR (t-tests with Bonferroni correction). Error bars show 95% CI of mean per cent difference from t-test. **c**, Per-individual count of VUSs and conflicting information (CI, left); and pathogenic variants (right) in RGC-ME for all unrelated samples. The lower bound, centre and upper bound of each box plot represents the 25, 50, and 75 percentiles of the distributions of counts. Points represent outliers, and whisker minima and maxima represent the smallest and largest points 1.5-times beyond the interquartile range. For **b**,**c**, a total of 749,584 unrelated individuals were included across 4 ancestries and individuals were assigned discrete ancestries based on overall FSA probabilities >50%; see Supplementary Table 1a for sample size breakdowns. The x-axis indicates ClinVar star rating which is a measure of the confidence of the annotation (pathogenic/benign).

**Extended Data Table 1 | Sample breakdown of individuals with rare biallelic pLOF variants by ancestry**

| | Total samples | Homozygous pLOFs # of genes with | | | Compound heterozygous pLOFs # of genes with | | | All biallelic pLOFs # of genes with | | |
|---|---|---|---|---|---|---|---|---|---|---|
| | | ≥1 carrier | ≥5 carriers | ≥10 carriers | ≥1 carrier | ≥5 carriers | ≥10 carriers | ≥1 carrier | ≥5 carriers | ≥10 carriers |
| AFR | 55281 | 1059 | 390 | 285 | 543 | 175 | 107 | 1258 | 631 | 513 |
| EAS | 5849 | 308 | 56 | 35 | 73 | 6 | 3 | 340 | 95 | 70 |
| EUR | 722477 | 2483 | 673 | 396 | 966 | 310 | 149 | 2729 | 1052 | 748 |
| IAM | 85125 | 1243 | 154 | 79 | 306 | 36 | 15 | 1367 | 308 | 221 |
| SAS* | 30752 | 2035 | 252 | 106 | 220 | 26 | 9 | 2071 | 307 | 160 |
| ALL | 983578 | 4686 | 1406 | 877 | 1205 | 522 | 294 | 4848 | 1632 | 1042 |

Counts of genes with rare homozygous and rare compound heterozygous pLOF variants (includes variants with AAF < 0.01). 'Total samples' denotes the number of individuals per ancestry in the related dataset (983,578 individuals) based on continental ancestry with a probability assignment greater than 50%.
*Includes cohorts with high rates of consanguinity.

| | |
|---|---|

# Reporting Summary

## Statistics

For all statistical analyses, confirm that the following items are present in the figure legend, table legend, main text, or Methods section.

| n/a | Confirmed | |
|---|---|---|
| ☐ | ☒ | The exact sample size (*n*) for each experimental group/condition, given as a discrete number and unit of measurement |
| ☐ | ☒ | A statement on whether measurements were taken from distinct samples or whether the same sample was measured repeatedly |
| ☐ | ☒ | The statistical test(s) used AND whether they are one- or two-sided<br>*Only common tests should be described solely by name; describe more complex techniques in the Methods section.* |
| ☐ | ☒ | A description of all covariates tested |
| ☐ | ☒ | A description of any assumptions or corrections, such as tests of normality and adjustment for multiple comparisons |
| ☐ | ☒ | A full description of the statistical parameters including central tendency (e.g. means) or other basic estimates (e.g. regression coefficient) AND variation (e.g. standard deviation) or associated estimates of uncertainty (e.g. confidence intervals) |
| ☐ | ☒ | For null hypothesis testing, the test statistic (e.g. $F$, $t$, $r$) with confidence intervals, effect sizes, degrees of freedom and $P$ value noted<br>*Give P values as exact values whenever suitable.* |
| ☐ | ☒ | For Bayesian analysis, information on the choice of priors and Markov chain Monte Carlo settings |
| ☐ | ☒ | For hierarchical and complex designs, identification of the appropriate level for tests and full reporting of outcomes |
| ☐ | ☒ | Estimates of effect sizes (e.g. Cohen's *d*, Pearson's *r*), indicating how they were calculated |

*Our web collection on statistics for biologists contains articles on many of the points above.*

## Software and code

Policy information about availability of computer code

| | |
|---|---|
| Data collection | No software was used for data collection. |
| Data analysis | Publicly available software and packages used in this study are described in the Supplementary Information. In summary, sequencing reads were generated using bcl2fastq v2.20 and were mapped to references using BWA MEM v0.7.17. Variants were identified using DeepVariant v0.10, aggregated with GLnexus v1.4.3, and converted to bed/bim/fam format using PLINK 1.9. Variants were annotated with VEP (Ensembl, v100.4) and pLOF variants were further annotated with the VEP LOFTEE plug-in. Array variants were phased using Eagle v2.4 and imputed using MINIMAC4. PLINK2 was used for principal components analysis and to compute Fst. The csq function in BCFtools v1.18 was used to annotate inframe indels resulting from a combination of frameshift indels on the same haplotype and bedtools v2.30.0 was used to determine variant genetic context and neighboring nucleotides. Picard LiftoverVcf v3.0.0 was used to transform sequence coordinates to GRCh38. Relatedness was determined with PRIMUS: https://primus.gs.washington.edu/primusweb/res/documentation.html. We adapted scripts from https://github.com/pjshort/dddMAPS to compute updated MAPS metrics. For identifying compound heterozygous variants, exome variants were merged with a well imputed common variant backbone and phased using SHAPEIT5 (https://github.com/odelaneau/shapeit5). Large-scale data manipulation used Scala 2.12 on a 10.4 LTS runtime (Apache Spark 3.2.1) with standard Spark functions. Beyond standard R packages, visualization tools, and data processing libraries (e.g. dplyr, ggplot2, data.table), we used Rstan (v2.33) to build Bayesian hierarchical models for calculating heterozygous selection coefficient, rmutil (v4.1.2) to project LOF accrual,and boot (v4.1.1) for bootstrapping. Python code used standard packages (e.g. scipy, numpy, pandas) for analysis and sqlalchemy (v2.0.23) to store and query tables. Custom code to generate LOF projection curves is available at https://github.com/rgcgithub/rgc_me_analysis. |

For manuscripts utilizing custom algorithms or software that are central to the research but not yet described in published literature, software must be made available to editors and reviewers. We strongly encourage code deposition in a community repository (e.g. GitHub). See the Nature Portfolio guidelines for submitting code & software for further information.

# Data

All manuscripts must include a data availability statement. This statement should provide the following information, where applicable:

- Accession codes, unique identifiers, or web links for publicly available datasets
- A description of any restrictions on data availability
- For clinical datasets or third party data, please ensure that the statement adheres to our policy

Genetic variation data for 821,979 unrelated individuals are made publicly available through the RGC Million Exome Browser (https://rgc-research.regeneron.com/me/home). Features include genomic locations, alleles, fine-scale ancestry assignments, population-specific allele frequencies, and functional annotations for the genetic variants. In addition, the public browser allows academic researchers to download "vcf" files. Exome-wide MTR scores are available for download from Figshare: https://doi.org/10.6084/m9.figshare.24587328.v2. Individual-level sequence data deposited with the UK Biobank are freely available to approved researchers. The human reference genome GRCh38 can be obtained from ftp://ftp-trace.ncbi.nlm.nih.gov/1000genomes/ftp/technical/reference/GRCh38_reference_genome/GRCh38_full_analysis_set_plus_decoy_hla.fa. Instructions for access to UK Biobank data are available at https://www.ukbiobank.ac.uk/enable-yourresearch. Information about data access policy for researchers interested in the MCPS data can be found at https://www.ctsu.ox.ac.uk/researc/prospective-blood-based-study-of-150-000-individuals-in-mexico. Geisinger Health System individual-level data are available to qualified academic, non-commercial researchers through an information transfer agreement by contacting Lance Adams (ljadams1@geisinger.edu). Information about the data access policy, procedures, and contact details for the cohorts included in this dataset can be obtained via the URLs given in the RGC ME browser at https://rgc-research.regeneron.com/me/data-contributors. This information is also provided in the Supplementary Table 1c with relevant references, if available.

# Research involving human participants, their data, or biological material

| | |
|---|---|
| Reporting on sex and gender | *Use the terms sex (biological attribute) and gender (shaped by social and cultural circumstances) carefully in order to avoid confusing both terms. Indicate if findings apply to only one sex or gender; describe whether sex and gender were considered in study design; whether sex and/or gender was determined based on self-reporting or assigned and methods used.*<br>*Provide in the source data disaggregated sex and gender data, where this information has been collected, and if consent has been obtained for sharing of individual-level data; provide overall numbers in this Reporting Summary. Please state if this information has not been collected.*<br>*Report sex- and gender-based analyses where performed, justify reasons for lack of sex- and gender-based analysis.* |
| Reporting on race, ethnicity, or other socially relevant groupings | *Please specify the socially constructed or socially relevant categorization variable(s) used in your manuscript and explain why they were used. Please note that such variables should not be used as proxies for other socially constructed/relevant variables (for example, race or ethnicity should not be used as a proxy for socioeconomic status).*<br>*Provide clear definitions of the relevant terms used, how they were provided (by the participants/respondents, the researchers, or third parties), and the method(s) used to classify people into the different categories (e.g. self-report, census or administrative data, social media data, etc.)*<br>*Please provide details about how you controlled for confounding variables in your analyses.* |
| Population characteristics | The RGC Million Exome dataset comprises harmonized whole-exome sequencing data from 983,578 individuals sequenced by the Regeneron Genetics Center (RGC).This study aggregates data from multiple cohorts, compiled retrospectively without any selection criteria based on age, gender, or genotypic information. The complete RGC-ME dataset (N=983,578) is made up of 57.55% females. Age data was available for 874,175 individuals, with a median age of 57 [47.0 - 64.7, Q1-Q3]. The data includes participants from biobanks: UK BioBank, Geisinger Health System, The Mexico City Prospective Cohort, Penn Medicine BioBank, BioMe BioBank, Dallas Heart Study, Amish Research Clinic, Center for Non-Communicable Diseases, Australian New Zealand MS Genetics Consortium and a variety of case control studies for complex diseases such as psoriasis, rheumatoid arthritis, diabetes. The source of all data along with links to the projects (where available) are listed in the RGC-ME web portal (https://rgc-research.regeneron.com/me/data-contributors). To ensure this sample set characterizes genetic variation representative of the general population, we excluded available samples from cohorts specifically enrolling participants with Mendelian diseases, neurodevelopmental disorders and blood cancers. |
| Recruitment | The data contributors for RGC-ME are listed in a table in the Data Availability section of the manuscript. The recruitment strategy of participants can be found in the corresponding URLs and/or reference publications. We did not specifically recruit subjects for this manuscript as this is a retrospective analysis. |
| Ethics oversight | Ethical approval for the UK Biobank was previously obtained from the North West Centre for Research Ethics Committee (11/NW/0382). The work described herein was approved by UK Biobank under application number 26041. Approval for Geisinger Health System MyCode analyses was provided by the Geisinger Health System Institutional Review Board under project number 2006-0258. Informed consent was obtained for all study participants. Appropriate consent for the The Penn Medicine BioBank was obtained from each participant regarding storage of biological specimens, genetic sequencing and genotyping, and access to all available EHR data. This study was approved by the Institutional Review Board of the University of Pennsylvania and complied with the principles set out in the Declaration of Helsinki. All subjects participating in the MAYO-RGC Project Generation provided informed consent for use of specimens and data in genetic and health research and ethical approval for Project Generation was provided by the Mayo Clinic IRB (#09-007763). All research performed in this study uses de-identified data (without any Protected Health Information data) with no possibility of re-identifying any of the participants. Approval for Indiana Biobank was provided by Indiana University Institutional Review Board under project number 1105005445. For MCPS study paricipants, approval for the study was given by the Mexican Ministry of Health, the Mexican National Council of Science and Technology (0595 P-M) and the Central Oxford Research Ethics Committee (C99.260) and the Ethics and Research commissions from the Medicine Faculty at the National Autonomous University of |

Mexico (UNAM) (FMED/CI/SPLR/067/2015). All study participants provided written informed consent. Study participants were recruited from the BioMe Biobank Program of The Charles Bronfman Institute for Personalized Medicine at Mount Sinai Medical Center from 2007 onward. The BioMe Biobank Program (Institutional Review Board 07-0529) operates under a Mount Sinai Institutional Review Board-approved research protocol. All study participants provided written informed consent. Informed consent was obtained for all study participants.

Note that full information on the approval of the study protocol must also be provided in the manuscript.

# Field-specific reporting

Please select the one below that is the best fit for your research. If you are not sure, read the appropriate sections before making your selection.

☒ Life sciences        ☐ Behavioural & social sciences        ☐ Ecological, evolutionary & environmental sciences

For a reference copy of the document with all sections, see nature.com/documents/nr-reporting-summary-flat.pdf

# Life sciences study design

All studies must disclose on these points even when the disclosure is negative.

| | |
|---|---|
| Sample size | Sample size was not predetermined. Most analyses were restricted to unrelated samples that passed quality control metrics. Supplementary Table 1a shows the details of analyses subsets and the sample numbers included in those analysis. For each analysis, we used the maximum number of samples available in that subset. These sample sizes are larger than earlier published reports and the analyses in the paper show improved estimation of constraint metrics (see Extended Fig. 3c). |
| Data exclusions | Variant level QC was performed as described in Supplementary Information section titled "Quality control (QC) of dataset". Variants flagged by the SVM prediction as "low quality" were excluded from the analyses. |
| Replication | This paper describes genetic variation observed in about a million individuals and describes the properties and insights from the data. There is no other dataset of comparable size. Nonetheless, we show that gene constraint metric, Shet, derived from RGC-ME correlates well with other published reports (Extended Fig 4, main manuscript line numbers 148 - 153) |
| Randomization | Randomization was not required for this study as this is a population-based study and not a case-control study. |
| Blinding | Blinding was not required for the analyses completed in this study as this is a population-based study and not a case-control study. |

# Reporting for specific materials, systems and methods

We require information from authors about some types of materials, experimental systems and methods used in many studies. Here, indicate whether each material, system or method listed is relevant to your study. If you are not sure if a list item applies to your research, read the appropriate section before selecting a response.

### Materials & experimental systems

| n/a | Involved in the study |
|---|---|
| ☒ | ☐ Antibodies |
| ☒ | ☐ Eukaryotic cell lines |
| ☒ | ☐ Palaeontology and archaeology |
| ☒ | ☐ Animals and other organisms |
| ☒ | ☐ Clinical data |
| ☒ | ☐ Dual use research of concern |
| ☒ | ☐ Plants |

### Methods

| n/a | Involved in the study |
|---|---|
| ☒ | ☐ ChIP-seq |
| ☒ | ☐ Flow cytometry |
| ☒ | ☐ MRI-based neuroimaging |

# Plants

| | |
|---|---|
| Seed stocks | *Report on the source of all seed stocks or other plant material used. If applicable, state the seed stock centre and catalogue number. If plant specimens were collected from the field, describe the collection location, date and sampling procedures.* |
| Novel plant genotypes | *Describe the methods by which all novel plant genotypes were produced. This includes those generated by transgenic approaches, gene editing, chemical/radiation-based mutagenesis and hybridization. For transgenic lines, describe the transformation method, the number of independent lines analyzed and the generation upon which experiments were performed. For gene-edited lines, describe the editor used, the endogenous sequence targeted for editing, the targeting guide RNA sequence (if applicable) and how the editor was applied.* |
| Authentication | *Describe any authentication procedures for each seed stock used or novel genotype generated. Describe any experiments used to assess the effect of a mutation and, where applicable, how potential secondary effects (e.g. second site T-DNA insertions, mosiacism, off-target gene editing) were examined.* |

