## [Peer Review File · Nature]

Manuscript Title: A deep catalog of protein-coding variation in 983,578 individuals

Reviewer Comments & Author Rebuttals

Reviewer Reports on the Initial Version:

Referees' comments:

Referee #1 (Remarks to the Author):

In this manuscript, the authors present an exome sequencing dataset of nearly a million individuals, representing a large increase in sample size over similar studies in the past. This data is used to study selective constraint against various classes of coding variation, which requires a large sample size because it involves estimating the depletion of an already-rare event. In particular, RGC-ME is 7 times larger than gnomAD. It is only ~2 times larger than the previously published UK Biobank exome sequencing dataset, which is a subset of RGC-ME, but to my knowledge there are no published estimates of constraint based on UK Biobank exomes. The significant increase in sample size, together with the central importance of selective constraint in human genetics research, makes it immediately interesting to researchers. However, I have major concerns about the manuscript as written.

Major comments

1. Estimates of selective constraint against heterozygous loss of function (s_{het}) are compared against estimates from 2017 (Cassa et al., N=61k). These should be updated to reflect the existing state of the art. First, they should be compared against LOEFF estimates from gnomAD, which involved a >2x larger sample size. Although the gnomAD paper did not quantify s_{het} (strangely), it is still possible to compare the rank-ordering of genes. What is the Spearman correlation? Are there any genes that are estimated to be constrained in gnomAD but not in GCM-ME? Second, they should be compared with the s_{het} estimates of Agarwal et al. 2023 eLife, which utilized the 55k non-Finnish EUR samples in gnomAD and fit a more realistic demographic model as compared with Cassa et al. Optionally, they could also be compared against the non-peer-reviewed s_{het} estimates from Zeng*, Spence* et al. bioRxiv (<https://doi.org/10.1101/2023.05.19.541520>), which are based on 125k gnomAD samples and a sophisticated new estimation method.
2. Estimates of missense regional constraint also need to be compared with existing estimates from refs. 32-33. Does the increase in sample size lead to improved sequence-length resolution?
3. Line 281 states that pKO genes in which a homozygous knockout is observed are “less constrained than expected,” but this statement is apparently based upon a strawman “expected” under which the number of homozygotes is independent of the number of heterozygotes. Instead, the expected fraction of homozygotes for an allele with frequency p is p^2 , and pLoFs in constrained genes have smaller p . Are pKOs significantly less constrained than expected *after accounting for the number of heterozygotes*? Exome-wide, is there a depletion of homozygous pKOs vs. expected (i.e., deviation from HWE after accounting for autozygosity)? If so, are the gene families discussed in lines 287-300

significantly less depleted for homozygotes?

4. In the section on differential allele frequency, it is heavily implied that the signals of high F_{st} are due to population specific selection. This is evidenced in line 313-314, 322-323, 329-331, and 345-346. Population-specific selection is a sensitive topic, and I am concerned that with its vagueness, this section is open for misinterpretation. The question is whether population specific selection on coding variation is widespread or rare; as currently written, the manuscript vaguely suggests that it is widespread. However, Supplementary Figure 8 shows that the fraction of variants with $F_{st} > 0.05$ (an *extremely* lenient threshold) is ~identical for missense vs. synonymous variants, indicating that most variants at this threshold are not under population specific selection. This is expected because genome-wide F_{st} is greater than 0.05 for all of these ancestry-group pairs. Similarly, in Supplementary Figure 7, the dotted line at 0.15 suggests that variants above that line have some special meaning (because typically, a line on a Manhattan plot is a stringent significance threshold). But again, genome-wide average F_{st} is routinely >0.1 between continental groups, so a variant-level threshold of 0.15 is very lenient.

On lines 329-336, a number of anecdotes are presented about the phenotypic effects of some of these variants with $F_{st} > 0.05$. The suggestion is that these phenotypic associations are responsible for the observed difference in allele frequency. Considering the large number of synonymous variants with $F_{st} > 0.05$ (and the large number of noncoding variants, if one were to look in a WGS dataset), this is unlikely. On lines 336-339, it is stated that variants with $F_{st} > 0.05$ are enriched for significant GWAS loci. However, the odds ratios reported are too large to be driven by selection, considering the lenient threshold. Does this analysis account for allele frequency in the discovery population? If not, perhaps the correct interpretation is that variants at high frequency in EUR are more likely to have $F_{st} > 0.05$ (Supplementary Figure 8) and also more likely to be in the GWAS catalog (because GWAS loci are common).

I think this section would still be of interest if it were purely descriptive – how do allele frequencies differ across ancestries? – rather than speculative. If the authors wish to retain content about population-specific selection, I think the claims should be explicit, and they should be backed by much stronger evidence. This would likely entail forward simulations under a null model with a realistic demography and non-population-specific negative selection.

5. The last Results subsection identifies thresholds for spliceAI and MMSplice scores such that splice-altering variants above that threshold have allele frequency spectra similar to missense deleterious variants. The analysis is motivated by the statement that existing tools “generally lack clear guidance for their use in detecting deleterious splice affecting variants.” However, the original spliceAI paper (Jaganathan et al., ref 58) contains the following text:

At common allele frequencies, high-scoring predicted cryptic splice variants (score > 0.8) are under strong negative selection, as evidenced by their relative depletion compared to expectation (Figure 4A). At this threshold, where most variants are expected to be close to fully penetrant in the RNA-seq data (Figure 2D), predicted synonymous and intronic cryptic splice mutations are depleted by 78% at common allele frequencies, which is comparable with the 82% depletion of frameshift, stop gain, and

essential GT or AG splice-disrupting variants.

Their analysis is extremely similar to that presented in this subsection. The only important difference is that Jaganathan et al. chose to calibrate vs. loss-of-function as opposed to missense deleterious variants, such that they end up choosing a more stringent threshold. I think this section should be re-framed to acknowledge this.

Minor comments

6. Lines 87-88: although the absolute number of non-EUR individuals is greater than existing studies, the fraction of non-EUR individuals remains low (and is much lower than TOPMed). I suggest tempering the statement that "RGC-ME has more population diversity than other large scale genetic variation datasets." Perhaps it would be more accurate to state that the dataset represents a large increase in sample size, including for non-European ancestries.

7. Lines 99-100: I think the distinction here between variants and sites could be clarified.

8. 217: I think you mean, "To identify genes depleted of missense variation"

9. 217-232: the approach used in this paragraph needs some justification. What is the rationale for looking just at enrichment in the top 1% of MTR, and throwing out variation below that threshold?

10. 268-270: My understanding is that pKOs are defined as genes with homozygous pLoF alleles, and that compound heterozygotes are excluded. If this is correct, is this choice made because the data are unphased? If so, what are the prospects for phasing RGC-ME? (This is significant because presumably, there are many more compound hets than homozygotes in these data).

11. 299-300: this hypothesis makes sense, but is it actually the case that these genes and gene families are paralog-enriched? (Note that gene duplication is thought to result predominately in functional redundancy/dosage sharing, not neofunctionalization; Lan and Pritchard 2016 Science).

12. On the web portal, I suggest that it should display gene-level and region-level constraint metrics in addition to variant allele frequencies (similar to the gnomAD browser).

Signed,
Luke O'Connor

Referee #2 (Remarks to the Author):

This manuscript presents a new publicly available resource of whole-exome DNA sequencing data from nearly 1 million individuals. Descriptive analyses of these data include derivation of estimates of loss-of-function and regional missense constraint, and analysis of homozygous knockouts and splice site variant effects using existing methods. The larger size of the data set compared to previous similar studies allows for the identification of more genes that evolve under purifying selection. The data set will be of great use to researchers working with human population variation data. However, there are methodological concerns regarding the execution and description of the analyses.

Comments

1. Not much information is given about how the data was collected. Is there overlap with existing datasets (UKB, gnomAD, 1KGP)?
2. When comparing high and low constraint groups derived from disease genes (l. 155), the numbers given in the text are not meaningful because it is not stated how many genes are in each group. They should be replaced with a precision-recall (PR) or receiver operating characteristic (ROC) curve. The final list of genes used and their classification ("high constraint" vs. "low constraint") should be provided in a supplementary table.
3. How does s_{het} correlate with the GERP score? How does it compare with other scores based on conservation (phastCons, phyloP)?
4. Are the differences in estimated constraint between ExAC and RGC-ME, where seven genes were constrained only in the smaller ExAC cohort, due to sub-population composition differences between the two datasets? What is the direct comparison of s_{het} estimates between RGC-ME and ExAC?
5. MTR (PMID: 28864458) assumes that all mutation events (synonymous, missense, nonsense) are equally likely. However, mutation rates are far from equal across nucleotide contexts and genomic regions. To account for nucleotide context dependence, the authors use mutation rate estimates from gnomAD for their s_{het} and MAPS analyses, but they do not do so for the MTR analysis. Therefore, the reliability of the MTR results is questionable. This is also consistent with the lack of any correlation between MTR and the conservation metric GERP, as other studies have found that human-specific and phylogenetic conservation scores are (weakly) correlated. Is there a correlation between MTR and being a 5/5 missense variant (cyan dots in Fig. 3C)? Methodologically, why do the authors not show an ROC/PR curve in Fig. 3A?
6. To address missense constraint, the authors could use measures presented in previous work, e.g. the missense Z-score from PMID: 25086666, which is acknowledged to be a gene-level estimate, but preferable to MTR because of the imprecision of the mutation rate model underlying MTR.
7. The reported biological processes enriched among the 849 genes with putative human-specific missense constraint ("neuronal and immune system", l. 213) do not closely reflect the actual list of results (Supp. Fig. 5A). While "neuronal system" indeed has the lowest p-value, the most significant enrichments follow for processes involved in gene expression, development, splicing and basic cellular signaling, for which the argument of species-specific selection cannot be so easily made, casting further doubt on the MTR estimates.
8. When analyzing the fixation index, was it calculated pairwise between all subpopulations or were the subpopulations partially pooled? What exactly is shown in Supp. Fig. 7? What type of selection acts on the annotated genes? Is F_{ST} given only for LoF variants, as the plot title suggests?
9. Which variants are included in the last F_{ST} analysis, of which only a few select results are

mentioned (l. 336)? Why is there no table with all enrichments?

10. Contrary to what is stated (l. 326), differentiated alleles in AFR and EAS sub-populations shown in Supp. Table 7.3 and 7.4 do not contain morphological traits associated with skin/hair pigmentation. Bilirubin, a pigment that is produced during the breakdown of hemoglobin, is not a specific skin or hair pigment.

11. Conceptually, I fail to see the added benefit of combining the MAPS score of a particular variant with its splicing score. Ultimately, this just amounts to thresholding the splicing score and any associations with, for example, ClinVar pathogenicity only show that the splicing score works well. Conversely, if also CADD scores were thresholded (e.g., at 30), the difference in enrichment with ClinVar pathogenicity between thresholded CADD and SAVs would likely be minimal.

12. While the similarity between LOFTEE HC splice sites and SAV splice sites in Fig. 5B suggests that both assignments correlate similarly with pathogenicity, the nonsignificant odds ratio of enrichment with experimentally validated deleterious splice site variants (l. 397, Fig. 5C) suggests that both measures may not capture true effects. What are the odds ratios when comparing SAVs with pathogenic vs. benign ClinVar variants (same as used in Fig. 3A), rather than comparing SAVs with other deleteriousness filters? What would Fig. 5C look like for the variants annotated as "splice_site" in A?

13. Germline mutation rates are known to vary not only between trinucleotide contexts, but also between genes, the most important factor for the latter being replication time. The downloaded gnomAD rates for trinucleotide contexts and methylation levels are not adjusted for this. In line with this, the negative MAPS scores in 5'UTR and intronic regions in Fig. 5 suggest that the mutational model is inaccurate. In particular, since 5'UTRs are enriched in CpG islands, this may suggest that the methylation correction of mutation rates is not working well. More sophisticated models of germline mutability have been in use for some time (e.g. PMID: 30218074, <https://doi.org/10.1101/2022.08.20.504670>, <https://doi.org/10.1101/2022.03.20.485034>), so the authors should make sure to revise their mutational model to account for the effects of replication time and other covariates of mutation rate that are not accounted for in their current model.

14. It is suggested that over 10k VUS are deleterious cryptic splice variants. Could the authors perform an analysis to support this claim? For example, are they enriched in experimentally validated SDVs compared to non-SDVs (compare Fig. 5C)?

Overall, the manuscript does not appear to have been written with great care, and I think it would benefit from careful revision. The following is a non-exhaustive list of inconsistencies, inaccuracies and missing information:

- gnomAD version numbers swapped (l. 86)?
- Two different percentages given for the fraction of mutated synonymous and stop-gained sites (l. 95 and 99).
- Supp. Fig. 1: "CpG" used to denote "CpG transitions" (same for "non-CpG")

- Supp. Fig. 1: "highly methylated" and "lowly methylated" not defined (comparing to classification 0, 1, 2)
- Supp. Fig. 1: No y axis label in the figure (top). What is shown is not the same as the data in the table.
- Ext. Fig. 2: Variance of s_{het} is not defined. Is it computed from the posterior density?
- No definition of "underpowered" (l. 167).
- "Most genes with few expected pLOF variants are indeed unconstrained" is not a valid statement. Instead, the constraint cannot be measured to be $s_{het} > 0.075$.
- Supp. Fig. 4: LOEUF (x axis) not defined, units not explained.
- "regions"  "genes" (l. 217)
- Set of 635 "LOF-constrained genes" not defined (l. 224).
- Why are ClinVar variants with a rating of zero stars used in Fig. 3A, when in the entire rest of the manuscript pathogenic ClinVar variants were defined as having at least two stars?
- "pKOs are significantly less constrained than expected" is not a valid statement, as there is no expectation given. Relative statement can only be made in comparison to something (e.g. the mean).
- It is not stated which criteria were used to select the variants shown in Supp. Table 7.3 and 7.4.
- Gene ATXN2 is not contained in Supp. Table 7.4, nor any other gene associated with COPD (l. 334).
- The "intersection set" of the two splice scores should be clearly defined at the beginning (l. 371) and consistently referred to as the "intersection set". Incorrect use of "both" (twice).
- "splice sites" (l. 373) seem to be canonical splice sites as defined by LOFTEE (LC & HC). This should be clearly stated.
- Inconsistent use of splice site sets in l. 379, switching from the default intersection set (with 11% of missense+synonymous variants) to the two separate sets, which give higher percentages.
- Extended Fig. 4: x axis ticks should be brackets, not thresholds.
- Different step size in Extended Fig. 3 and Methods (l. 837).
- There is a category ("mouse lethal") missing from the bar chart in Fig. S1.
- Formula formatting (l. 737, 813).
- Fig. 5: Clearly name the set of missense variants predicted to be deleterious by five algorithms and use that definition consistently throughout (x axis label in A).
- Fig. 5: Missing category labels in C.
- Fig. 5: "upstream" and "downstream" not defined.
- Fig. 5: CADD score not defined, not mentioned in text or caption.
- Fig. 5: Inconsistent labeling in caption and table for B.
- Why are there almost 310k variants in Fig. 5B when only ~293k SAVs were identified (l. 370)? How many intronic, upstream and downstream variants are there?

Referee #3 (Remarks to the Author):

The authors present a bioinformatic analysis of the exomes of almost 1M participants drawn from various biobanks and studies around the world. The phenotypes of the participants have not been included in the analysis. Uniquely for a public variant resource of this scale, the DNA library preparation and sequencing were performed by a single organisation, limiting technical heterogeneity in the data generation process (although two different types of sequencer were used). The key components of the paper are as follows:

- a catalogue of 16.5M single nucleotide variants (SNVs) and small insertions/deletions (indels) in protein coding genes,
- analyses of selection pressures in humans using statistics such as s_{het} , MTR and F_{st} ,
- an analysis of variants likely to affect splicing, and
- a survey of variants previously identified through clinical genetic reporting and deposited in ClinVar.

The bioinformatic analyses are mostly descriptive or concern reapplications of previously developed methods, with some adaptations. Nevertheless, the execution of the methods is on a much larger scale than in previous work, and of a very high standard. The absolute number of unrelated non-Europeans in this collection is ~164,000, which is about 70% higher than TOPMed Freeze 8. The output will undoubtedly be a valuable resource for genetics researchers and clinical geneticists around the world.

Major Remarks:

(1) I have some concerns about the robustness of conclusions based on observations in single individuals. In a dataset as large as this, the probability of observing a certain type of artefact in at least one participant may be relatively high, even if the overall accuracy of variant and genotype calling is also reasonably high. Note that the variant false discovery rate (FDR) is estimated to be 8% (as stated on Line 615, "Precision=0.92"). The FDR is not broken down by variant type (SNV vs indel), but presumably false positives (FPs) are enriched in indels compared to SNVs. I don't think estimates of the genotyping error rates are given, nor estimates of the genotype probabilities conditional on a variant being a FP or being of a certain type (SNV or indel). However, variant calling errors and genotyping errors could, combined, lead to false variant calls such as false predicted knockouts (pKOs), especially those induced by false indel calls annotated as frameshift variants. To what extent could erroneous variant calls affect the validity of statements based on observations in single participants such as "62.6% of homozygous pLOF variants are found in one participant each" and "1,838 of the putative gene knockouts (pKO) have not been previously reported"?

(2) I found the analysis of predicted splicing scores using recently developed methods such as SpliceAI and MMSplice compelling, in particular the use of MAPS to identify biologically/clinically useful thresholds. However, it is not ideal that variants of all types were aggregated in the score thresholding. I assume that a relatively high proportion of high-scoring variants are canonical splice site variants, which have been known for some time to have important molecular consequences (and indeed represent the consequence category with the highest mutability-adjusted proportion of singletons (MAPS) score in Fig. 5A). It would be preferable, instead, to identify the threshold required to achieve a biologically or clinically useful MAPS separately for each consequence category. For example, when considering a canonical splice site variant, a threshold of zero would give a MAPS of ~0.14, but when considering a missense variant, a higher threshold would be required to give a high MAPS, and so on. Specific thresholds for different variant consequence classes would be particularly helpful in a clinical genetic setting.

(3) For publication in a journal of this calibre, I would have expected the inclusion of copy number variant (CNV) analysis, especially considering the availability of the SNP data generated alongside which could help control false CNV calls from the exome sequencing data.

(4) It would be helpful to provide more detail in the main text and in methods regarding the aggregation and exclusion of participants in this collection. For example, some effort was made to exclude individuals with rare diseases (lines 561-562), but the precise methodology for doing this is not given. Was it based on ICD10 codes? If so, what rules were used to exclude participants? Although the exclusion criteria will have led to a depletion of cases with rare diseases, some patients will undoubtedly have been included anyway, with implications for the interpretation of results given throughout the manuscript (e.g., the pKO section).

(5) The clinical genetics section is the least compelling, especially considering the absence of any phenotype data on the participants carrying the "pathogenic" variants. Below are a few comments that may help improve it, assuming phenotype data cannot be brought to bear on this analysis.

- The first paragraph concerns a look-up of ClinVar variants in the exome data and includes the statement "Each individual harbors on average 1.6 pathogenic variants, presumably as heterozygous carriers for recessive diseases." I believe it is not necessary to presume, because the authors could look up the genotypes for these variants and obtain the mode(s) of inheritance (MOI) of the relevant genes in a gene panel database, such as PanelApp. (The OMIM data appear to have MOI data for only 280 autosomal dominant genes and 1,087 autosomal recessive genes, which is a small fraction of the number of known aetiological genes for rare diseases, so that resource is not ideal for this purpose.)

- The second paragraph restricts the ClinVar look-up to 76 autosomal genes in the ACMG Secondary Findings (SF) list. Here, the statement "2.38% of RGC-ME carry an actionable variant" considers variants responsible for dominant and recessive disorders together. It would be interesting to report what proportion of participants have an allelic configuration that, in principle, increases their risk of disease (i.e., with a P/LP variant on both alleles in the case of recessive disorders). The percentage of at-risk individuals (2.38%) seems high, given this concerns only 76 genes, so it would be interesting to provide the corresponding percentage for all known aetiological genes for comparison, bearing in mind that only a few percent of individuals in general are thought to have a rare disease.

- In the third paragraph, the absolute numbers given in the comparison of ancestry groups after sub-sampling are hard to interpret, because they depend on the fact that the smallest ancestry group (SAS) happens to have ~30k participants, which is arbitrary. For this reason, the text might be more interpretable if it places greater emphasis on percentages and ratios. Lines 455 to 464 in particular are hard to parse and interpret, because variants can have different allele frequencies and may be observed in multiple individuals of the same or different ancestries (and furthermore, ancestry is defined according to a 50% assignment threshold, which might not necessarily encompass the haplotype harbouring the variant being considered - an issue that deserves mentioning). In my opinion, it would be preferable to focus on the per-individual results. They show that the distributions of the per-individual number of ClinVar pathogenic variants across the ancestry groups are remarkably similar, with median counts equal to 7 for AFR and SAS and 8 for EUR and IAM (Supp.

Fig. 11, right panel). I would elaborate on this result, because, in the context of that easily-missed supplementary figure, the statement "individuals of European ancestry have a median of 18.2% more known pathogenic variants than individuals of African ancestry" seems selective and a bit misleading. (It also seems at odds with the $8/7 = 14.29\%$ increase implied by the figure, which, in addition, ought to include statistical testing results). The higher number of VUSs amongst individuals of AFR ancestry is to be expected because AFR individuals have more variants in general due to greater genetic diversity. For these reasons, it seems unclear whether the observations can or indeed need to be attributed to "ascertainment bias" and limited "access to genetic medicine."

- Regarding the fifth paragraph, it seems plausible that some VUSs in ClinVar will have a higher allele frequency (AF) in non-Europeans simply because accurate AFs in the relevant non-European populations were not available at the time of classification. Can this be assessed? I don't think a pathogenic classification would ever have been assigned to a variant with knowledge that its AF was $>1\%$ in certain populations.

- In the sixth paragraph, the claim that variants in regions of greater human constraint are "likely pathogenic" seems overstated, and the term "likely pathogenic" should be reserved to ACMG criteria to avoid confusion. Reclassifying VUSs is a tremendous challenge. If the authors think the constraint metrics and splice prediction score thresholds provided could be used to reclassify VUSs in significant numbers in practice, with or without additional experimental work, it would be helpful for this vision to be described more fully.

Minor Remarks:

- The "diversity" of the collection is repeatedly touted in the first part of the manuscript. However, only 18% of participants are non-European (Supp. Table 1), so it is not very diverse. The percentage is about the same as genetic studies with distributions approximately aligned with the UK census (e.g., <https://doi.org/10.1038/s41586-020-2434-2>) and significantly less than gnomAD v3 (<https://gnomad.broadinstitute.org/help/what-populations-are-represented-in-the-gnomad-data>), 47% of participants of which are non-European. The strength of the RGC-ME dataset is not the diversity (i.e., the heterogeneity in ancestry groups), but the absolute number of individuals from different non-European ancestries who are included (amounting to 179,560 individuals, including relatives). Perhaps the authors could amend the text to reflect this reality more faithfully. In addition, the statement "exome sequencing of 985,830 individuals of diverse ancestry" might be more clearly rewritten as "exome sequencing of 985,830 individuals across a diverse array of ancestries," as diversity is a property of the collection rather than the individual. It is also not, strictly speaking, a property of data ("ancestrally diverse exonic data" in line 71).

- Line 193: "Enrichment in pathogenic missense variants is significant for all regions where $MTR \neq 1$." I didn't quite follow this. $MTR=1$ only when the observed fraction is exactly equal to the expected fraction. Why would there be enrichment for pathogenic missense variants when $MTR < 1$ as well as when $MTR > 1$? Could the authors include P-values wherever enrichment is referred to (here and elsewhere)? Here, it would also be helpful to show the distribution of MTR for ClinVar pathogenic variants.

- Fig. 3: shouldn't "rank" be "percentile"?

- The language is ambiguous in certain sections. E.g., I assume the "pKOs" in "pKOs are significantly less constrained" refers to genes for which at least one individual is homozygous for a pLOF variant. (A pKO is an individual-and-gene-specific observation rather than a gene characteristic.)
- I wonder if the authors could speculate as to the reasons for the pharmacogenomic results around line 289 - drift or selection?
- Lines 304 to 311 presuppose that the participants with pKOs do not suffer clinical consequences as a result of these pKOs. How confident can one be about this? Please refer back to point (4).
- Line 336 contains a stray ")"
- Line 352: The sentence "Here, we use human genetic data to define splice prediction thresholds that identify deleterious variants that affect splicing" might be more clearly stated as "Here, we use human genetic data to optimise splice prediction score thresholds that enrich for deleterious variants that affect splicing."
- Fig. 5a: what do the colours refer to? This is not stated in the legend.
- Fig. 5c: what is shown along the y-axis?
- Line 422: "previously unobserved" is not quite right because a variant had to be observed previously to end up in ClinVar
- Line 599: "transmitted singletons" - if a variant is known to be transmitted, presumably that's because it was seen in multiple related individuals, but in that case the variant would not be a singleton according to the usual definition.
- Line 720: "These sites will experience fewer mutations than expected" doesn't seem correct: the sites do not experience fewer mutations, mutant alleles are selected against.
- Extended Fig. 1: the vertical line denoting a threshold is misleading given that the x-axis is not continuous (each set of horizontally arranged bars along the x-axis corresponds to a specific value of s_{het}).
- Extended Fig. 3: I can't work out why there are horizontal confidence intervals
- Extended Fig. 4 legend: "~500 variants have" should read "~500 VUSs have"
- Supp. Fig. 1 y-axis unlabelled.

Author Rebuttals to Initial Comments:

Updates to the manuscript

We thank all the referees for their insightful comments. We have updated the manuscript with details of additional analyses, new figures, and tables. Changes with respect to the first submission are as follows:

1. We have removed a cohort of individuals with myelodysplastic syndromes. Therefore, our analysis is based on a slightly smaller dataset of 983,578 individuals (initial submission was based on 985,830 individuals).
2. We have included new supplementary figures and tables as shown below.
3. Supplementary Tables 8-12 and 14-15 are included with the Supplementary Figures and Tables.
4. Supplementary data table 3 is available through Figshare.
5. All other Supplementary data tables are included as separate files

Table/Figure	Description	Notes
Extended Figure 4	Updated to include local ancestry specific AAF on right y-axis, which corroborates sub-sampling ancestry specific AAF in each bin (x-axis). The local ancestry specific AAF for each variant is defined as the summation of ancestry component probabilities over all 822K unrelated individuals and reflects local ancestry specific to the genomic segment.	Response to Referee 3, Comment 5.3
New Supplementary Figure 3	Receiver operator curves showing the discrimination between the “high” and “low” constraint genes using different constraint metrics: $S_{\text{het-RGCME}}$, $S_{\text{het-ABC}}$, and LOEUF, for transcripts with values from all three methods.	Response to Referee 2, Comment 2
New Supplementary Figure 4	Comparison between $S_{\text{het-RGCME}}$, $S_{\text{het-ABC}}$, and LOEUF based on hard cutoffs to determine highly constrained genes.	Response to Referee 1, Comment 1
New Supplementary Figure 5	Breakdown of $S_{\text{het-RGCME}}$, $S_{\text{het-ABC}}$, and LOEUF scores for genes that lacked concurrence between the 3 scores shows that strict threshold cutoffs mask the continuous distribution of scores particularly if the cutoffs depend on lower/upper bounds instead of the mean.	Response to Referee 1, Comment 1
New Supplementary Figure 6	Our estimates of $S_{\text{het-RGCME}}$ benefit from 1) larger sample sizes – demonstrated by comparing with precision in $S_{\text{het-ABC}}$, and 2) a method not sufficiently powered for shorter genes. LOEUF, a site-based metric, is highly dependent on the length of a protein. When there are few expected LOFs due to length/sequence composition of the underlying gene, LOEUF is underpowered to detect constraint.	Response to Referee 1, Comment 1
New Supplementary Figure 8	Schematic of MAPS-derived filters for MTR.	Response to Referee 1, Minor Comment 9
New Supplementary Figure 9	MTR-based segmentation shows better resolution than MPC-segments.	Response to Referee 1, Comment 2
New Supplementary Figure 10	Summary statistics for MTR constrained regions: 1) distribution of the number of MTR constrained regions per gene and 2) ratio of lengths compared with MPC-regions.	Response to Referee 1, Comment 2
Supplementary Figure 11	Altered Fig. 3 to make better use of space. Removed gene enrichment analysis as a response to Referee 2, Comment 7.	Modified figure with the addition of Panel A.

		Moved Panel A from previous Fig 3
Supplementary Figure 14	New analysis using splice score thresholds derived using exclusively non-splice-sites (non-SS) shows 1) fraction of predicted splice affecting variants (SAV) validated as splice disrupting variants and 2) odds ratios for deleterious non-SS SAVs in coding variant groups.	Moved Panel C from previous Suppl. Fig. 9 Response to Referee, Comment
Supplementary Figure 17	Percent difference of mean per-individual counts in ClinVar categories (pathogenic 0+, 1+, 2+, VUS+CI) across continental ancestries using all 822k unrelated samples, with respect to EUR, shows an inverse relationship between confidence of pathogenicity and ancestry specific per-individual count relative to EUR.	Response to Referee 3, Comment 5.3
New Supplementary Table S4	List of continuous segments of missense constrained regions based on MTR values and the top 15-percentile threshold.	Response to Referee 1, Comment 2
New Supplementary Table S5	Jaccard index analysis between the MTR-constrained regions and protein domain categories from UniProt.	Response to Referee 1, Comment 2
New Supplementary Table 14	Support vector machine (SVM) quality control method performance metrics on a test set of 77,005 variants.	Response to Referee 3, Comment 1
New Supplementary Table 15	Proportion of likely high-quality variant calls by variant type in the QC pass dataset based on the observed fraction of QC pass variants and the metrics derived from SVM test set classification shows that, across variant types, we estimate the proportion of post-QC "true" variants to be no lower than 97.5%.	Response to Referee 3, Comment 1

Responses to Referee 1 Major Comments

Referee 1 Comment 1	Estimates of selective constraint against heterozygous loss of function (s_{het}) are compared against estimates from 2017 (Cassa et al., N=61k). These should be updated to reflect the existing state of the art. First, they should be compared against LOEFF estimates from gnomAD, which involved a >2x larger sample size. Although the gnomAD paper did not quantify s_{het} (strangely), it is still possible to compare the rank-ordering of genes. What is the Spearman correlation? Are there any genes that are estimated to be constrained in gnomAD but not in GCM-ME? Second, they should be compared with the s_{het} estimates of Agarwal et al. 2023 eLife, which utilized the 55k non-Finnish EUR samples in gnomAD and fit a more realistic demographic model as compared with Cassa et al. Optionally, they could also be compared against the non-peer-reviewed s_{het} estimates from Zeng*, Spence* et al. bioRxiv (https://doi.org/10.1101/2023.05.19.541520), which are based on 125k gnomAD samples and a sophisticated new estimation method.
Authors' response	Thank you for the suggestions. As requested, we have updated the manuscript with additional comparisons.  a. We have compared s_{het} computed with RGC-ME to LOEUF reported by Karczewski et al. (2020) and s_{het} estimates from Agarwal et al (2023) which we refer to as $s_{het-ABC}$. Spearman correlations to LOEUF and $s_{het-ABC}$ have been added to this section and are relatively high (absolute correlation value ~ 0.75). We retained comparisons to the ExAC data as a 1-to-1 comparison with the same s_{het} computation method (first described in Cassa et al. and now updated with Seplyarskiy, et al.) b. We have also performed extensive comparisons between the three constraint metrics. LOEUF, a site-based metric, is highly dependent on the length of a protein. When there are few expected LOFs due to length/sequence composition of the underlying gene, LOEUF is underpowered to detect constraint. We elaborate on this aspect in the text and added three new figures: Supplementary Figures 4, 5 and 6. While $s_{het-ABC}$ does not suffer from this aspect, s_{het} derived from RGC-ME is more precise due to a larger sample size. Out of the 2,689 LOEUF-constrained genes, 496 are unconstrained according to s_{het}, 270 of which are also constrained according to $s_{het-ABC}$. We examine the distribution of s_{het} values from RGC-ME for the remaining 226 genes that are only s_{het} unconstrained in the new Supplementary Fig 5A, which shows the mean (mean=0.052, median = 0.056) and 95% lower bound (mean=0.041, median = 0.043) are relatively high. LOFs in genes with these scores would be predicted to have $\sim 5\%$ impact on relative fitness. The main points that we want to highlight from these analyses are:  a. LOEUF is underpowered to detect constraint for genes with few expected LOF counts even if sample size increases. b. While it is reasonable to use hard cutoffs to come up with a list of constrained genes, the continuous nature of fitness costs and the nuanced interpretation of values close to the threshold should not be overlooked.

Changes in text	Lines in text: 180 – 201 We compared constrained genes defined by hard-cutoffs and found 3,657 s_{het}-constrained transcripts that also had LOEUF and $s_{\text{het-ABC}}$ estimates (Supplementary Fig. 4). 496 genes were LOEUF-constrained and s_{het}-unconstrained, 54% of which (270) were also unconstrained based on $s_{\text{het-ABC}}$ maximum a posteriori (MAP) estimates (<0.073) or 95% confidence interval (CI) lower bound (<0.021). 226 genes that are constrained based on both $s_{\text{het-ABC}}$ and LOEUF but unconstrained in RGC-ME had high s_{het} means (mean=0.052, median = 0.056), demonstrating that strict cutoffs mask the continuous nature of constraint scores (Supplementary Fig. 5). Overall, $s_{\text{het-ABC}}$ values for the 3,657 constrained genes were high (mean=0.171, median=0.148) and 75% (2,733 genes) had $s_{\text{het-ABC}}$ MAP > 0.073. Genes that were deemed constrained in both RGC-ME and by $s_{\text{het-ABC}}$ confer greater confidence in determining constraint, as the underlying datasets from which they have been derived and methodology are both distinct. Compared to $s_{\text{het-ABC}}$, estimates from RGC-ME benefitted from a nearly 15x increase in sample size and consequently had median HPD ranges of half the size (range-s_{het}=0.27 compared with range-$s_{\text{het-ABC}}$ =0.53, Supplementary Fig. 6A). 40% of the s_{het}-constrained genes were LOEUF-unconstrained (with LOEUF\geq0.35), 13.7% of which were autosomal dominant or haploinsufficient genes (Supplementary Fig. 4). In the case of AD/HI genes, only 19% of the expected LOFs were observed on average in the LOEUF dataset and their mean $s_{\text{het-ABC}}$ was 0.11 (median=0.09), providing consistent evidence by all three methods that these genes are under selection. Despite the ratio of observed-to-expected pLOFs suggesting that the AD/HI genes are constrained, they are deemed to be unconstrained when evaluated using the LOEUF metric.
--

Referee 1 Comment 2	Estimates of missense regional constraint also need to be compared with existing estimates from refs. 32-33. Does the increase in sample size lead to improved sequence-length resolution?
Authors' response	Data from the 822k unrelated sample set suggests that sequence-length resolution is improved. We demonstrate discrimination between ClinVar 2+ benign and pathogenic variants for MTR computed on 21-codon windows as strong as scores computed on 31-codon windows up to the top 3-percentile of MTR. In addition, we identified contiguous regions of missense constraint based on residues in the top 15-percentile of MTR exome-wide. We find a greater number and smaller regions of constraint compared with another missense regional constraint metric incorporated in MPC (which we call MPC-segments). We benchmark MTR constrained regions against MPC-segments, as no other score generates transcript-wide regions of constraint based exclusively on missense variation. These data have been added to the manuscript. Points that we want to emphasize:  1. Due to increased sample size, we can calculate MTR with smaller window size of 21 amino acids for increased resolution (we show this in the updated Fig 3A with 21 aa window) 2. Compared to previously described missense-constrained regions (i.e., MPC-segments), we can identify a higher number of constrained regions within a gene due to improved resolution (new figures: Supplementary Fig. 9 and 10). These constrained segments are enriched for pathogenic ClinVar and de novo variants compared to benign variants.
Changes in text	Lines in text: 21 amino acid windows (253-258, Fig 3A)

	In addition, the increased power from 822k samples resulted in higher resolution to distinguish pathogenic from benign variants for MTR computed with 21-codon windows, albeit at the expense of having fewer missense variants scored overall (295,176 constrained missense variants). Lines in text: MTR constrained regions (276-288) A previous report estimated regional missense constraint from ~60k ExAC samples based on observed to expected missense ratio and derived a composite missense deleterious score called MPC. We refer to the MPC regional constraint as MPC-segments and benchmark MTR constrained regions against them. We defined MTR constrained regions as continuous segments within a protein containing variants with MTR values in the top 15-percentile threshold (see Methods, Supplementary Fig. 9). MTR constrained regions have a median length of 25 residues (14-40, Q1-Q3). We compared MTR constrained regions to MPC-segments with an observed-to-expected ratio of missense variants ($\gamma \leq 0.6$, defined as constrained regions by Samocha et al.). Overall, we identified 8-fold more constrained MTR regions compared to MPC-segments ($\gamma \leq 0.6$) across 1,853 transcripts that had data for both (Supplementary Fig. 10A). Among 421,012 MTR constrained amino acids, average “missense depletion” based on MPC-segments would misclassify 165,815 (39.4%) of them as “unconstrained” ($\gamma > 0.6$; Supplementary Fig. 10B).
Referee 1 Comment 3	Line 281 states that pKO genes in which a homozygous knockout is observed are “less constrained than expected,” but this statement is apparently based upon a strawman “expected” under which the number of homozygotes is independent of the number of heterozygotes. Instead, the expected fraction of homozygotes for an allele with frequency p is p^2, and pLOFs in constrained genes have smaller p. Are pKOs significantly less constrained than expected *after accounting for the number of heterozygotes*? Exome-wide, is there a depletion of homozygous pKOs vs. expected (i.e., deviation from HWE after accounting for autozygosity)? If so, are the gene families discussed in lines 287-300 significantly less depleted for homozygotes?
Authors’ response	We meant that pKO genes had significantly lower s_{het} than all other genes. The reviewer brings up a good point, though, that these genes likely have greater cumulative MAF for pLOFs across the board and would be expected to have a greater number of LOFs (and correspondingly lower s_{het}). Therefore, we performed additional analyses that compare the number of genes with fewer homozygous carriers than expected (assuming HWE) between various groups, i.e., constrained versus unconstrained, and genes in large families versus all other genes. For this analysis, we included variants with AAF < 1% in any of the 95 fine-scale ancestry components. The median ratio of observed pKO carriers to expected was 0 for both constrained and unconstrained genes, suggesting overall depletion of rare, homozygous pLOFs across the exome. In this dataset, we are aware of increased homozygosity in certain cohorts, e.g., CNCD, a study of individuals with high levels of consanguinity from Pakistan, and MCPS, a study of individuals from Mexico City reported to have increased homozygosity (Ziyatdinov et al., 2023). However, even without making explicit adjustments for autozygosity, we observe a decrease in rare homozygous pLOFs across the entire exome relative to expectation.
Changes in text	Lines in text: 384-396 pKOs were significantly less constrained with lower s_{het} values (on average -0.075 [95% CI: -0.078, -0.072]) relative to all other genes. Only 2.77% of pKOs had $s_{het} > 0.073$ compared to 21.6% of all human genes, and 47.2% of pKOs were in the lowest quintile of s_{het} scores exome-wide ($s_{het} < 6.265 \times 10^{-3}$). A caveat is that s_{het}, like most gene-specific measures of constraint, is designed to capture selective pressures on heterozygous pLOF variant carriers. Although genes harboring biallelic pLOF variants are

	under less heterozygous selective pressure, existing sample sizes are inadequate to directly compute selection on homozygous variation. We also estimated the expected number of rare, homozygous pLOF carriers per variant based on the assumption of Hardy-Weinberg equilibrium ($N_{\text{tot}} \times \hat{p}_{\text{LOF}}^2$, see Methods). 97.9% of constrained genes had fewer than half of the expected homozygous carriers compared with 74.2% of unconstrained genes. The median ratio of observed pLOF variant homozygotes to expected was 0 for both constrained and unconstrained genes, suggesting overall depletion of rare, homozygous pLOFs across the exome.
--	--

Referee 1 Comment 4	In the section on differential allele frequency, it is heavily implied that the signals of high F_{st} are due to population specific selection. This is evidenced in line 313-314, 322-323, 329-331, and 345-346. Population-specific selection is a sensitive topic, and I am concerned that with its vagueness, this section is open for misinterpretation. The question is whether population specific selection on coding variation is widespread or rare; as currently written, the manuscript vaguely suggests that it is widespread. However, Supplementary Figure 8 shows that the fraction of variants with $F_{\text{st}} > 0.05$ (an *extremely* lenient threshold) is ~identical for missense vs. synonymous variants, indicating that most variants at this threshold are not under population specific selection. This is expected because genome-wide F_{st} is greater than 0.05 for all of these ancestry-group pairs. Similarly, in Supplementary Figure 7, the dotted line at 0.15 suggests that variants above that line have some special meaning (because typically, a line on a Manhattan plot is a stringent significance threshold). But again, genome-wide average F_{st} is routinely >0.1 between continental groups, so a variant-level threshold of 0.15 is very lenient. On lines 329-336, a number of anecdotes are presented about the phenotypic effects of some of these variants with $F_{\text{st}} > 0.05$. The suggestion is that these phenotypic associations are responsible for the observed difference in allele frequency. Considering the large number of synonymous variants with $F_{\text{st}} > 0.05$ (and the large number of noncoding variants, if one were to look in a WGS dataset), this is unlikely. On lines 336-339, it is stated that variants with $F_{\text{st}} > 0.05$ are enriched for significant GWAS loci. However, the odds ratios reported are too large to be driven by selection, considering the lenient threshold. Does this analysis account for allele frequency in the discovery population? If not, perhaps the correct interpretation is that variants at high frequency in EUR are more likely to have $F_{\text{st}} > 0.05$ (Supplementary Figure 8) and also more likely to be in the GWAS catalog (because GWAS loci are common). I think this section would still be of interest if it were purely descriptive – how do allele frequencies differ across ancestries? – rather than speculative. If the authors wish to retain content about population-specific selection, I think the claims should be explicit, and they should be backed by much stronger evidence. This would likely entail forward simulations under a null model with a realistic demography and non-population-specific negative selection.
Authors' response	Thank you very much for the thoughtful critique of this section. We did not intend to imply any kind of population-specific selection. We agree with the reviewer's concerns about the potential for misinterpretation and have modified this section to be purely descriptive. We have removed the text that may be interpreted as related to selection. This section now contains a brief description of results and supplementary tables that include variants with $\max F_{\text{ST}} > 0.15$ across major ethnicities. We provide this as a resource of differentiated alleles based on the largest exome dataset available to date.
Changes in text	Lines in text: 610-624

	The breadth of continental ancestries represented in RGC-ME provides an opportunity to identify coding variants with allele frequency differences between populations arising due to selection or drift. The higher frequency of differentiated alleles in selected populations compared to well-studied European populations can result in improved power in association analysis and aid in the identification of associations of genetic variation to medically relevant phenotypes. We estimated the genetic differentiation between six major populations (AFR, IAM, EAS, EUR, MEA, and SAS) by calculating pairwise Hudson’s F_{ST} at the variant level (Supplementary Table 13). These data provide a comprehensive catalog of population differentiated variants. Several highly differentiated functional variants were found in genes known to be subject to natural selection. These include genes related to skin pigmentation (e.g., SLC24A5, SLC45A2, OCA2, MC1R, TYR), the immune system (e.g., TLRs, F5, OAS1, APOL1, HLAs, ABO), metabolism (e.g., ADH1B, ALDH2, LCT), and high-altitude adaptation (e.g., EPAS1). Variants with max $F_{ST} > 0.15$ are highlighted in Supplementary Table 13, along with the alternate allele frequencies in the respective ancestry.
--	---

Referee 1 Comment 5	The last Results subsection identifies thresholds for spliceAI and MMSplice scores such that splice-altering variants above that threshold have allele frequency spectra similar to missense deleterious variants. The analysis is motivated by the statement that existing tools “generally lack clear guidance for their use in detecting deleterious splice affecting variants.” However, the (Jaganathan et al., ref 58) contains the following text: At common allele frequencies, high-scoring predicted cryptic splice variants (score > 0.8) are under strong negative selection, as evidenced by their relative depletion compared to expectation (Figure 4A). At this threshold, where most variants are expected to be close to fully penetrant in the RNA-seq data (Figure 2D), predicted synonymous and intronic cryptic splice mutations are depleted by 78% at common allele frequencies, which is comparable with the 82% depletion of frameshift, stop gain, and essential GT or AG splice-disrupting variants. Their analysis is extremely similar to that presented in this subsection. The only important difference is that Jaganathan et al. chose to calibrate vs. loss-of-function as opposed to missense deleterious variants, such that they end up choosing a more stringent threshold. I think this section should be re-framed to acknowledge this.
Authors’ response	This is a fair comment. The section has been reframed accordingly.
Changes in text	Lines in text: 439-448 Several prediction tools have been developed to understand the effects of genetic variants on alternative splicing. While these tools primarily assess if a genetic variation affects splicing, some also provide a pathogenicity metric or score threshold as a measure of deleteriousness. For example, it has been shown that predicted cryptic splice sites with SpliceAI scores > 0.8 are validated at high rates using RNASeq and are as depleted at common allele frequencies as pLOF variants (Jaganathan et al.). Here, we used human genetic data to optimize splice prediction score thresholds enriched for deleterious variants that affect splicing. We systematically quantified the deleteriousness of variants at various splice prediction score thresholds by calibrating to MAPS scores that are comparable to that of 5/5 missense variants, i.e., variants predicted to be deleterious by five out of five prediction methods in dbNSFP (v3.2, see Methods).

Responses to Referee 1 Minor Comments

Referee 1 Comment 6	Lines 87-88: although the absolute number of non-EUR individuals is greater than existing studies, the fraction of non-EUR individuals remains low (and is much lower than TOPMed). I suggest tempering the statement that "RGC-ME has more population diversity than other large scale genetic variation datasets." Perhaps it would be more accurate to state that the dataset represents a large increase in sample size, including for non-European ancestries.
Authors' response	We have modified the sentence and highlight the large increase in non-European samples. The count of non-EUR in RGC-ME now includes individuals of unassigned ancestry for consistency with external cohorts and is ~190k (23% of the dataset). Individuals of Ashkenazi Jewish, Amish, and Finnish ancestry are classified as EUR.
Changes in text	Lines 83-87 Over 190k of the unrelated participants (23%) are of non-EUR ancestry in RGC-ME compared to 35k in gnomAD genomes (v3.1.2), 53k in gnomAD exomes (v2.1.1), and 91k in TOPMed Freeze 8, indicating that RGC-ME represents a large increase of non-European participants in genetic variation datasets (Fig. 1A, Supplementary Table 1).

Referee 1 Comment 7	Lines 99-100: I think the distinction here between variants and sites could be clarified.
Authors' response	We have updated the text.
Changes in text	Lines 92-95 We identified 16,425,629 unique mutated genomic positions (i.e., sites) in autosomal and X chromosomal coding regions, with one unique reference-alternate allele change (i.e., variant) every two bases on average.

Referee 1 Comment 8	Line 217: I think you mean, "To identify genes depleted of missense variation"
Authors' response	We have modified the sentence (now line 347) which clarifies the context.
Changes in text	Line 347: 3,955 genes contained regions depleted in missense variation with a significant proportion of their coding sequence in the top 15-percentile of MTR exome-wide (binomial test with $\pi_0=0.15$, $p<0.05$ after multiple testing correction, Supplementary Table 6).

Referee 1 Comment 9	217-232: the approach used in this paragraph needs some justification. What is the rationale for looking just at enrichment in the top 1% of MTR, and throwing out variation below that threshold?
Authors' response	We used 1-percentile as a stringent threshold to define regions that are highly depleted of missense variation. In this resubmission, we have updated the threshold to a 15-percentile threshold and its justification based on variant deleteriousness (see MAPS analysis) is provided.
Changes in text	Lines 260-273

	Deleterious variants are expected to have lower allele frequencies than neutral variants due to negative selection. We can infer the degree of selection between different functional classes of variation by comparing the proportion of singletons in each class. Accounting for both background mutation rate and methylation level, we computed the deleteriousness of variants using an updated mutability adjusted proportion of singletons (MAPS) metric Lek et al. . We used the MAPS metric to derive an MTR score threshold based on empirical allele frequency data that corresponds to missense variants predicted to be deleterious by five prediction algorithms in dbNSFP (v3.2), i.e., 5/5 missense variants. Variants with MTR values in the top 15-percentile exome-wide threshold (MTR < 0.841) were predicted to be as deleterious as 5/5 missense (Supplementary Fig. 8). For 31-codon windows, 9.13% (953,960) of all missense variants (excluding known ClinVar pathogenic variants) observed in RGC-ME have MTR scores in the top 15-percentile. These missense variants in the top 15-percentile exome-wide MTR are potentially deleterious and can be useful for prioritization in disease gene discovery projects.
--	---

Referee 1 Comment 10	268-270: My understanding is that pKOs are defined as genes with homozygous pLoF alleles, and that compound heterozygotes are excluded. If this is correct, is this choice made because the data are unphased? If so, what are the prospects for phasing RGC-ME? (This is significant because presumably, there are many more compound hets than homozygotes in these data).
--

Authors' response	We thank the reviewer for this comment. We have now phased the RGC-ME dataset with SHAPEIT5 and used the phased dataset to identify compound heterozygotes. We identified compound heterozygous variants in 1,205 genes and have added this analysis to the manuscript. Of these, 162 genes did not have a homozygous pLoF variant. We found that more genes are pKOs due to homozygous pLoF variants than compound heterozygotes (4,686 versus 1,205, respectively). Our findings are consistent with recent preprints that found pLoF variants in the same gene in an individual are more likely to occur on the same haplotype than opposite haplotypes (Guo et al., Lassen et al.)
-------------------	--

Changes in text	Lines in text: 374-377 Furthermore, we identified 1,205 genes with carriers of two rare (AAF<1%) heterozygous pLoF variants in trans , i.e., compound heterozygotes, 162 of which lacked homozygous pLoFs. In total, 4,848 genes were discovered with carriers of biallelic pLoF variants where both copies of a gene were impacted by pLoF variation and could be described as putative gene knockouts (pKO).
-----------------	---

Referee 1 Comment 11	1299-300: this hypothesis makes sense, but is it actually the case that these genes and gene families are paralog-enriched? (Note that gene duplication is thought to result predominately in functional redundancy/dosage sharing, not neofunctionalization; Lan and Pritchard 2016 Science).
--

Authors' response	There is evidence of gene duplications in the CYP, SLC, and ABC gene families resulting in paralogous elements. CYP genes are known to have diversified through duplication (https://doi.org/10.1093/jb/mvq001), and we highlight paralogous pairs with pKO genes in Supplementary Figure 12A.  • CYP: Other efforts have described each CYP gene and its likely paralogs more extensively: https://doi.org/10.3390%2Fijms17071020 • SLC carriers have been organized into 65 families according to homology and function: https://doi.org/10.1080/09687688.2018.1448123,  ◦ more recently, bioinformatically: https://doi.org/10.1371/journal.pone.0271062
-------------------	--

	 • ABC: Analysis of sequence diversity in human ABC genes describe recent duplications and sequence similarity, particularly in a cluster of genes on Chr 17: https://doi.org/10.1016/S0022-2275(20)31588-1, https://doi.org/10.1093/hmg/5.10.1649 We have also updated the text to reflect that these genes may also be non-essential, as is the case for olfactory receptor, OR, genes.
Changes in text	Lines in text: 426-427 The presence of human knockouts in these gene families suggests that there may be functional redundancy between homologous genes or that they are not essential.
Referee 1 Comment 12	On the web portal, I suggest that it should display gene-level and region-level constraint metrics in addition to variant allele frequencies (similar to the gnomAD browser).
Authors' response	We have updated the web portal (https://rgc-research.regeneron.com/me/home) as suggested. We have included s_{het} values for gene constraint and an MTR track that depicts regional missense constraint across the gene.

Responses to Referee 2 Major Comments

Referee 2 Comment 1	1. Not much information is given about how the data was collected. Is there overlap with existing datasets (UKB, gnomAD, 1KGP)?
Authors' response	Yes, UKB data is included in RGC-ME. We have included details of the cohorts in the Methods section as well as in the public RGC-ME web portal (https://rgc-research.regeneron.com/me/data-contributors).
Changes in text	Lines in text: 679-697 We aggregated high quality whole-exome sequencing data from 983,578 individuals after removing samples in a rigorous quality control process. Samples were removed based on sequencing metrics including sequenced gender not matching the gender listed in the manifest, contamination, low coverage, and unresolved duplications. To ensure this sample set characterizes genetic variation representative of the general population, we excluded available samples from cohorts specifically enrolling participants with Mendelian diseases, neurodevelopmental disorders, blood cancers, and use-restrictions inconsistent with the analysis presented here. The data includes participants from biobanks: UK BioBank, Geisinger Health System, The Mexico City Prospective Cohort, Penn Medicine BioBank (https://pmbb.med.upenn.edu), BioMe BioBank (https://icahn.mssm.edu/research/ipm/programs/biome-biobank), Dallas Heart Study (https://www.utsouthwestern.edu/education/medical-school/departments/internal-medicine/research/dallas-heart/), Amish Research Clinic (https://www.medschool.umaryland.edu/endocrinology/Amish-Research-Program/About-Us/), Center for non-communicable diseases (https://www.cncdpk.com), Australian New Zealand MS Genetics Consortium (https://www.msaustralia.org.au/anzgene/) and a variety of case control studies for complex diseases such as psoriasis, rheumatoid arthritis, diabetes. The source of all data along with links to the projects (where available) are listed in the RGC-ME web portal (https://rgc-research.regeneron.com/me/data-contributors).
Referee 2 Comment 2	When comparing high and low constraint groups derived from disease genes (l. 155), the numbers given in the text are not meaningful because it is not stated how many genes are in each group. They should be replaced with a precision-recall (PR) or receiver operating characteristic (ROC) curve. The final list of genes used and their classification (“high constraint” vs. “low constraint”) should be provided in a supplementary table.
Authors' response	 • There are 1,481 and 3,739 genes in the high and low constraint categories, respectively. As suggested by the referee, we now show discrimination between the high and low constraint genes determined with s_{het} cutoffs using a ROC curve (new Supplementary Figure 3). We also show that s_{het} derived from RGC-ME has higher sensitivity and specificity compared to other published LOF constraint measures, such as LOEUF and an alternate method for estimating s_{het} based on Approximate Bayesian Computing that we refer to as $s_{het-ABC}$. • We included a column in Supplementary Table 2 to indicate genes that are in the “high” or “low constraint” group.
Changes in text	Lines in text: 153-159, 169-172 We used s_{het} to identify highly constrained genes in RGC-ME by comparing s_{het} scores of the canonical transcripts of known high constraint genes (HI and autosomal dominant [AD], including

	developmental) with s_{het} scores for low constraint categories of genes (haplosufficient and genes with rare homozygous pLOF variants from RGC-ME; Supplementary Fig. 3). Among 1,481 genes in the “high” and 3,739 genes in the “low” constraint groups, 89.0% with s_{het} greater than the mean (0.073) and 67.5% of genes with s_{het} above the median (0.021) belonged in the high group (Supplementary Table 2). s_{het}, derived from RGC-ME, had a higher sensitivity and specificity in differentiating between constrained and unconstrained genes compared to other published LOF constraint measures, such as LOEUF¹ and an alternate method for estimating s_{het} based on Approximate Bayesian Computing, which we refer to as $s_{het-ABC}$² (Supplementary Fig. 3).
--	---

Referee 2 Comment 3	How does s_{het} correlate with the GERP score? How does it compare with other scores based on conservation (phastCons, phyloP)?
Authors’ response	Correlation with GERP was added to the manuscript ($\rho=0.411$ with GERP++ RS scores averaged over all possible LOFTEE High Confidence pLOF variations within a gene). Rank correlation was slightly higher with phyloP, $\rho=0.472$ (100-way, vertebrate alignment). GERP is a measure of conservation based on fixed variation between distantly related species and does not adequately capture selection events that have changed between lineages.
Changes in text	Lines in text: 174-176 However, correlations were more modest between s_{het} and GERP++, an evolutionary constraint score (Spearman $\rho=0.411$ with GERP scores averaged over all possible pLOF variations within a gene).

Referee 2 Comment 4	Are the differences in estimated constraint between ExAC and RGC-ME, where seven genes were constrained only in the smaller ExAC cohort, due to sub-population composition differences between the two datasets? What is the direct comparison of s_{het} estimates between RGC-ME and ExAC?
Authors’ response	Overall, the correlation between ExAC and RGC-ME estimates are high ($\rho=0.7658$). We updated our ExAC s_{het} comparison set to a recently released version that uses a novel method to compute mutation rate (Seplyarskiy et al.). Concordance is strong between the two sets of s_{het} estimates (128 HI genes are constrained in both, 12 in ExAC only, and 7 in RGC-ME only). s_{het} values of the 12 HI genes not considered highly constrained using RGC-ME data are relatively high (mean=0.0458 and lower bound=0.0353), indicating that these genes could be expected to cause a ~5% deficit in fitness. We applied hard cutoffs to define highly constrained genes, though this does not sufficiently account for the continuous nature of fitness costs (see new Supplementary Fig. 4-5 and lines in text 180-191 for more discussion). The two genes out of 12 with the lowest s_{het} were SOX5 and GLI1, and neither had a non-EUR ancestry-enriched pLOF variant. The most common variant in SOX5, 12:23604534:C:T, has a higher frequency in EUR (AAF=1.3x10 ⁻⁴) compared with the other ancestries (ranging from AFR AAF=7.3x10 ⁻⁶ to unobserved in EAS and IAM); the 3 most common variants in GLI2 – 2:120971939:A:C, 2:120927411:C:T, 2:120797469:G:A – were all rare and slightly more common in EUR. We expect the deviations between ExAC and RGC-ME in calling HI genes highly constrained are more likely due to the use of strict thresholds to define such genes.

Referee 2 Comment 5	MTR (PMID: 28864458) assumes that all mutation events (synonymous, missense, nonsense) are equally likely. However, mutation rates are far from equal across nucleotide contexts and genomic regions. To account for nucleotide context dependence, the authors use mutation rate estimates from gnomAD for their s_het and MAPS analyses, but they do not do so for the MTR analysis. Therefore, the reliability of the MTR results is questionable. This is also consistent with the lack of any correlation between MTR and the conservation metric GERP, as other studies have found that human-specific and phylogenetic conservation scores are (weakly) correlated. Is there a correlation between MTR and being a 5/5 missense variant (cyan dots in Fig. 3C)? Methodologically, why do the authors not show an ROC/PR curve in Fig. 3A?
Authors' response	We adjusted the MTR calculation to account for mutation rate by substituting the expected number of synonymous and missense (derived from context-dependent trimer mutation rates) instead of all possible counts in the denominator. The difference attributable to this change is modest (see Fig 1R, added to the last page of this document). We have used the updated mutation-adjusted MTR for all analyses. The rank correlation between MTR and being a 5/5 missense variant is weak ($\rho=-0.133$). MTR is informed exclusively by the observed-to-expected ratio of variant sites in our dataset and captures inter-human variation while missense predictors (PolyPhen, SIFT, MutationTaster etc.) take evolutionary conservation metrics such as GERP, phyloP, and other features into account. MTR and missense prediction scores provide complementary information (intraspecies versus interspecies). Therefore, we would not expect strong correlations between evolutionary conservation-based metrics and MTR. While MTR effectively discriminates between ClinVar pathogenic and benign in the top percentile ranks of MTR values exome-wide (Fig. 3), by itself it is likely not the best predictor for pathogenicity compared to missense predictors that use evolutionary conservation as a discriminating feature. However, MTR can be a useful feature for improving missense prediction algorithms. For example, population frequency from humans and primate data has been shown to improve missense effect predictions in the recently published prediction tool AlphaMissense (Cheng et al.). The significance of MTR lies in its potential to identify missense-constrained regions within genes from the large amount of human sequencing data that we can generate now, which we demonstrate in the updated manuscript and in the MTR track view in the RGC ME browser. Identifying regions within a gene that are missense-constrained is useful even in genes that are deemed unconstrained based on gene-level metrics.

Referee 2 Comment 6	To address missense constraint, the authors could use measures presented in previous work, e.g. the missense Z-score from PMID: 25086666, which is acknowledged to be a gene-level estimate, but preferable to MTR because of the imprecision of the mutation rate model underlying MTR.
Authors' response	We updated MTR to take mutation rate into account (see Comment 5 above). As alluded to in the above comment, MTR is complementary to evolutionary conservation. Additionally, missense variation allows us to infer constraint at a higher resolution than gene level metrics, allowing us to identify regions potentially functional regions of the protein.

Referee 2 Comment 7	The reported biological processes enriched among the 849 genes with putative human-specific missense constraint ("neuronal and immune system", l. 213) do not closely reflect the actual list of results (Supp. Fig. 5A). While "neuronal system" indeed has the lowest p-value, the most significant enrichments follow for processes involved in gene expression, development, splicing and basic cellular signaling, for which the argument of species-specific selection cannot be so easily made, casting further doubt on the MTR estimates.
Authors' response	The reviewer brings up a good point. However, we would not interpret this as casting further doubt on the MTR estimates; instead, it is perhaps reflective of the limitations of gene enrichment analyses in this context. We have removed this result from the analysis.

Referee 2 Comment 8	When analyzing the fixation index, was it calculated pairwise between all subpopulations or were the subpopulations partially pooled? What exactly is shown in Supp. Fig. 7? What type of selection acts on the annotated genes? Is F_{ST} given only for LoF variants, as the plot title suggests?
Authors' response	Supplementary Table 13 reports the results of pairwise fixation index values calculated for each population compared to pooled genotype data from all other populations (e.g., AFR vs others, and vice versa). It is important to note that high F_{ST} values do not necessarily imply selection although a small proportion of high F_{ST} (> 0.15) variants are in genes that were previously reported to be targets of positive selection. Genetic drift, demographic processes, and selection can all increase the genetic differentiation between populations. We have removed Supplementary Figure 7 in this revision. Instead, we have added additional information about known loci under selection (previously reported local adaptation and citation PMID) in Supplementary Table 13.

Referee 2 Comment 9	Which variants are included in the last F_{ST} analysis, of which only a few select results are mentioned (l. 336)? Why is there no table with all enrichments?
Authors' response	This section has been removed from this submission in response to Reviewer 1's comment primarily to steer away from any kind of implied population-specific selection. While that was not our intention, we removed this section to avoid any ambiguous interpretation.

Referee 2 Comment 10	Contrary to what is stated (l. 326), differentiated alleles in AFR and EAS sub-populations shown in Supp. Table 7.3 and 7.4 do not contain morphological traits associated with skin/hair pigmentation. Bilirubin, a pigment that is produced during the breakdown of hemoglobin, is not a specific skin or hair pigment.
Authors' response	Thanks for bringing this to our attention. The tables 7.3 and 7.4 in the previous submission were restricted to differentiated variants in AFR and EAS where the minor allele frequency in EUR was less than 1% . This filter excluded certain skin pigmentation genes. We have now added an updated supplementary table (Supplementary Table 13) where all highly differentiated variants ($F_{ST} > 0.15$) are listed, and we have highlighted skin pigmentation genes along with other known targets of selection.

Referee 2 Comment 11	Conceptually, I fail to see the added benefit of combining the MAPS score of a particular variant with its splicing score. Ultimately, this just amounts to thresholding the splicing score and any associations with, for example, ClinVar pathogenicity only show that the splicing score works well. Conversely, if also CADD scores were thresholded (e.g., at 30), the difference in enrichment with ClinVar pathogenicity between thresholded CADD and SAVs would likely be minimal.
--

Authors' response	The reviewer is correct that a goal of combining MAPS scores of variant groups with splicing scores was to derive thresholds; however, another key benefit provided by the MAPS analysis was to systematically quantify the deleteriousness of potential cryptic splice variants at various score thresholds. We aimed to provide readers with a greater understanding of the relationship between variants in increasing score bins and the degree of deleteriousness that may be expected from them. Predicted splice variants identified with the SpliceAI and MMSplice MAPS-derived thresholds can be interpreted as both affecting splicing and being as deleterious as 5/5 missense variants. Thus, the thresholds can be thought of as setting biologically or clinically useful cutoffs. The enrichment of deleterious splice affecting variants among ClinVar pathogenic variants was to demonstrate that the predicted variants identified by the MAPS-derived thresholds can achieve comparable or better discrimination between pathogenic and non-pathogenic variants compared to CADD20 or 5/5 missense. This serves as one method to validate our approach. While CADD may be a good discriminator of deleteriousness, anchoring on MAPS-derived splice prediction thresholds identifies missense variants that affect splicing and provides a functional interpretation of the missense variant's effect.
-------------------	--

Referee 2 Comment 12	While the similarity between LOFTEE HC splice sites and SAV splice sites in Fig. 5B suggests that both assignments correlate similarly with pathogenicity, the nonsignificant odds ratio of enrichment with experimentally validated deleterious splice site variants (l. 397, Fig. 5C) suggests that both measures may not capture true effects. What are the odds ratios when comparing SAVs with pathogenic vs. benign ClinVar variants (same as used in Fig. 3A), rather than comparing SAVs with other deleteriousness filters? What would Fig. 5C look like for the variants annotated as "splice_site" in A?
Authors' response	We have added analysis that reports odds ratios comparing SAVs between pathogenic vs. benign ClinVar variants (Supplementary Table 9). Splice sites are now labeled in Fig. 5 and we apologize for the omission of labels in the previous submission.
Changes in text	Lines in text: 497-498 Similar results were also obtained when we evaluated the enrichment of pathogenic variants compared to benign variants (Supplementary Table 9A).

Referee 2 Comment 13	Germline mutation rates are known to vary not only between trinucleotide contexts, but also between genes, the most important factor for the latter being replication time. The downloaded gnomAD rates for trinucleotide contexts and methylation levels are not adjusted for this. In line with this, the negative MAPS scores in 5'UTR and intronic regions in Fig. 5 suggest that the mutational model is inaccurate. In particular, since 5'UTRs are enriched in CpG islands, this may suggest that the methylation correction of mutation rates is not working well. More sophisticated models of germline mutability have been in use for some time (e.g. PMID: 30218074, https://doi.org/10.1101/2022.08.20.504670 , https://doi.org/10.1101/2022.03.20.485034), so the authors should make sure to revise their mutational model to account for the effects of replication time and other covariates of mutation rate that are not accounted for in their current model.
Authors' response	We thank the reviewer for the suggestion. As recommended by the reviewer, we have updated the mutation rate model that we use for MAPS, S_{het} , and MTR to the recently described genomic-feature aware model from gnomAD (Chen et al). The collection of 13 regional genomic features included GC content, low-complexity region, short and long interspersed nuclear element, distance from the telomere and the centromere, male and female recombination rate, DNA methylation, CpG island, nucleosome density, and maternal and paternal de novo mutation cluster.

Referee 2 Comment 14	It is suggested that over 10k VUS are deleterious cryptic splice variants. Could the authors perform an analysis to support this claim? For example, are they enriched in experimentally validated SDVs compared to non-SDVs (compare Fig. 5C)?
Authors' response	Analyses and corresponding discussion about VUS in SDVs were added in the main text, updated Fig. 5C , and newly added Supplementary Fig 14B .
Changes in text	Lines in text: 508-511 "Variants of unknown significance" (VUS) in Clinvar predicted as deleterious SAVs (intersection set) were also significantly enriched in experimentally validated SDVs (Fig. 5C). Of the 563 predicted deleterious SAVs assayed in the experimental data, 346 (61.5%) were SDVs and more than half were cryptic splice sites, including 13 ClinVar VUS (Supplementary Fig. 14B).

Responses to Referee 2 Minor Comments

Overall, the manuscript does not appear to have been written with great care, and I think it would benefit from careful revision. The following is a non-exhaustive list of inconsistencies, inaccuracies and missing information:

Referee 2 Minor Comments	Overall, the manuscript does not appear to have been written with great care, and I think it would benefit from careful revision. The following is a non-exhaustive list of inconsistencies, inaccuracies and missing information
Authors' response	We thank the reviewer for the comments and appreciate the thorough review of our manuscript. Below we have addressed each point.
Changes in text	 - gnomAD version numbers swapped (l. 86)? The version numbers are correct – v2.1.1 included ~125,000 exomes and v3.1 included ~76,000 whole genomes. 3.1 was not an incremental version over gnomAD 2.1.1 as the two datasets were on different genome builds. - Two different percentages given for the fraction of mutated synonymous and stop-gained sites (l. 95 and 99) One is the proportion of observed synonymous/SG variants among sites, i.e., chromosome-position pairs; the second is the observed proportion among all variants, which refer to specific reference-alternate allele changes. Across RGC-ME, we observed 16.4M distinct mutated coding sites and 20.1M coding variants. We have clarified the difference in the text (lines 92-95). - Supp. Fig. 1: "CpG" used to denote "CpG transitions" (same for "non-CpG") - Supp. Fig. 1: "highly methylated" and "lowly methylated" not defined (comparing to classification 0, 1, 2) - Supp. Fig. 1: No y axis label in the figure (top). What is shown is not the same as the data in the table. Supplementary Fig. 1 has been updated with a clarification of CpG transitions, quantification of methylation levels, and defined axes. - Ext. Fig. 2: Variance of s_{het} is not defined. Is it computed from the posterior density? The variance of s_{het} was computed from the standard deviations of MCMC samples. This has been added to the legend.

- No definition of "underpowered" (l. 167).
 In a recent publication, gnomAD authors determined that genes with ≤ 5 expected LOFs were underpowered to detect pLOF constraint using the LOEUF metric. We have adopted their definition here and added the citation (line 216).

- "Most genes with few expected pLOF variants are indeed unconstrained" is not a valid statement. Instead, the constraint cannot be measured to be $s_{het} > 0.075$.
 Yes, good point, we have altered the language (line 218-219 in text).

- Supp. Fig. 4: LOEUF (x axis) not defined, units not explained. Fixed

- "regions"  "genes" (l. 217) Fixed

- Set of 635 "LOF-constrained genes" not defined (l. 224). Clarified in text

- Why are ClinVar variants with a rating of zero stars used in Fig. 3A, when in the entire rest of the manuscript pathogenic ClinVar variants were defined as having at least two stars?
 We have updated the manuscript to consistently use 2+ stars for all analyses.

- "pKOs are significantly less constrained than expected" is not a valid statement, as there is no expectation given. Relative statement can only be made in comparison to something (e.g. the mean).
 We have altered the text to report that pKOs have significantly lower s_{het} than all other genes (line 384 in text).

- It is not stated which criteria were used to select the variants shown in Supp. Table 7.3 and 7.4.

- Gene ATXN2 is not contained in Supp. Table 7.4, nor any other gene associated with COPD (l. 334).
 Suppl Tables 7.3-4 have been removed in response to Reviewer 1's critique.

- The "intersection set" of the two splice scores should be clearly defined at the beginning (l. 371) and consistently referred to as the "intersection set". Incorrect use of "both" (twice).
 We have added the definition of the intersection set to line 486.

- "splice sites" (l. 373) seem to be canonical splice sites as defined by LOFTEE (LC & HC). This should be clearly stated.
 We have added the definition of canonical splice site to line 477.

- Inconsistent use of splice site sets in l. 379, switching from the default intersection set (with 11% of missense+synonymous variants) to the two separate sets, which give higher percentages.
 We have updated the manuscript to consistently use the intersection set for all analyses.

- Extended Fig. 4: x axis ticks should be brackets, not thresholds. Fixed

- Different step size in Extended Fig. 3 and Methods (l. 837).
 We apologize for the typo – the step-size should have been 0.02 and it is the same for both prediction scores. We have removed the right-hand plots labeled 'proportion of filtered variants' that were defined by the number of variants filtered in each score threshold, if that was causing further confusion.

- There is a category ("mouse lethal") missing from the bar chart in Fig. S1. *Fixed*
- Formula formatting (l. 737, 813).

We have made sure to consistently italicize variables and not subscripts – we hope that addresses the concern.

- Fig. 5: Clearly name the set of missense variants predicted to be deleterious by five algorithms and use that definition consistently throughout (x axis label in A).

The five algorithms (SIFT, Polyphen2_HDIV, Polyphen2_HVAR, LRT, and MutationTaster) are listed in the Methods and that definition is used consistently throughout the text. We refer to them as 5/5 missense variants, i.e., variants described as deleterious by five out of five missense prediction algorithms.

- Fig. 5: Missing category labels in C. *Fixed*
- Fig. 5: "upstream" and "downstream" not defined. *Fixed*
- Fig. 5: CADD score not defined, not mentioned in text or caption. *Fixed*
- Fig. 5: Inconsistent labeling in caption and table for B. *Fixed*

- Why are there almost 310k variants in Fig. 5B when only ~293k SAVs were identified (l. 370)? How many intronic, upstream and downstream variants are there?

The 'All' category in Fig. 5B summarized the total number of splice affecting variants, including non-coding variants, while 'Total' values from Supp Fig. 9A (which provided the 293k value) only accounted for the functional categories of variants summarized in that table, *excluding* intronic, upstream, downstream and UTR region variants. For consistency, we also added a row for 'coding variants' in Fig. 5B. The analysis shown in Fig. 5 was performed on unique variants while the summary in Suppl Fig. 13 was performed on unique variant-transcript pairs; therefore, the total number of coding predicted deleterious splice affecting variants is slightly different in the two figures. All other comments for Fig. 5 have been fixed; CADD score is now defined in main text (line 492).

Responses to Referee 3 Major Comments

Referee 3 Comment 1	I have some concerns about the robustness of conclusions based on observations in single individuals. In a dataset as large as this, the probability of observing a certain type of artefact in at least one participant may be relatively high, even if the overall accuracy of variant and genotype calling is also reasonably high. Note that the variant false discovery rate (FDR) is estimated to be 8% (as stated on Line 615, "Precision=0.92"). The FDR is not broken down by variant type (SNV vs indel), but presumably false positives (FPs) are enriched in indels compared to SNVs. I don't think estimates of the genotyping error rates are given, nor estimates of the genotype probabilities conditional on a variant being a FP or being of a certain type (SNV or indel). However, variant calling errors and genotyping errors could, combined, lead to false variant calls such as false predicted knockouts (pKOs), especially those induced by false indel calls annotated as frameshift variants. To what extent could erroneous variant calls affect the validity of statements based on observations in single participants such as "62.6% of homozygous pLOF variants are found in one participant each" and "1,838 of the putative gene knockouts (pKO) have not been previously reported"?
Authors' response	We thank the reviewer for the insightful feedback. The precision for indels is indeed lower than SNPs in the test dataset. However, this is largely driven by an imbalance in the number of positive- and negative-labeled indels. For all indels, we estimate a false positive rate of 2.5%, which suggests that the vast majority of low-quality indels are appropriately filtered out. Additional metrics from model classification can now be found in Supplementary Table 14. To address reviewer concerns regarding homozygous pLOFs, we quantified the proportion of QC-pass variants that are likely "true" variant calls (Supplementary Table 15). Specifically, we broke SNPs and indels down into 3 categories: heterozygotes only (MAH=0), single homozygotes (MAH=1), and multiple homozygotes (MAH≥2). For single homozygotes, we estimate that 98.5% of QC-pass SNPs are "true" variant calls and 97.5% of indels are "true" variant calls. No category was estimated to have lower than 97.5% of post-QC variants to be "true" variant calls (see newly added Supplementary Table 15). This would suggest that very few homozygous pLOFs are likely to be erroneous variant calls, although we did not explicitly derive estimates for these variant classes due to the size limitations of the labeled test dataset. Additionally, we validated a subset of the indels by manual evaluation of sequencing reads on IGV and benchmarked the rare, homozygous pLOF calls against Genome In A Bottle. Finally, we identified events where two frameshift indels lead to an inframe consequence and removed 25 genes from the pKO list.

Changes in text Lines	Lines in text: 747-774 A detailed breakdown of performance for SNPs and indels (Supplementary Table 14) shows that indels have lower precision (0.853) than SNPs (0.926) in our test set. Therefore, we performed the following additional analyses to confirm the validity of pKO annotations:  1. We estimated the proportion of likely high-quality variant calls by variant type in the QC pass dataset based on the observed fraction of QC pass variants and the metrics derived from SVM test set classification (Supplementary Table 15). For all variant types, we estimate the proportion of post-QC “true” variants to be no lower than 97.5%. 2. In a benchmarking experiment, variants in seven “Genome In A Bottle” benchmark samples sequenced in-house showed an average precision of 99.6% for SNVs and 99.1% for indels for events with genotype qualities (GQ) greater than 20. The 8,577 homozygous variants have an average GQ of 23.8 and represent high quality genotype calls. 3. In addition, we also visually validated 202 homozygous frameshift indels in pKO genes unique to RGC-ME using the following criteria:  a. 161 singleton variants that are either absent in external datasets or do not pass QC in gnomAD v3.1.2 genomes and v2.1.1 exomes, and TOPMed Freeze 8. b. 17 variants where pLOF alt/alt genotypes had GQ < 10. c. 24 variants that had 1 or 2 pLOF homozygotes but fewer heterozygous carriers of the variant. 197/202 (97.5%) of the indels were validated as true positive indels upon visual inspection of sequencing reads. 4. Finally, we removed 25 genes where two frameshifts within the same gene in an individual results in an inframe event. For this purpose, we utilized “csq” from samtools to find all possible combinations of homozygous frameshift variants and any other frameshift variants within the same gene that leads to a compound inframe variant. Individuals that carried both variants in an identified pair (hom-alt/hom-alt or hom-alt/het-alt) were flagged and their contributions to the overall homozygous pLOF count for one or both variants were changed to 0. After these filtering steps, we were left with 4,686 pKOs.
Referee 3 Comment 2	I found the analysis of predicted splicing scores using recently developed methods such as SpliceAI and MMSplice compelling, in particular the use of MAPS to identify biologically/clinically useful thresholds. However, it is not ideal that variants of all types were aggregated in the score thresholding. I assume that a relatively high proportion of high-scoring variants are canonical splice site variants, which have been known for some time to have important molecular consequences (and indeed represent the consequence category with the highest mutability-adjusted proportion of singletons (MAPS) score in Fig. 5A). It would be preferable, instead, to identify the threshold required to achieve a biologically or clinically useful MAPS separately for each consequence category. For example, when considering a canonical splice site variant, a threshold of zero would give a MAPS of ~0.14, but when considering a missense variant, a higher threshold would be required to give a high MAPS, and so on. Specific thresholds for different variant consequence classes would be particularly helpful in a clinical genetic setting.
Authors’ response	We excluded essential splice sites from the main set of analysis and derived MAPS thresholds for coding non-SS variants separately. This is discussed in the main text, updated Extended Fig. 3B and the new Supplementary Fig 9.
Changes in text	Lines in text: 513-522

	To derive stringent thresholds, we removed canonical splice sites and grouped all the remaining coding non-splice site (nonSS) variants in different splice prediction score bins and derived a score threshold at which the MAPS score is $\geq 5/5$ missense. These thresholds corresponded to a SpliceAI score of 0.43 and MMSplice score of 0.97 (Extended Fig. 3B). Using these splicing prediction score thresholds for coding nonSS variants, we observed consistent and significant pathogenic enrichment results when comparing deleterious coding nonSS and missense SAVs with corresponding variant categories filtered by $CADD \geq 20$ (Supplementary Table 9B, C). Consistent results were also obtained when we compared the enrichment of deleterious SAVs in SDVs to non-SDVs after applying thresholds for coding nonSS variants (Supplementary Fig. 14A, C).
Referee 3 Comment 3	For publication in a journal of this calibre, I would have expected the inclusion of copy number variant (CNV) analysis, especially considering the availability of the SNP data generated alongside which could help control false CNV calls from the exome sequencing data.
Authors' response	Copy number variants are an important class of variation. However, calling CNVs from exome-wide has many challenges and analysis of CNV data in terms of functional significance warrants its own paper. Given the scale of our data and the challenges and complexities, this is beyond the scope of this paper.
Referee 3 Comment 4	It would be helpful to provide more detail in the main text and in methods regarding the aggregation and exclusion of participants in this collection. For example, some effort was made to exclude individuals with rare diseases (lines 561-562), but the precise methodology for doing this is not given. Was it based on ICD10 codes? If so, what rules were used to exclude participants? Although the exclusion criteria will have led to a depletion of cases with rare diseases, some patients will undoubtedly have been included anyway, with implications for the interpretation of results given throughout the manuscript (e.g., the pKO section)
Authors' response	We have added additional information about the study cohorts and collaborations to the Methods section and the RGC-ME Browser (https://rgc-research.regeneron.com/me/data-contributors). We excluded entire cohorts specifically designed to recruit participants with Mendelian disease. This was to ensure the dataset was not enriched with patients carrying rare disease phenotypes. No other participant exclusion criteria were used. The resulting dataset is not presumed to be "healthy," but rather represents a cross-section of a normal population sample and reflects population prevalence.
Changes in text	Lines in text: 680-686 Samples were removed based on sequencing metrics including sequenced gender not matching the gender listed in the manifest, contamination, low coverage, and unresolved duplications. To ensure this sample set characterizes genetic variation representative of the general population, we excluded available samples from cohorts specifically enrolling participants with Mendelian diseases, neurodevelopmental disorders, blood cancers, and use-restrictions inconsistent with the analysis presented here.

Referee 3 Comment 5.1	The clinical genetics section is the least compelling, especially considering the absence of any phenotype data on the participants carrying the "pathogenic" variants. Below are a few comments that may help improve it, assuming phenotype data cannot be brought to bear on this analysis. - The first paragraph concerns a look-up of ClinVar variants in the exome data and includes the statement "Each individual harbors on average 1.6 pathogenic variants, presumably as heterozygous carriers for recessive diseases." I believe it is not necessary to presume, because the authors could look up the genotypes for these variants and obtain the mode(s) of inheritance (MOI) of the relevant genes in a gene panel database, such as PanelApp. (The OMIM data appear to have MOI data for only 280 autosomal dominant genes and 1,087 autosomal recessive genes, which is a small fraction of the number of known aetiological genes for rare diseases, so that resource is not ideal for this purpose.)
Authors' response	The 280 AD and 1,087 AR genes included only those with variants observed in the dataset – the total number of AR and AD genes in OMIM is larger. The manuscript has been updated to clarify this point. In addition, we added PanelApp genes as suggested by the Referee (unambiguous for AR and AD with OMIM) to the list. This is an excellent point about obtaining the mode of inheritance for the genes that comprise the average 1.6 pathogenic allele load. We added the observation that 0.98 of this average count is attributable to heterozygous carriers of AR genes. In addition, we added the percent of the population that are carriers (homozygous in the AR case) for all other genes with known etiology that are not in ACMG.
Changes in text	Lines in text: 531-536 Each individual harbors on average 1.58 pathogenic variant sites, the majority comprising heterozygous carriers of pathogenic variants. Specifically, 61.4% of the 822k unrelated individuals were heterozygous carriers of pathogenic recessive alleles in 1,143 out of 2,659 known autosomal recessive (AR) genes, with an average of 0.98 pathogenic alleles per person. 0.21% of the unrelated samples were homozygotes of pathogenic variants in 167 AR genes; 3.64% were heterozygous carriers of 353 AD genes out of 1,629 total. Lines in text: 556-558 Focusing on non-ACMG genes in the 822K unrelated samples, we find that 1.27% of individuals were heterozygous carriers of pathogenic variants in AD genes and 0.21% were homozygotes of pathogenic variants in AR genes.
Referee 3 Comment 5.2	- The second paragraph restricts the ClinVar look-up to 76 autosomal genes in the ACMG Secondary Findings (SF) list. Here, the statement "2.38% of RGC-ME carry an actionable variant" considers variants responsible for dominant and recessive disorders together. It would be interesting to report what proportion of participants have an allelic configuration that, in principle, increases their risk of disease (i.e., with a P/LP variant on both alleles in the case of recessive disorders). The percentage of at-risk individuals (2.38%) seems high, given this concerns only 76 genes, so it would be interesting to provide the corresponding percentage for all known aetiological genes for comparison, bearing in mind that only a few percent of individuals in general are thought to have a rare disease.
Authors' response	The percent of the population that are homozygotes of pathogenic variation in AR genes has been added in this submission (addressed in previous comment). For carriers of actionable variants in ACMG genes, we have clarified that only individuals with homozygous pathogenic alleles were

	counted toward the total. The percent of at-risk individuals is consistent with findings reported in other studies (reference in text below).
Changes in text	Lines 541-554 Among the 822K unrelated individuals, 22,846 (2.77%) had at least one ClinVar-reported (≥ 2 stars) pathogenic missense or pLOF variant for 72 out of the 76 autosomal genes on the ACMG list (Supplementary Table 11; for AR genes, only individuals with homozygous pathogenic variants were counted toward this and subsequent pathogenic totals). We also tallied carriers of likely-pathogenic (LP) pLOF variants (novel variants not yet reported as pathogenic in ClinVar) in 44 genes where truncation is known to lead to disease. 2,357 (0.3%) individuals in RGC-ME carried 1,407 LP variants across 40 out of 44 of these genes. In total, 3.06% of RGC-ME were carriers of pathogenic or LP variants. Excluding individuals with high frequency pathogenic variants in the HFE (Cys282Tyr) and TTR (Val142Ile) genes, 2.38% of RGC-ME carried an actionable variant (Supplementary Table 11). This number is comparable to other reports of actionable variants ranging from 2% to 4.1% (Dewey et al., van Hout et al., Halldorsson et al. for gene sets that include ACMG 2.0 and ACMG 3.0.
Referee 3 Comment 5.3	- In the third paragraph, the absolute numbers given in the comparison of ancestry groups after sub-sampling are hard to interpret, because they depend on the fact that the smallest ancestry group (SAS) happens to have ~30k participants, which is arbitrary. For this reason, the text might be more interpretable if it places greater emphasis on percentages and ratios. Lines 455 to 464 in particular are hard to parse and interpret, because variants can have different allele frequencies and may be observed in multiple individuals of the same or different ancestries (and furthermore, ancestry is defined according to a 50% assignment threshold, which might not necessarily encompass the haplotype harbouring the variant being considered - an issue that deserves mentioning). In my opinion, it would be preferable to focus on the per-individual results. They show that the distributions of the per-individual number of ClinVar pathogenic variants across the ancestry groups are remarkably similar, with median counts equal to 7 for AFR and SAS and 8 for EUR and IAM (Supp. Fig. 11, right panel). I would elaborate on this result, because, in the context of that easily-missed supplementary figure, the statement "individuals of European ancestry have a median of 18.2% more known pathogenic variants than individuals of African ancestry" seems selective and a bit misleading. (It also seems at odds with the $8/7 = 14.29\%$ increase implied by the figure, which, in addition, ought to include statistical testing results). The higher number of VUSs amongst individuals of AFR ancestry is to be expected because AFR individuals have more variants in general due to greater genetic diversity. For these reasons, it seems unclear whether the observations can or indeed need to be attributed to "ascertainment bias" and limited "access to genetic medicine."
Authors' response	We subset to ~30k individuals to compare ancestry specific AAF while minimizing the bias of sample size. Rare variants are better captured in populations with larger samples sizes and to make a fair comparison we compare ancestry specific AAF differences in equal sub-samples. Nevertheless, this is a good point that the ancestry specific AAF from sub-sampling at a variant may not reflect the haplotype because we sample from individuals with >50% ancestry probability. As such, we have updated this section to crosscheck the sub-sample AAF with proportional variant-level AAF that were computed with all 822k unrelated samples and reflect ancestral local genomic segments. We find that the local AAF differences corroborate our observations from sub-sampling (updated Extended Fig. 4). The per-individual results show that Europeans have on average of 63% more well-characterized (2+) pathogenic variants than individuals of African ancestry; conversely, Europeans have 25.6% fewer VUS and 18.6% fewer variants across all functional types (new Supplementary Fig. 17). The 7 and 8 median counts for AFR/SAS and EUR/IAM in Supplementary Fig. 16 (formerly Suppl. Fig 11), respectively, were only for ClinVar 0+.

	We observe a statistically significant inverse relationship between the evidence level of pathogenicity (2+ having the highest confidence, VUS the least) and the level of European enrichment. We attribute this to the overrepresentation of individuals with European ancestry in clinical sequencing – this may be due to historical lag in individuals getting genetic testing for disorders and access to genetic medicine, or any number of interrelated factors. This has been clarified and expanded upon in the manuscript.
Changes in text	Lines in text: 572-596 In addition to ancestry-specific AAF derived from individuals assigned hard-coded ancestries in sub-sampled groups, we also computed proportional haplotype-based AAF from the 822k unrelated dataset to verify that the AAF reflects local ancestry specific to the genomic segment. Variants that were $\geq 100x$ more common in non-EUR sub-samples had average local AAF=0.012 (AFR) and 0.0047 (SAS), compared to the EUR local AAF average of 9.51×10^{-5}. In contrast, 1% of VUS found in non-EUR populations (4,009 variants, $sd=30.2$) had ≥ 100-fold AAF compared to EUR on average (mean local non-EUR AAF=0.0062 compared to EUR=9.25×10^{-5}). In AFR, over 500 and 700 of VUS were in AF bins 1%-5% and 0.5%-1%, respectively (mean local AAF were 1.86% and 0.671%; Extended Fig. 4A). Approximately 34% of unique pathogenic coding variants per sub-sample were only observed in non-EUR individuals, indicating that sampling diverse populations is necessary for comprehensive identification of rare variation. We computed per-sample counts of variants using all unrelated individuals with continental ancestry probability $>50\%$ (Supplementary Table 1). Per sample, EUR had on average 63% more well-characterized (2+) pathogenic variants than AFR. Conversely, EUR had 25.6% fewer VUS (Supplementary Fig. 16) and 18.6% fewer variants across all functional types (Supplementary Fig. 2). In AFR, a consistent pattern of significantly fewer high confidence (2+) pathogenic variants (-0.576 [-0.567, -0.585], t-test) to a surplus of VUS (42.13 [421.97, 42.28]) all variant effect types compared with EUR suggests that the most well-characterized and studied pathogenic variants are depleted in this population (Supplementary Fig. 17).

Referee 3 Comment 5.4	- Regarding the fifth paragraph, it seems plausible that some VUSs in ClinVar will have a higher allele frequency (AF) in non-Europeans simply because accurate AFs in the relevant non-European populations were not available at the time of classification. Can this be assessed? I don't think a pathogenic classification would ever have been assigned to a variant with knowledge that its AF was $>1\%$ in certain populations.
Authors' response	It may be the case that accurate AFs in the relevant non-European populations were lacking at the time of classification, but we do not have a systematic way to investigate the historical knowledge available. This is an argument in favor of regular reanalysis and updating of ClinVar assessments.

Referee 3 Comment 5.5	- In the sixth paragraph, the claim that variants in regions of greater human constraint are "likely pathogenic" seems overstated, and the term "likely pathogenic" should be reserved to ACMG criteria to avoid confusion. Reclassifying VUSs is a tremendous challenge. If the authors think the constraint metrics and splice prediction score thresholds provided could be used to reclassify VUSs in significant numbers in practice, with or without additional experimental work, it would be helpful for this vision to be described more fully.
Authors' response	We agree that the term "likely pathogenic" should be reserved to ACMG criteria. We have updated the manuscript to reflect the change. We have also simplified the paragraph to clarify the utility of MTR scores and predicted cryptic splice sites for improving variant interpretation and prioritization efforts.

Changes in text	Lines in text: 600-607 Although VUS have less empirical evidence for pathogenicity, they comprise the bulk of ClinVar with over 1 million variants. Notably, VUS in regions of low MTR may be deleterious, comprising 5,868 (0.68%) VUS in the top 1 percentile MTR constrained regions and 95,476 VUS (11%) in the top 15 percentile (Supplementary Table 12). Using the MAPS-derived splicing score thresholds, we also identified over 11,000 candidate cryptic splice variants among VUS (1,366 synonymous variants in 822 genes and 10,407 missense variants in 3,501 genes), offering potential insights into their functional consequences for clinical prioritization and interpretation efforts.
-----------------	--

Responses to Referee 3 Minor Comments

Referee 3 Comment 1	- The "diversity" of the collection is repeatedly touted in the first part of the manuscript. However, only 18% of participants are non-European (Supp. Table 1), so it is not very diverse. The percentage is about the same as genetic studies with distributions approximately aligned with the UK census (e.g., https://doi.org/10.1038/s41586-020-2434-2) and significantly less than gnomAD v3 (https://gnomad.broadinstitute.org/help/what-populations-are-represented-in-the-gnomad-data), 47% of participants of which are non-European. The strength of the RGC-ME dataset is not the diversity (i.e., the heterogeneity in ancestry groups), but the absolute number of individuals from different non-European ancestries who are included (amounting to 179,560 individuals, including relatives). Perhaps the authors could amend the text to reflect this reality more faithfully. In addition, the statement "exome sequencing of 985,830 individuals of diverse ancestry" might be more clearly rewritten as "exome sequencing of 985,830 individuals across a diverse array of ancestries," as diversity is a property of the collection rather than the individual. It is also not, strictly speaking, a property of data ("ancestrally diverse exonic data" in line 71).
Authors' response	We have updated the number of non-EUR – now 191,362 unrelated individuals, 23% of the dataset – in our count for RGC-ME to be more consistent with external datasets and include individuals of unassigned ancestry. (Individuals of Ashkenazi Jewish, Amish, and Finnish ancestry were classified in EUR). In addition, we have carefully considered the characterization of diversity in RGC-ME for more consistency and accuracy throughout the text.

Referee 3 Comment 2	Line 193: "Enrichment in pathogenic missense variants is significant for all regions where MTR != 1..." I didn't quite follow this. MTR=1 only when the observed fraction is exactly equal to the expected fraction. Why would there be enrichment for pathogenic missense variants when MTR<1 as well as when MTR>1? Could the authors include P-values wherever enrichment is referred to (here and elsewhere)? Here, it would also be helpful to show the distribution of MTR for ClinVar pathogenic variants.
Authors' response	We have updated the text to reflect our observations of significant enrichment for MTR that are significantly < 1 and are in the top 20-percentile of scores exome-wide. 95% confidence intervals around the odds ratios demonstrate significance. We show the distribution of MTR for ClinVar benign and pathogenic variants in the updated Fig. 3B.
Changes in text	Lines 246-249

	Compared with benign missense variants, pathogenic missense variants annotated in ClinVar (≥ 2 stars) were highly enriched in the top exome-wide percentile of MTR constrained regions (odds ratio = 100.0 and 65.1, computed with 21- and 31- codons, respectively; Fig. 3A). Missense variants ranking in top MTR scores up to the 20-percentile were predictive for being pathogenic.
--	--

Referee 3 Comment 3	- Fig. 3: shouldn't "rank" be "percentile"? Fixed - The language is ambiguous in certain sections. E.g., I assume the "pKOs" in "pKOs are significantly less constrained" refers to genes for which at least one individual is homozygous for a pLOF variant. (A pKO is an individual-and-gene-specific observation rather than a gene characteristic.) Correct. We use the term pKOs to describe genes that have at least one rare, biallelic pLOF variant (either a homozygous allele or compound heterozygote). Once we observe such a variant, we can think of "pKO" as a gene annotation. - Lines 304 to 311 presuppose that the participants with pKOs do not suffer clinical consequences as a result of these pKOs. How confident can one be about this? Please refer back to point (4). We do not presume all samples in the population are healthy, only that they represent a cross-section of the overall population. See above answer to Comment 4. It is important to phenotype the individual carrying the homozygous pLOF variant to better understand the effect of the loss-of-function of that gene. We have captured this sentiment in the first submission: "In-depth phenotyping of human knockouts can help researchers better understand the efficacy and side-effect profiles of these potential drug targets" (lines 433-435 in text). - Line 336 contains a stray ")" Fixed - Line 352: The sentence "Here, we use human genetic data to define splice prediction thresholds that identify deleterious variants that affect splicing" might be more clearly stated as "Here, we use human genetic data to optimise splice prediction score thresholds that enrich for deleterious variants that affect splicing." Thank you for the suggestion. The sentence has been rewritten (lines 444-445). - Fig. 5a: what do the colours refer to? This is not stated in the legend. The colors form groupings of variant effect classes and are not necessarily meaningful but reflect our prior belief in each groups' deleteriousness. - Fig. 5c: what is shown along the y-axis? Labels have been added. - Line 422: "previously unobserved" is not quite right because a variant had to be observed previously to end up in ClinVar This is a good point, and we have fixed the sentence (line 530). - Line 599: "transmitted singletons" - if a variant is known to be transmitted, presumably that's because it was seen in multiple related individuals, but in that case the variant would not be a singleton according to the usual definition. Transmitted singletons are alleles present only twice in the full dataset (once in a parent and once in a child) and are identified from trios in our data. We use such variants in the model training data as true positive rare variants. - Line 720: "These sites will experience fewer mutations than expected" doesn't seem correct: the sites do not experience fewer mutations, mutant alleles are selected against. Fixed - Extended Fig. 1: the vertical line denoting a threshold is misleading given that the x-axis is not
---

continuous (each set of horizontally arranged bars along the x-axis corresponds to a specific value of s_{het}). We have removed the line.

- Extended Fig. 3: I can't work out why there are horizontal confidence intervals
For each set of variants filtered by a given threshold, we can calculate the MAPS score and corresponding standard error of the mean of the proportion of singletons.

- Extended Fig. 4 legend: "~500 variants have" should read "~500 VUSs have" Fixed

- Supp. Fig. 1 y-axis unlabelled Fixed

Figure R1. Visualizing the difference in results based on incorporating mutation rates into the MTR calculation. MTR was computed using (Y-axis) all possible missense and synonymous in the denominator, versus (X-axis) an updated formation that estimates expected missense and synonymous using mutation rates based on local sequence context.

- A. Comparison of direct MTR values
- B. Comparison on a gene level for the proportion of each gene in the top 1%, 5%, and 10% of exome-wide MTR percentiles.

Reviewer Reports on the First Revision:

Referees' comments:

Referee #1 (Remarks to the Author):

The revised manuscript is greatly improved, and most of my comments have been fully addressed. I only have one remaining comment.

1. I am still not convinced that the section on pKOs demonstrates a depletion of homozygous pLoFs vs. the neutral expectation. The revised manuscript reports the median observed/expected ratio (where "expected" is correctly computed) as well as the fraction of constrained vs. unconstrained genes with observed/expected $< \frac{1}{2}$. The issue with this analysis is that for many genes, and presumably for most constrained genes, the expected count is $\ll 1$. If expected were 0.1 for every gene, then even under the null, observed/expected would be 0 for ~90% of genes, and in particular the median would be 0. This is easily addressed by taking the mean instead of the median. I suggest not reporting the median ratio (which is definitely misleading) and instead reporting the mean, or even better the ratio of means (i.e., mean observed over mean expected). (It might be necessary to remove consanguinity-enriched groups before doing this.)

Referee #2 (Remarks to the Author):

The authors have responded appropriately to most of my comments. Some responses remain:

Regarding Main Comment 1, it is surprising to see that almost half of the dataset (N=454,787) consists of UK Biobank (UKB) exomes, which have already been thoroughly analyzed by the same authors in Ref. 5 (Backman et al., 2021) and that this information was omitted in the first submission.

I think that the corresponding paragraph that has now been added to the Methods section (which was not highlighted correctly, as almost the entire paragraph is new) still lacks some clarity, as the numbers of individuals in the different datasets are not given. I think it would be very useful to include a table with this information. Of note, the UKB exome sequencing dataset has also recently been added to the gnomAD browser (v4).

With respect to Main Comment 6 and related to Main Comment 2 of Referee 1, in their comparison with previous measures of missense constraint the authors use the number of constrained regions identified by each of the two compared measures (MTR vs. MPC). Specifically, they state that MTR identified 8 times more regions than MPC and that the regions not identified by MPC were "misclassified" as unconstrained. In my opinion, this statement cannot reasonably be made in the absence of ground truth data that would allow a quantitative assessment based on false positive and false negative predictions. Furthermore, the statement seems to derive in large part from regions that barely meet the relatively arbitrary threshold of 15% (MTR $<$ 0.841), as the average "mean MTR value of the constrained region" across all genes in Supp Table S4 is \sim 0.78.

Minor: How can it be explained that there are genes with a "mean restricted region MTR value" > 0.841 if restricted regions were defined as "continuous segments within a protein containing variants with MTR values in the top 15-percentile threshold"? In addition, the transcript IDs in column 2 of Table S4 appear to be incomplete.

Referee #3 (Remarks to the Author):

I have no major comments, and only the following minor remarks:

- The article is now quite wordy. Certain passages are technical to a degree that might not be appropriate for a general scientific audience). In my opinion, R2's previous comment that the paper would benefit from careful revision still applies (see examples below).

- Related to the above, the article describes numerous detailed technical comparisons of quantitative metrics. Some of these comparisons rely on arbitrary thresholds that could in principle be tweaked to obtain a desired result. I suspect there is a better, more elegant, and statistically grounded way to compare multiple scores with respect to an outcome (e.g., by modelling the variance explained by each score using a regression model). For example, the MTR vs MPC comparison uses thresholds to partition regions into constrained and unconstrained regions, and on that basis states that MPC would "misclassify" 168,815 (39.4%) of the 421,012 MTR-constrained codons. One could just as well state that MTR would misclassify MPC-constrained codons. A comparison between the two metrics in relation to a third objective outcome would be in order here.

- Clearly state whether a bar shows a confidence interval or the standard error of the mean throughout

Examples of imprecise or incorrect language:

- "genomic positions that can lead to synonymous [...] changes" - positions do not lead to changes, variants do

- "all possible pLOF variations within a gene" - all possible variants, not variations

Author Rebuttals to First Revision:

Updates to the manuscript

We thank all the referees for their insightful comments. We have updated the manuscript with additional analyses, figures, and tables. **Please note new text that has been added is highlighted in yellow both in the main manuscript as well as Supplementary Information.** Text that has been moved from the main document to the **Supplementary Information** is highlighted in grey. Changes with respect to the previous submission are as follows:

1. We have moved some technical sections into Supplementary Information (SI)

Line numbers (previous submission)	Content moved to Supplementary Information	Line numbers in SI (in revision)
Lines 169-210	Detailed comparisons between different gene constraint metrics	Lines 315-348
Lines 409-427	Discussion of pKOs and constraint in specific gene families	Lines 556-572
Lines 560-584	Comparisons of per-ancestry pathogenic variant counts containing technical explanation of local ancestry	Lines 602-648
Lines 609-624	Discussion of allele differentiation fits in with description of methods for comparison between different ancestries	Lines 650-668

2. Changes to supplementary figures and tables are described below

Table/figure	Description	In response to
New Supplementary Table 1C	Number of individuals in each dataset comprising RGC-ME	Referee 2, Comment 1
Updated Supplementary Table 4	Updated MTR regions with simplified segmentation method	Referee 2 Comment 3
Updated Supplementary Table 5	Updated Jaccard Index calculation with the updated MTR constrained regions	Referee 2 Comment 3
Updated Supplementary Figure 9	Updated MTR regions with simplified segmentation method	Referee 2 Comment 3
New Supplementary Figure 10	Venn diagram showing the overlap of constrained MTR regions on MPC defined constrained regions. Updated Figure 11B using updated MTR regions based on simplified MTR segmentation method and complete constrained MPC-segments dataset	Referee 2 Comment 2, Referee 3 Comment 2
New Supplementary Figure 11	Odds ratios comparing enrichment of pathogenic versus benign ClinVar variants and de novo variants in cases versus controls in MTR constrained regions & corresponding data table	Referee 2 Comment 2, Referee 3 Comment 2
New Supplementary Table 15	Expected and observed homozygous and heterozygous carriers of doubleton variants	Referee 1, Comment 1

3. We caught an error in the FDR calculation in MTR, affecting the count of variants that we deem significant at $FDR < 0.1$, but not the MTR values themselves. The revised Supplementary Table 3 has been uploaded here: <https://doi.org/10.6084/m9.figshare.24587328>. Additional figures/tables that have been updated include Fig. 3A and Supplementary Table 12.

Responses to Referee 1 Comments

Referee 1 Comment 1	I am still not convinced that the section on pKOs demonstrates a depletion of homozygous pLOFs vs. the neutral expectation. The revised manuscript reports the median observed/expected ratio (where “expected” is correctly computed) as well as the fraction of constrained vs. unconstrained genes with observed/expected < ½. The issue with this analysis is that for many genes, and presumably for most constrained genes, the expected count is <<1. If expected were 0.1 for every gene, then even under the null, observed/expected would be 0 for ~90% of genes, and in particular the median would be 0. This is easily addressed by taking the mean instead of the median. I suggest not reporting the median ratio (which is definitely misleading) and instead reporting the mean, or even better the ratio of means (i.e., mean observed over mean expected). (It might be necessary to remove consanguinity-enriched groups before doing this.)
Authors’ response	We appreciate the reviewer’s careful and thoughtful feedback. After much reflection, we have come up with a summary that we hope directly addresses evidence for depletion of homozygous pLOF variants relative to the “neutral” expectation. There are challenges to consider in answering this question. Compared to simple Hardy-Weinberg expectations, our data includes many more pLOF homozygotes than would be expected – generally, this is due to a combination of background inbreeding and population structure, both of which can increase homozygosity. To address the issue, we focused on the rarest variants for which homozygotes could be observed: doubletons (that is, variants with exactly two alleles in our sample). Each of these variants can be observed in either two heterozygotes or in a single homozygote. In a sample of N individuals (in our case, N = 821,979), the Hardy-Weinberg expectation is that a homozygote carrier would be observed for a small proportion of cases ($\sim 1 / (2N - 1)$) and that two heterozygotes would be observed the remainder of the time ($\sim (2N-2) / (2N - 1)$). There are 1,580,917 doubleton missense variants, and 5,857 of those include one homozygote. This is in an increase of >6,000x fold in the number of homozygotes compared to HWE expectations and corresponds to an inbreeding coefficient of ~0.370%. A similar pattern is seen for synonymous variants, with 679,335 doubletons, 2,490 homozygotes among them, and an estimated inbreeding coefficient of 0.366%. Among pLOFs, we see 129,405 doubleton variants, 406 homozygotes and estimate a smaller inbreeding coefficient of only 0.314%. This corresponds to a depletion of 15% of pLOF homozygotes due to natural selection. This account explains both the increased number of pLOF homozygotes relative to simple Hardy-Weinberg expectations and shows that, compared to other types of variants, we see fewer homozygotes for the rarest pLOFs in our data. To make sure that our results are not skewed by cohorts with high rates of relatedness, we excluded the Mexico City Prospective Study and The Center for Non-Communicable Diseases and performed the above analysis in a subset of the data that includes five population cohorts in RGC-ME (UKB, Geisinger Health System, Mt. Sinai BioMe BioBank, Penn Medicine BioBank, and Dallas Heart Study, as shown in Supplementary Table 15 below). With this subset, the average inbreeding coefficient is approximately 0.25% for missense and synonymous homozygous variants, and 0.220% for pLOFs, which corresponds to a 13% reduction of pLOF homozygotes.
Changes in text	We have added a short account of the depletion of pLOF homozygotes among very rare doubleton variants, relative to calibrated expectations from missense and synonymous variants, to the main text and supplementary information (lines 328-338 [Main Manuscript], 507-554 [Supplementary Information]).

Main text: Among very rare doubleton variants (for which we observed exactly 2 copies of the alternate allele), we observed a clear excess of homozygotes that is likely explained by population structure and background inbreeding. For example, among missense and synonymous variants we observed 5,857 and 2,490 homozygotes among 1,580,917 and 679,335 doubleton variants, respectively, compared to a Hardy-Weinberg expectation of <1 homozygote in each case (Supplementary Information). The observed values correspond to a background inbreeding coefficient of 0.37%. Among pLOF variants, we observed only 406 homozygotes among 129,405 doubleton variants. While these numbers are much larger than HWE expectations, they are ~15% smaller than the expected number of 479 homozygotes derived based on an inbreeding coefficient of 0.37%. This suggests that a notable proportion of these homozygotes were never observed in our sample population.

Supplementary Information: We compared expected heterozygotes and homozygotes among the rarest variants for which homozygotes could be observed: doubletons (that is, variants with exactly two alleles in our sample, median allele frequency= 1.217×10^{-6}). The Hardy-Weinberg equilibrium (HWE) expectation is that a heterozygote would result with probability $\sim (2N-2)/(2N-1)$ and a homozygote with probability $\sim 1/(2N-1)$, where N is the total sample size. We observe higher than expected counts of homozygotes in the complete RGC-ME dataset of 822K unrelated individuals as well as in 626K individuals from 5 large population cohorts [UKB, Geisinger Biobank, Sinai BioMe Biobank, Penn Medicine Biobank, Dallas Heart Study] (Supplementary Table 15).

The increased number of missense and synonymous variant carriers suggested a background inbreeding coefficient of ~0.25–0.37%, depending on the data set analyzed. The inbreeding coefficient, F , is calculated with respect to the ratio of observed and expected heterozygotes: $F = 1 - \text{obs}(\text{hets})/\text{exp}(\text{hets})$. This resulted in thousands of doubleton variants observed as homozygotes, when only a single homozygote is expected. When we used the inbreeding coefficients estimated from doubleton missense and synonymous variants to compare pLOF variation, we found an average depletion of ~14% in the number of expected homozygotes, suggesting that ~14% of pLOF homozygotes were removed from the population before they could be observed in our study. Taking the inbreeding coefficient, F , into account, the number of pLOF homozygotes expected was 479 and 260, respectively, for the 822K and 626K datasets. These expected counts were ~14% higher than our observed counts of 406 and 228, respectively. The frequency of homozygotes with respect to F can be computed as: $\text{Pr}(\text{homAA}_F) = F * p + (1 - F) * p^2$, where p = Alternate Allele Freq. Across doubleton pLOFs, the expected homozygous carriers is the sum of these frequencies times sample size, i.e., $\sum \text{Pr}(\text{homAA}_F) * N$. Even this reduced number of homozygotes was still much higher than would be expected under HWE.

Supplementary Table 15: Expected and observed homozygous and heterozygous carriers of doubleton variants

POP	VAR	# SAMPLES	# VARIANTS	OBSERVED HOM	OBSERVED HET	P(HOM)	P(HET)	EXPECTED HOM	EXPECTED HET	O/E RATIO HOM	O/E RATIO HET	F
1	MIS	821979	1580917	5857	1575060	6.08E-07	1.000	0.962	1580916.0	6090.396	0.996	0.37
1	SYN	821979	679335	2490	676845	6.08E-07	1.000	0.413	679334.6	6025.536	0.996	0.36
1	LOF	821979	129405	406	128999	6.08E-07	1.000	0.079	129404.9	5157.608	0.997	0.31
2	MIS	626412	1296620	3321	1293299	7.98E-07	1.000	1.035	1296619.0	3208.745	0.997	0.25

2	SYN	626412	565146	1395	563751	7.98E-07	1.000	0.451	565145.5	3092.384	0.998	0.24
2	LOF	626412	103775	228	103547	7.98E-07	1.000	0.083	103774.9	2752.422	0.998	0.22

Column descriptions

Pop = different tested populations: **(1)** 822k unrelated RGC ME dataset, **(2)** Data from 626k unrelated individuals from 5 large population cohorts [UKB, Geisinger Biobank, Sinai BioMe Biobank, Penn Medicine Biobank, Dallas Heart Study]

Var = variant effect/consequence: missense (MIS), synonymous (SYN), and pLOF (LOF)

samples = N , max number of samples sequenced across variants

Obs homs and hets = observed homozygotes and heterozygotes

$p(\text{hom}) = 1/(2N-1)$, probability of homozygote

$p(\text{het}) = (2N-2)/(2N-1)$, probability of heterozygote

Expected homs and hets = expected number of homozygotes and heterozygotes, i.e., $\sum p(\text{hom})$ over all variants, etc.

O/E ratio hom and het = observed homs / expected homs, etc.

$F = 1 - [\text{obs}(\text{hets})/\text{exp}(\text{hets})]$, inbreeding coefficient

Taking F into account, the frequency of homozygotes $\text{Pr}(\text{homAA}_F) = F * p + (1 - F) * p^2$,

where p = Alternate Allele Freq, F = inbreeding coefficient

Across doubleton pLOFs, expected homozygous carriers = $\sum \text{Pr}(\text{homAA}_F) * N$

Responses to Referee 2 Comments

Referee 2Comment 1	Regarding Main Comment 1, it is surprising to see that almost half of the dataset ($N=454,787$) consists of UK Biobank (UKB) exomes, which have already been thoroughly analyzed by the same authors in Ref. 5 (Backman et al., 2021) and that this information was omitted in the first submission. I think that the corresponding paragraph that has now been added to the Methods section (which was not highlighted correctly, as almost the entire paragraph is new) still lacks some clarity, as the numbers of individuals in the different datasets are not given. I think it would be
---

	very useful to include a table with this information. Of note, the UKB exome sequencing dataset has also recently been added to the gnomAD browser (v4).
Authors' response	We have included the number of individuals in each dataset in Supplementary Table 1C

Referee 2 Comment 2	With respect to Main Comment 6 and related to Main Comment 2 of Referee 1, in their comparison with previous measures of missense constraint the authors use the number of constrained regions identified by each of the two compared measures (MTR vs. MPC). Specifically, they state that MTR identified 8 times more regions than MPC and that the regions not identified by MPC were "misclassified" as unconstrained. In my opinion, this statement cannot reasonably be made in the absence of ground truth data that would allow a quantitative assessment based on false positive and false negative predictions. Furthermore, the statement seems to derive in large part from regions that barely meet the relatively arbitrary threshold of 15% ($MTR < 0.841$), as the average "mean MTR value of the constrained region" across all genes in Supp Table S4 is ~ 0.78.
Authors' response	We agree with the reviewer that our results lack the basis for stating that MPC-segments "misclassified" regions as unconstrained and we have removed this from the text. Regarding the MTR threshold, we would like to clarify that the MTR threshold of the exome-wide top 15-percentile was chosen based on empirically derived MAPS scores that indicate that a missense variant with $MTR \leq 0.841$ (corresponding to the observed MTR 15-percentile) is under selection comparable to missense variants predicted to be deleterious by 5 out of 5 prediction methods (Supplementary Figure 8 in the previous submission). In terms of assessing outcomes alongside ground truth, we have demonstrated that among constrained MTR regions defined by the exome-wide MTR 15-percentile threshold, pathogenic ClinVar variants are enriched compared with benign and case de novo variants are enriched compared with control in our previous submission. We have now clarified this further by adding a new figure, Supplementary Fig. 11. This pattern of enrichment of pathogenic versus benign ClinVar variants and of de novo variants in cases versus controls extends to the top 40-percentile (Supplementary Fig. 11A, 11B). While the mean MTR for constrained regions is 0.752 and relatively close to the 15-percentile threshold, we also demonstrate that the value of MTR scores for identifying functional variants and regions is robust over a wide range of MTR thresholds. In addition, we observed enriched overlap of MTR constrained regions in protein domains, including ubiquitin-conjugating core domains and DNA-binding regions (Supplementary Table 5). These results support our conclusion that MTR can be valuable for identifying functionally important regions and variants. We also made an update to the MPC data. During this revision, we realized that we overlooked the total number of constrained MPC regions that were described in two different supplementary tables (Supplementary Tables 1 and 4) of Samocha et al. We have now revised our comparison to incorporate the full extent of their data, which covers approximately 18,000 genes (1,789 are completely constrained, 2,700 had distinct constrained segments, and the rest are entirely unconstrained). With this update, the total number of MTR constrained regions is 8.59 times larger than that of constrained MPC-segments, which is similar to our earlier estimate of 8 times.
Changes in text	Lines in text: 229-241, 266-272 (Main Manuscript)

	MTR is valuable as a metric of regional constraint that may capture functionally important segments within genes. We defined MTR constrained regions as continuous segments within a protein containing variants with MTR in the top 15-percentile threshold (Supplementary Information, Supplementary Fig. 8). Constrained MTR regions overlap with results from a previous report (Supplementary Fig. 10A) that estimated regional missense constraint from ~60k ExAC samples based on observed-to-expected missense ratio (γ) and derived a composite missense deleterious score called MPC¹. We refer to the MPC regional constraint as MPC-segments and benchmark MTR constrained regions against them. MTR constrained regions have a median length of 22 residues [14-35, Q1-Q3] compared to 358 [208, 579] in MPC-segments. Overall, we identified 8.59-fold more constrained MTR regions compared to MPC-segments ($\gamma \leq 0.612$, top 15-percentile) across 2,832 transcripts that had data for both (Supplementary Fig. 10B). We examined the distribution of de novo missense variants in MTR constrained regions and observed a significant enrichment ($p=2.61 \times 10^{-10}$) in individuals with neurodevelopmental disorders (Supplementary Fig. 11A, B). Case variants are 1.85 times [1.50 - 2.31, 95% CI] more likely to occur in constrained regions compared to controls. As expected, well-supported (≥ 2 star) ClinVar pathogenic missense variants are also highly enriched ($p \approx 0$) in MTR constrained regions. Pathogenic variants are 8.82 times [8.17 - 9.53, 95% CI] more likely to occur in missense constrained regions than benign variants.
--	--

Referee 2 Comment 3	How can it be explained that there are genes with a "mean restricted region MTR value" > 0.841 if restricted regions were defined as "continuous segments within a protein containing variants with MTR values in the top 15-percentile threshold"? In addition, the transcript IDs in column 2 of Table S4 appear to be incomplete.
Authors' response	We thank the reviewer for the sharp observation that some of the constrained MTR segments had mean MTR > 0.841. That was the result of an artifact in our previous segmentation method that affected 409 segments in 400 genes. The p-value of the Mann-Whitney U test was not significant enough to distinguish between these short segments and adjacent larger unconstrained segments. The means of some merged segments were therefore larger than 0.841 since they were dominated by the longer unconstrained segments. Using one global p-value threshold to differentiate these segments was incorrect for these 409 corner cases. Therefore, we have simplified the segmentation method and removed the step where we merged adjacent regions based on a p-value threshold for significance. The new MTR constrained regions are updated in Supplementary Table 4 and Supplementary Figs 9-10. We reran the Jaccard index calculation for identifying overlap with functional domains accordingly and made corresponding updates to Supplementary Table 5.
Changes in text	The new method is described in Supplementary Information, lines 381 – 387. Please note that this is a separate document.

Responses to Referee 3 Comments

Referee 3 Comment 1	The article is now quite wordy. Certain passages are technical to a degree that might not be appropriate for a general scientific audience). In my opinion, R2's previous comment that the paper would benefit from careful revision still applies (see examples below). Examples of imprecise or incorrect language: - "genomic positions that can lead to synonymous [...] changes" - positions do not lead to changes, variants do - "all possible pLOF variations within a gene" - all possible variants, not variations
Authors' response	We agree with the reviewer about the wordiness of the document and have moved technical details to Supplementary Information and streamlined the main text for improved clarity.
Changes in text Lines	Lines in text: We changed "genomic positions that can lead to synonymous [...] changes" to "synonymous [...] variants" (line 94). Here, we were highlighting the idea that X% of all possible mutated positions, i.e. chr-position pairs, were observed in the dataset.

Referee 3 Comment 2	Related to the above, the article describes numerous detailed technical comparisons of quantitative metrics. Some of these comparisons rely on arbitrary thresholds that could in principle be tweaked to obtain a desired result. I suspect there is a better, more elegant, and statistically grounded way to compare multiple scores with respect to an outcome (e.g., by modelling the variance explained by each score using a regression model). For example, the MTR vs MPC comparison uses thresholds to partition regions into constrained and unconstrained regions, and on that basis states that MPC would "misclassify" 168,815 (39.4%) of the 421,012 MTR-constrained codons. One could just as well state that MTR would misclassify MPC-constrained codons. A comparison between the two metrics in relation to a third objective outcome would be in order here.
Authors' response	Supplementary Figure 3 in our previous submission shows that irrespective of the threshold chosen for the different gene constraint metrics (S_{het}, $S_{het-ABC}$ and LOEUF), S_{het} derived from RGC-ME best discriminates between the high and low group of genes. The MTR threshold of the top 15-percentile (0.841) was chosen based on empirically derived MAPS scores suggesting that a missense variant with MTR=0.841 is as deleterious as missense 5/5 (Supplementary Figure 8). Among constrained MTR regions defined by the top 15-percentile threshold, we demonstrated enrichment of pathogenic ClinVar variants compared with benign and case de novo variants compared with control. Nevertheless, we agree that our results lack the basis for stating that MPC-segments "misclassified" regions and have removed this statement. Please refer to our response to Referee 2 Comment 2 for further discussion.
Changes in text	Lines in text: 229-241, 266-272 (Main Manuscript), New Supplementary Fig.11A,B

Referee 3 Comment 3	Clearly state whether a bar shows a confidence interval or the standard error of the mean throughout
Authors' response	We have included this detail in all figures.

- 1 Kaitlin E. Samocha, J. A. K., Konrad J. Karczewski, Anne H. O'Donnell-Luria, Emma Pierce-Hoffman, Daniel G. MacArthur, Benjamin M. Neale, Mark J. Daly. Regional missense constraint improves variant deleteriousness prediction. *bioRxiv*, doi:148353; doi: <https://doi.org/10.1101/148353> (2017).

Reviewer Reports on the Second Revision:

Referees' comments:

Referee #1 (Remarks to the Author):

The authors have addressed my comments. I only have one light suggestion for the new, much-improved section on recessive selection, which is that the authors should consider computing a p-value for their estimated 15% depletion of pLoF homozygotes. (This can be done using a Fisher's exact test; using a heuristic, I think the p-value should be close to 0.001).

Referee #2 (Remarks to the Author):

I thank the authors for adequately addressing my remaining concerns and commend them for the resource of homogeneously curated exome variant data they provide to the community.

Referee #3 made no comments to the authors.

Author Rebuttals to Second Revision:

Updates to the manuscript

We thank all the referees for their feedback. We have updated the manuscript in response to Referee 1's suggestion.

Response to Referee 1 Comments

Referee 1 Comment 1	The authors have addressed my comments. I only have one light suggestion for the new, much-improved section on recessive selection, which is that the authors should consider computing a p-value for their estimated 15% depletion of pLoF homozygotes. (This can be done using a Fisher's exact test; using a heuristic, I think the p-value should be close to 0.001).
Authors' response	This is a great suggestion. We have computed a p-value for the estimated depletion of pLOF homozygotes
Changes in text	Lines 293 – 295 in main manuscript While this number is much larger than HWE expectations, it is ~15% less than the expected 479 homozygotes calculated using an inbreeding coefficient of 0.37% ($p=0.0095$, Fisher's exact test). Lines 693 - 695 in supplementary information The number of observed pLOF homozygotes is 406, significantly lower than the expected counts of 479, taking the inbreeding coefficient into account ($p=0.0095$, one-tailed Fisher's exact test comparing pLOFs to missense).

Response to Referee 2 Comments

Referee 2 Comment 1	I thank the authors for adequately addressing my remaining concerns and commend them for the resource of homogeneously curated exome variant data they provide to the community.
Authors' response	We thank the reviewer for feedback and comments that helped to improve the manuscript.